# Sex- and region-specific cortical and hippocampal whole genome transcriptome profiles from control and APP/PS1 Alzheimer's disease mice

**Anna Papazoglou**[1], **Christina Henseler**[1], **Sandra Weickhardt**[2], **Jenni Teipelke**[1], **Panagiota Papazoglou**[1], **Johanna Daubner**[1], **Teresa Schiffer**[1], **Damian Krings**[1], **Karl Broich**[2], **Jürgen Hescheler**[3,4], **Agapios Sachinidis**[3,4], **Dan Ehninger**[5,6], **Catharina Scholl**[2], **Britta Haenisch**[2,6,7], **Marco Weiergräber**[1,2,3,4]*

**1** Experimental Neuropsychopharmacology, Federal Institute for Drugs and Medical Devices (Bundesinstitut für Arzneimittel und Medizinprodukte, BfArM), Bonn, Germany, **2** Federal Institute for Drugs and Medical Devices (Bundesinstitut für Arzneimittel und Medizinprodukte, BfArM), Bonn, Germany, **3** Faculty of Medicine, Institute of Neurophysiology, University of Cologne, Cologne, Germany, **4** Center of Physiology and Pathophysiology, Faculty of Medicine, University of Cologne, Cologne, Germany, **5** Translational Biogerontology, German Center for Neurodegenerative Diseases (Deutsches Zentrum für Neurodegenerative Erkrankungen, DZNE), Bonn, Germany, **6** German Center for Neurodegenerative Diseases (Deutsches Zentrum für Neurodegenerative Erkrankungen, DZNE), Bonn, Germany, **7** Center for Translational Medicine, Medical Faculty, University of Bonn, Bonn, Germany

* marco.weiergraeber@bfarm.de

**Data Availability Statement:** For full access to the original transcriptome data files, please refer to the Mendeley database (MENDELEY DATA (doi: 10.

## Abstract

A variety of Alzheimer's disease (AD) mouse models has been established and characterized within the last decades. To get an integrative view of the sophisticated etiopathogenesis of AD, whole genome transcriptome studies turned out to be indispensable. Here we carried out microarray data collection based on RNA extracted from the retrosplenial cortex and hippocampus of age-matched, eight months old male and female APP/PS1 AD mice and control animals to perform sex- and brain region specific analysis of transcriptome profiles. The results of our studies reveal novel, detailed insight into differentially expressed signature genes and related fold changes in the individual APP/PS1 subgroups. Gene ontology and Venn analysis unmasked that intersectional, upregulated genes were predominantly involved in, e.g., activation of microglial, astrocytic and neutrophilic cells, innate immune response/immune effector response, neuroinflammation, phagosome/proteasome activation, and synaptic transmission. The number of (intersectional) downregulated genes was substantially less in the different subgroups and related GO categories included, e.g., the synaptic vesicle docking/fusion machinery, synaptic transmission, rRNA processing, ubiquitination, proteasome degradation, histone modification and cellular senescence. Importantly, this is the first study to systematically unravel sex- and brain region-specific transcriptome fingerprints/signature genes in APP/PS1 mice. The latter will be of central relevance in future preclinical and clinical AD related studies, biomarker characterization and personalized medicinal approaches.

17632/z9264694b4.1), direct URL to data: https://data.mendeley.com/datasets/z9264694b4/1).

**Funding:** The study was internally funded by the Federal Institute for Drugs and Medical Devices (Bundesinstitut für Arzneimittel und Medizinprodukte, BfArM, Bonn, Germany). The funders had no role in study design, data collection and analysis, decision to publish, or preparation of the manuscript

**Competing interests:** The authors have declared that no competing interests exist.

**Abbreviations:** A, amyloid beta; ASC, apoptosis-associated speck-like protein containing a CARD; AD, Alzheimer's disease; APP, amyloid precursor protein; BACE, -site amyloid precursor protein cleaving enzyme 1; BROI, brain region of interest; CARD, a C-terminal caspase-recruitment domain (CARD); Cd, cluster of differentiation; CNRQ, calibrated normalized relative quantity; CNS, central nervous system; CSF, cerebrospinal fluid; Cx, cortex; cRNA, complementary RNA; Ct, cycle threshold; DAM, disease-activated microglia; DEGs, differentially expressed genes; $ddH_2O$, double distilled water; ECoG, electrocorticogram; EEG, electroencephalogram; EHipG, electrohippocampogram; FAD, familial Alzheimer's disease; FC, fold change; FFT, Fast-Fourier-Transformation; GO, gene ontology; GWAS, genome-wide association studies; Hip, hippocampus; HPRT, hypoxanthine-guanine-phosphoribosyltransferase; i.p., intraperitoneal; KEGG, Kyoto Encyclopedia of Genes and Genomes; l(i)ncRNA, long (interfering) non-coding RNA; MCODE, Molecular Complex Detection; mRNA, messenger RNA; PCA, principal component analysis; PPI, protein-protein interaction; PrP, prion protein; PS, presenilin; PYD, N-terminal PYRIN-PAAD-DAPIN domain; qPCR, quantitative polymerase chain reaction; ROS, reactive oxygen species; RS, retrosplenial; RT, reverse transcriptase; SEM, standard error of the mean; WT, wild-type.

## Introduction

Alzheimer's disease (AD) is a complex neurodegenerative disorder accompanied by progressive cognitive decline, memory deterioration, neuropsychiatric symptoms and eventually death [1, 2]. From a pathophysiological point of view, the formation and accumulation of amyloid beta (Aβ) together with the extracellular deposits of amyloid plaques represent one of the hallmarks and devastating histopathological characteristics of AD in the central nervous system (CNS) [3, 4]. Mechanistically, amyloid peptides of different lengths originate from a common biochemical substrate, i.e., the amyloid precursor protein (APP), upon sequential cleavage steps. Two specific secretases are responsible for these endoproteolysis/cleavage steps, i.e., β-site APP cleaving enzyme 1 (BACE-1), and γ-secretase [5, 6]. Notably, pro-amyloidogenic APP mutations can enhance the release of cytotoxic Aβ peptide fragments of different length. For example, the Swedish double mutation KM670/671NL aggravates BACE cleavage and increases the abundance of both $A\beta_{1-40}$ and $A\beta_{1-42}$, thereby mediating pro-amyloidogenic effects [7]. Furthermore, other common mutations exist that affect, e.g., presenilin (PS)1 and PS2. Presenilins act as catalytic sites for the γ-secretase and can severely trigger the $A\beta_{1-42}$ formation and aggravate pro-amyloidogenicity [8]. The common Aβ theory of AD in humans is complemented by the intraneuronal formation of hyperphosphorylated tau (τ) protein, finally evolving into neurofibrillary tangles [9, 10]. However, there is increasing evidence from preclinical, clinical and drug research and development studies that cutting the sophisticated etiopathogenesis of AD down to Aβ- and τ- formation might be an oversimplification, misleading and too shortsighted [11–16]. Indeed, multiple studies in the last decades elicited that AD is a multifactorial disease that can be triggered/modified by highly diverse factors involving numerous molecular, cellular and supra cellular pathways acting on the neuronal nano-, micro-, meso- and macro-scale [17, 18]. Although some of these results were obtained from human AD patient brain tissue, the majority of related experimental data was obtained from rodents, particularly AD mouse models [19–22].

More than 100 AD mouse models have been generated so far and many of them exhibit typical age- and AD-related structural and functional abnormalities, e.g., Aβ plaques, axonal and synaptic dystrophy, reduced synaptic plasticity, and disturbance of learning and memory [23–25]. Importantly, depending on the mutagenesis/transgenesis strategy carried out in the individual mouse lines, these AD mouse models can significantly differ in histopathological features, symptomatic profile, disease progression characteristics and mortality. Recapitulatory, AD mouse models should fulfill fundamental translational categories, i.e., isomorphism, homology, and predictability to a maximum extent [26]. Examples of well-established AD mouse lines include, i.a., the 5XFAD, 3xTg and APP/PS1 models. The 5XFAD animals with co-integrated transgenes bred as a single allele, represent a pathophysiologically "aggressive" model of AD exerting juvenile-onset amyloid pathology (∼ 3 months) with a rapid onset phenotype. However, 5XFAD mice represent a non-physiological combination of familial Alzheimer's disease (FAD) mutations with marked intracellular Aβ accumulation [22]. The 3xTg line mimics an early- to mid-life amyloid pathology model that includes hyperphosphorylated tau (τ). It captures both Aβ and phospho-τ features of AD and exhibits variable pathology between colonies and sexes. In addition, genetic drift has been observed [22]. The APP/PS1 model exerts early-onset (∼6 months) amyloid pathology. It is well-characterized and co-integrated transgenes are bred as a single allele, which however, makes it hard to control for independent transgene effect [22].

In our study, an APPswePS1dE9 (APP/PS1) AD mouse model has been utilized, carrying the human APP with Swedish double mutation (APPswe) co-integrated with human PS1 with exon 9 deletion (PS1dE9) [11–14, 27, 28]. These mutations result in an overproduction of APP and PS1 splice variants and a subsequent neural Aβ overload. APPswePS1dE9 mice exhibit a mortality of 10–15% [29]. Cognitive/memory impairment and behavioral deficits observed, e.g., in spatial navigation tasks, reference learning, Morris' water maze and radial arm maze, become obvious at an age of 6 to 12 months of age [30]. Furthermore, seizure activity is often encountered in AD mouse mutants including APP/PS1 and intracerebral Aβ accumulation was shown to be linked to epileptogenesis [13, 14, 31, 32]. Thus, seizure activity might be responsible for sudden death in this mouse line as well [31, 33]. Overall, the APP/PS1 AD line represents a well-established model of high value in translational terms [22].

We previously demonstrated that neurodegenerative alterations in the motor cortex and septohippocampal system of APP/PS1 mice (but also in other AD lines such as 5XFAD) can result in complex central dysrhythmia [32], particularly in the theta (4–8 Hz) and two distinct gamma frequency ranges (30–50 Hz and 50–70 Hz) [13, 14]. Importantly, these systemic *in vivo* macro-scale electrophysiological findings in the electrocorticogram (ECoG) and electro-hippocampogram (EHipG) of APP/PS1 mice, turned out to be clearly age-, sex- and brain region-specific and demonstrated that subgroup-specific analysis in AD research is indispensable [13, 14]. The latter has substantial implications for AD drug research and development and individualized/personalized pharmacological AD treatment. Previous studies in APP/PS1 mice further revealed selective electrical activity and individual frequency characteristics in ECoGs and other deflections related to circadian rhythmicity and sleep [6, 29, 30, 33–35].

To get an overall insight into the multifactorial, pathophysiological alterations in APP/PS1 mice, a detailed investigation of genome-wide transcriptional alterations in both sexes and different brain regions of interest (BROIs) is essential. Clearly, transcriptome studies in human AD patient CNS material had been carried out in the past [36] and the same holds true for various rat and mouse AD models [36]. However, many of these studies, e.g., in APP/PS1 mice, encounter specific limitations: (i) female and male animals were pooled (in a balanced or unbalanced way) and were not investigated in a sex-specific manner or there was no information provided about the sex distribution [37–40], (ii) only one sex (generally males) was studied [41–43], (iii) the entire brain was investigated and not individual BROIs, although different brain regions can be differentially affected by AD pathogenesis [38]. Thus, the complex sex- and region-specific investigation of transcriptomes of AD models such as APP/PS1 mice, is mandatory [36, 44–46].

Here we present for the first time, transcriptome results from the retrosplenial (RS) cortex and hippocampus of age-matched (8 months old) female and male wild-type (WT) control and APP/PS1 AD mice. The hippocampus as an interface structure for learning and memory processes [47–49] and the RS cortex involved in spatial memory and environmental orientation [50, 51] are early affected in mild cognitive impairment (MCI) and manifest AD and therefore represent major BROIs in AD pathophysiology [52, 53]. Our results provide the first sex-specific and hippocampal/RS cortex-specific analysis of transcriptional alterations in APP/PS1 AD mice. We enable insight into co-upregulated and co-downregulated (intersectional) gene candidates and those differentially expressed genes (DEGs) that serve as signature/fingerprint gene sets in individual BROIs of males or females. We further demonstrate that DEGs in the individual subgroups are primarily related to the fields of microglia activation, immunological response, inflammation, synaptic integration, learning and memory and ictogenesis in AD. In addition, we also analyzed long (intergenetic) non-coding RNAs (l(i)ncRNA, > 200 nucleotides) that impact on the regulation of (post)transcriptional, translational and epigenetic phenomena in AD [54–57].

In summary, our data allow detailed insight into the sex- and tissue-specific alterations in metabolic/biochemical, immunological, inflammatory, neurodegenerative, and excitatory processes related to AD pathobiochemistry and pathophysiology.

## Materials and methods

### Experimental animals

Transcriptome data were obtained from double transgenic APPswePS1dE9 (APP/PS1) mice with a C57BL/6J background. This AD mouse line carries a chimeric mouse/human APP with two Swedish mutations (APPswe) co-integrated with human PS1 with exon 9 deletion (PS1-dE9) [13, 14, 58]. Mutant mice (B6.Cg-Tg(APPswe, PSEN1dE9)85Dbo/Mmjax, MMRRC stock no. 34832-JAX) and their WT littermates were purchased from Jackson Laboratory (Germany). In total, eight control animals (four ♂, age: 32.72 ± 0.38 wks; four ♀, age: 32.14 ± 0.25 wks) and eight APPswePS1dE9 (APP/PS1) mice (three ♂, age: 32.81 ± 0.24 wks; five ♀, age: 32.66 ± 0.39 wks) were utilized for hippocampal and cortical extirpation and subsequent transcriptome analysis. Importantly, both sexes were included in our study design and analyzed separately (no unspecified pooled sexes or balanced sex distribution in the individual subgroups). Note that we cannot comment on the status of the estrous cycle of the female experimental animals in our study. In general, female mice at 8 months of age are still fertile, but not suggested to be used for breeding anymore.

All experimental mice were housed in groups of 3–4 in clear Makrolon cages type II with *ad libitum* access to drinking water and standard food pellets under pathogen-free conditions. Mice were maintained inside ventilated cabinets (Type Uniprotect, Zoonlab, Germany) at an ambient temperature of 21 ± 2°C, 50–60% relative humidity, and on a conventional 12 h/12 h light/dark cycle beginning at 5:00 am. All animals were strictly adapted to the circadian pattern preceding cortical and hippocampal extirpation and RNA isolation (see below).

All animal procedures were performed in accordance with the Guidelines of the German Council on Animal Care and all protocols were approved by the Local Institutional and National Committee on Animal Care (Landesamt für Natur, Umwelt und Verbraucherschutz, LANUV, Germany). The authors further certify that all animal experimentation complied with the ARRIVE guidelines and were carried out in accordance with the U.K. Animals (Scientific Procedures) Act, 1986 and associated guidelines; EU Directive 2010/63/EU for animal experiments; or the National Institutes of Health guide for the care and use of laboratory animals (NIH Publications No. 8023, revised 1978). Maximum effort was made to reduce the number of animals necessary to obtain data and suffering of the animals according to the 3R strategy.

### Genotyping - DNA preparation from tail biopsies

Every experimental animal was genotyped twice using DNA isolated from tail biopsies. DNA preparation was carried out using peqGOLD DNA Mini Kit (PEQLAB Biotechnologie GmbH, Germany) according to the manufacturer's instructions. The isolated genomic DNA was stored at +4°C until further use.

### Genotyping - Polymerase Chain Reaction (PCR)

For every sample, a PCR reaction mix containing three different primer pairs (APP forward: 5'-AGGACTGACCACTCGACCAG-3'; APP reverse: 5'-CGGGGGTCTAGTTCTGC-3'; PS1 forward: 5'-AATAGAGAACGGCAGGAGCA-3'; PS1 reverse: 5'-GCCATGAGGGCACTAATCAT-3'; WT control forward (muscarinic receptor 5, *Chrm5*) 5'- ACCTTGGACCAAATCTGAGTGTA-3';

WT control reverse (muscarinic receptor 5, *Chrm5*): 5'- GGCCAAGCTGAGCAGGTAAT-3'), ddH$_2$O (PCR grade), Red Taq Ready Master Mix (Sigma Aldrich Chemie GmbH, Germany) and isolated sample-specific genomic DNA ($\sim$ 20 ng/ml) was prepared (**S1 Table in S1 File**).

The PCR reaction mix was gently vortexed for 3 sec followed by a brief centrifugation step for 5 sec at 2000xg using a micro centrifuge (ROTILABO, Carl Roth GmbH & Co. KG, Germany). For each genotyping PCR, positive controls (WT DNA and DNA from a validated mutant mouse) and a negative control (no DNA) were added for experimental validation. PCRs were carried out using a Bio-Rad thermal cycler (type C1000, Bio-Rad Laboratories GmbH, Germany) and the following amplification parameters were applied: 3 min pre-incubation at 94˚C, 35 cycles (each cycle composed of 94˚C, 30 sec denaturation; 48˚C, 30 sec annealing; 72˚C, 1 min extension) and finally 10 min incubation at 72˚C. The amplification mix was stored at 4˚C till further use.

PCR products were analyzed by horizontal agarose gel (1.5% in 0.5x TBE buffer, pH8) electrophoresis and visualized by ethidium bromide (0.3 µg/ml). The ChemiDoc Touch System (Bio-Rad Laboratories GmbH, Germany) was used for gel imaging and identification of the amplified DNA fragments (**S1 Fig in S1 File**).

## RS cortex and hippocampus preparation

Experimental animals were deeply anaesthetized using i.p. injection of ketamine (100 mg/kg) / xylazine (10 mg/kg). To ensure that the animals were fully anaesthetized, the absence of tail and foot pinch reflexes was verified. Animals were decapitated, the brain was removed immediately and placed in a clean RNase-free petri-dish kept on ice and filled with pure RNAlater reagent (Qiagen GmbH, Germany). By using a scalpel, forceps and a thin brush, the whole hippocampus and a piece (2–3 mm$^3$) of the RS cortex were dissected from both brain hemispheres of each experimental animal (**S2 Fig in S1 File**). The tissue fragments were placed in a 2 ml RNase free reaction tube, snap frozen in liquid nitrogen and stored at -80˚C until RNA preparation. To avoid potential interference of transcriptional profiles with circadian rhythmicity, tissue preparation was always performed *ante meridiem* between 8 am and 11 am.

## RNA isolation from the RS cortex and hippocampus

Total RNA was isolated using the RNeasy Lipid Tissue Mini Kit (Qiagen GmbH, Germany) according to the manufacturer's instructions (incl. an optional DNase digestion step). Cortical and hippocampal tissue samples were removed from -80˚C, immediately lysed in QIAzol lysis buffer (Qiagen GmbH, Germany) and homogenized using the Tissue Ruptor (Qiagen GmbH, Germany), a handheld-rotor-stator homogenizer with disposable probes. Following phenol-chloroform separation, DNase digestion and three washing steps, total RNA was eluted in 30 µl RNase-free ddH$_2$O. The quantity and quality of the eluted RNA was checked using Nano-Drop$^®$ ND-1000 (Thermo Fisher Scientific, USA) according to the manufacturer's instructions. The ratio of absorbance at 260 and 280 nm (260/280) was used to assess RNA purity with a ratio of $\sim$ 2.0 being generally accepted as "pure" for RNA related experimental approaches (see NanoDrop$^®$ user manual). The ratio of absorbance at 260 and 230 nm (260/230) is a secondary measure of nucleic acid purity. They are commonly in the range of 1.8–2.2 (see NanoDrop$^®$ user manual). Our RNA probes from APP/PS1 and control BROI samples for microarray and qPCR experiments exhibited absorbance ratios of $\sim$ 2.0 (for 260 nm/280 nm) and of $\sim$ 2.05 (for 260 nm/230 nm).

## One-color microarray-based gene expression analysis

The *One-Color Microarray-Based Gene Expression* system by Agilent Technologies Germany GmbH & Co. KG (Germany) utilizes the *Low Input Quick Amp Labeling Kit* to generate cDNA (1$^{st}$ and 2$^{nd}$ strand cDNA) via AffinityScript-RT (reverse transcriptase) which is a genetically engineered, highly thermostable version of *Moloney Murine Leukemia Virus* (MMLV) enzyme reverse transcriptase (RT). Subsequently, the samples were labeled with a T7 RNA polymerase blend. This polymerase incorporates Cyanine 3-CTP during amplification, generating a one-color fluorescent complimentary RNA (cRNA) as target material. The RNA sample input is supposed to range from 10–200 ng (for details on the procedure, see manufacturer's instructions in the *One-Color RNA Spike-In Kit* and *Low Input Quick Amp Labeling Kit*, both from Agilent Technologies Germany GmbH & Co. KG (Germany). The amplification from total RNA to amplified cRNA is typically around 100-fold. Subsequently, the labeled/amplified cRNA was purified using the *RNeasy Mini Kit* (Qiagen GmbH, Germany) followed by quantification of the cRNA using the NanoDrop ND-1000 UV-VIS spectrophotometer (Thermo-Fisher Scientific, USA). After finalization of cRNA sample preparation, hybridization of samples was carried out for 17 hrs (65˚C) using the *Gene Expression Hybridization Kit* (Agilent Technologies Germany GmbH & Co. KG, Germany) and the *SurePrint G3 Mouse Gene Expression v2 8x60K Microarray Kit* (Agilent Technologies Germany GmbH & Co. KG, Germany) according to the manufacturer's instructions. Key features of the *SurePrint G3 Mouse Gene Expression v2 8x60K Microarray* chip (Agilent Technologies Germany GmbH & Co. KG, Germany) include up-to-date content sourced from RefSeq, Ensembl, RIKEN, UniGene, and GenBank databases to provide full coverage of the mouse transcriptome, wide dynamic range of over 5 logs to ensure detection of low and high expressors for the most biologically relevant data and the possibility to proceed with one or two colors based on experimental needs. The specific *SurePrint G3 Mouse Gene Expression v2 8x60K Microarray Kit* used in our study is referred to as G4852B (Agilent Product Number). In our study, the microarray setup carries eight arrays on a slide in a 4 x 2 grid, with each array covering 62,976 features arranged in 1,064 rows and 170 columns (8 x 60k format). The biological features include 27457 Entrez Genes (unique) and 4578 l(i)ncRNAs (unique). 96 ERCC and 10 E1A served as control probes. To minimize inter-run variability, we always performed hybridization of the maximum microarray load. In addition, we did not process subgroup specific probes as a whole in one hybridization step. Instead, probes from the individual subgroups were randomly distributed on the eight slots hybridization slide to prevent potential subgroup-specific inter-run confounders.

After the washing procedure with *Gene Expression Wash Buffer Kit* (Agilent Technologies Germany GmbH & Co. KG, Germany) to remove unspecific bindings, the microarrays were prepared for scanning and feature extraction. The microarray scan was performed using the *Agilent SureScan Microarray Scanner* (Agilent Technologies Germany GmbH & Co. KG, Germany). *Feature Extraction Software* (Agilent Technologies Germany GmbH & Co. KG, Germany) was used to extract the information from the probe features of the microarray scan data, providing information about gene expression/transcripts for further analysis. Details on the materials and software used here are also listed in **S2 and S3 Tables in S1 File**.

Please note that for transparency reasons, the raw read out data files were extracted as txt.-files/csv.-files and are freely available at the Mendeley Data repository (Weiergräber, Marco (2023), "Whole genome transcriptome data from the retrosplenial cortex and hippocampus of female and male control and APP/PS1 Alzheimer's disease mice.", Mendeley Data, V2, doi: 10.17632/z9264694b4.2 [59].

## Experimental transcriptome study groups

In this transcriptome study, APP/PS1 AD and WT controls were compared in four individual settings, i.e., ♀ Rs cortex, ♀ hippocampus, ♂ RS cortex and ♂ hippocampus. The primary focus is on genotype-, sex- and region-specific alterations in DEGs and l(i)ncRNAs in the individual subgroups.

## Transcriptome analysis

**Data extraction, DEGs and l(i)ncRNA characterization parameters.** The *Feature Extraction Software* (Agilent Technologies Germany GmbH & Co. KG, Germany) was used to extract relevant information from the probe features of the microarray scan data, providing information about gene expression/transcripts for further analysis. The DEGs were extracted including related probe name, gene symbol (Primary HUGO gene symbol), gene name, gene description, sequence and GO terms, p-value, and fold change (FC). Note that the significance of DEGs was tested using one-way ANOVA and *post hoc* analysis using Tukey HSD (honestly significant difference) test. In addition, the Benjamini-Hochberg procedure for decrease of the false discovery rate (FDR) was applied [60].

**Principal component analysis (PCA) and heatmap chartography/hierarchical clustering.** The PCA and heatmap generation/hierarchical clustering was carried out using Gene-Spring GX 14.9 (Agilent Technologies Germany GmbH & Co. KG, Germany). In the PCA, our large multi-dimensional transcriptome data sets are depicted as 3D scatter plots to visualize potential clustering of subgroup triplicates/quadruplicates/quintuplicates. The coordinates of the X-, Y- and Z-axis represent the so-called PC scores.

**Gene ontology (GO) and enrichment analysis - protein-protein interaction (PPI) and related pathways.** GO analysis (Gene Ontology Consortium; http://www.geneontology.org) including the major fields of biological processes, cellular components, and molecular functions, was utilized to characterize the properties of DEGs and l(i)ncRNAs in the RS cortex and hippocampus of male and female APP/PS1 AD mice in comparison to WT animals and to identify gene sets related to annotated GO terms that are over- or under-represented. For pathway and process enrichment analysis, the Metascape software (https://metascape.org; [61]) had been used. For each list of DEGs extracted via *Feature Extraction Software* (Agilent Technologies Germany GmbH & Co. KG, Germany), complex pathway and process enrichment analyses have been performed based on the following ontology sources: GO Biological Processes, KEGG Pathway (Kyoto Encyclopedia of Genes and Genomes; http://www.genome.jp/kegg/), Reactome Gene Sets, and WikiPathways [61]. The latter were applied to identify biological enriched pathways of AD-related mRNA and l(i)ncRNA alterations detected in the microarray experiments. As enrichment background, the entire genes/genome had been used. Individual terms with a p-value < 0.01, a minimum count of 3, and an enrichment factor > 1.5 were characterized and grouped into clusters due to their membership similarities. Note that the enrichment factor is defined as the ratio between the observed counts and the expected random counts. The p-values were calculated based on the cumulative hypergeometric distribution [62]. For further procedural details, please refer to the Metascape software documentation (https://metascape.org; [61]).

Using the Metascape software, top clusters with their representative enriched terms (one per cluster) were defined. Only input genes with at least one ontology term annotation were included in the calculation. Significance is depicted as the log10(P) (https://metascape.org; [61]).

In addition, a PPI enrichment analysis was performed using also Metascape. For each given list of DEGs, the PPI enrichment analysis included the following databases: STRING [63], Bio-Grid [64], OmniPath [65], and InWeb_IM [66]. Only physical interactions in STRING

(physical score > 0.132) and BioGrid were considered. The resultant network comprises the subset of proteins that establish physical interactions with at least one other member in the list. If the network contains between 3 and 500 proteins, the Molecular Complex Detection (MCODE) algorithm [67] has been applied to identify densely connected network components (https://metascape.org; [61]).

Pathway and process enrichment analysis has been applied to each MCODE component independently, and the three best-scoring terms by p-value have been retained as the functional description of the corresponding components. The related p-values represent the probability of seeing at least x genes out of the total number of n genes in the list annotated to a particular GO term, given the proportion of genes in the whole genome that are annotated to that GO term. Given different p-values and n values, the related annotated number x of genes varies. The closer the p-value is to zero, the more significant is the particular GO term associated with the group of genes.

Note that GO analysis was conducted for upregulated genes with a FC >1.5. For the downregulated genes, a FC < -1.5 did not result in significantly enriched terms due to the low number of significantly downregulated genes. To get an impression of enriched terms we selectively lowered the FC cutoff for downregulated genes in the GO setting to < -1.2.

**Venn visualization and analysis.** Intersectional, i.e., co-upregulated or co-downregulated DEGs in the RS cortex and hippocampus of female APP/PS1 AD vs. WT mice, the RS cortex and hippocampus of male APP/PS1 AD vs. WT mice, the RS cortex of male and female APP/PS1 AD vs. WT animals and the hippocampus of male and female APP/PS1 AD vs. WT mice were characterized using the Venn diagram function. The latter was performed using GeneSpring GX 14.9 (Agilent Technologies Germany GmbH & Co. KG, Germany) or VENNY2.1 (https://bioinfogp.cnb.csic.es/tools/venny/index.html; [68]). A specific focus was set on signature genes/fingerprint gene sets, exclusively up- or downregulated in the individual subgroups. Note that for comparability reasons, the FC cutoff was set > 1.5 for upregulated genes and < -1.5 for downregulated genes.

**Statistical analysis - general aspects.** Unless otherwise specified, data were analyzed and presented in Microsoft Excel and GraphPad Prism (Version 9.1.2). Statistics were carried out as outlined above for DEGs and l(i)ncRNAs. Unless otherwise stated (see section 2.8.3.), a p-value < 0.05 was considered significant (*). In many analytical settings (see above), p-values are presented as (-)log10(P) for better visualization.

## Quantitative real-time PCR (qPCR)

Two quantitative real-time PCR (qPCR) approaches were carried out: the first was used to unravel whether selected DEGs in another AD mouse model, i.e., 5XFAD, are also differentially up- or downregulated in the APP/PS1 model [13, 14, 31, 32]. This approach focusses on AD line-specific and cross-line DEGs. For this purpose, qPCRs were carried out in eight WT controls (four ♂, age 32.71 ± 0.37 wks, four ♀, age 32.93 ± 0.21 wks) and eight APP/PS1 AD mice (four ♂, age 32.53 ± 0.32 wks, four ♀, age 33,29 ± 0.00 wks). Note, that the aforementioned animals were exclusively used for this approach and not for our microarray studies. The second qPCR approach was carried out to validate a selected number of cortical DEGs (*Siglech*, *Ptpn6*, *Laptm5*, *Plek*, *Arpp21*, *Shisa9*) from our transcriptome results. Here, RNA was used from the same animals as utilized for our microarray studies (eight APP/PS1 AD mice (four ♂, four ♀) and eight WT control animals (three ♂, five ♀), see above).

The cDNA synthesis from all RNA samples was performed using anchored-oligo(dt)18 and hexamer primer in a two-step reverse transcriptase (RT) PCR approach (*Transcriptor First Strand cDNA Synthesis Kit*, Qiagen GmbH, Germany). The qPCR reaction protocol was based

on LightCycler 480 SYBR Green I Master (Roche, Roche Life Science, Germany). For details on primers for the specific gene candidates (*Cacna1c*, *Cacna1d*, *Plcd4*, *Casp8*, *Chrm1*, *Chrm3*, *Chrm5*, *Siglech*, *Ptpn6*, *Laptm5*, *Plek*, *Arpp21*, *Shisa9*) please refer to Siwek et al. (2015) [32] and **S4 Table in S1 File**. The qPCR was performed in a Light Cycler 480 System (Roche Life Science, Germany) thermocycler. The following cycler protocol was conducted for all primer pairs: 95˚C (10 min, pre-incubation step); 95˚C (10 s, melting step); 60˚C (20 s, annealing step); and 72˚C (30 s, extension step), 35 cycles. Melting curve analysis was performed to evaluate the specificity of the amplification. Product detection and characterization was based on SYBR Green I Master (Roche Life Science, Germany), a ready-to-use hot start reaction mix containing FastStart Taq DNA Polymerase and DNA double-strand-specific SYBR Green I dye. Deionized, nuclease-free water (no cDNA) and total RNA samples (RT excluded) were used as negative controls. The *Hprt* was used as an internal reference gene (positive control). The Ct (cycle threshold) values were calculated using the LightCycler 480 System software (Roche Life Science, Germany). The FCs of *Cacna1c*, *Cacna1d*, *Plcd4*, *Casp8*, *Chrm1*, *Chrm3*, and *Chrm5* gene transcripts for the first qPCR approach and *Siglech*, *Ptpn6*, *Laptm5*, *Plek*, *Arpp21*, *Shisa9* for the second qPCR approach in APP/PS1 transgenic mice related to WT controls were calculated according to Schmittgen & Livak, 2008 [69]. The Cp values provided by the LightCycler 480 software (Roche, Germany) were imported to qBase+ software (Biogazelle, Belgium) and analyzed based on a delta-Cq quantification model with PCR efficiency correction, reference gene normalization considering the reference target stability of the selected housekeeping gene (*Hprt*) and inter-run calibration. The results were calculated as Calibrated Normalized Relative Quantity (CNRQ) and statistically investigated by the Mann-Whitney test.

## Results

### Principal component analysis

PCA analysis as a data quality control measure was used to reduce the dimensionality of the large multi-dimensional transcriptome data sets, to increase their interpretability and to limit information loss at the same time (see 3D scatter plot in **S3A Fig in S1 File**). In these complex transcriptome data sets, the genes represent the variables and PCA allows us to get an impression of the similarity between samples. Based on a transformation of the high-dimensional transcriptome data into an orthogonal basis, the first principal component is aligned with the largest source of variance, the second principal component to the largest remaining source of variance and so on. Consequently, the similarities of the individual transcriptome data sets in individual subgroups are more amenable to visual exploration.

The PCA analysis in our study revealed that the individual samples related to the specific subgroups exhibit little separation between the groups of replicates (triplicates, quadruplicates, quintuplicates) and most of them cluster together and separate from arrays of other subgroups (**S3A Fig in S1 File**). It should be acknowledged here that the experimental animals are prone to increased variability *per se*. For example, the animals are housed in groups of 3–4 exhibiting characteristic social hierarchy, which can potentially exert substantial influence on CNS transcriptomics [70]. Dominant versus subordinate male mice were reported to display distinct transcriptome patterns, particularly related to the immune and inflammatory system [70]. The latter had been attributed to the higher corticosterone and stress levels in subordinate mice compared to alpha animals [70]. Importantly, hierarchical phenomena are also present in female mice. The female estrus cycle does not seem to exert significant effects on aggressive behavior, although dominant females exhibit an elongated estrus cycle in comparison to subordinate females [71, 72]. Whereas the plasma estradiol levels were similar between dominant and subordinate females, subordinate females displayed largely elevated basal corticosterone

levels compared to dominant females [71]. Thus, the biochemical and physiological implications of social hierarchy need to be considered as an inevitable confounding factor.

PCA results were also plotted for genotype comparison (control vs. APP/PS1 samples, **S3B Fig in S1 File**), for sex comparison (male vs. female samples, **S3C Fig in S1 File**) and for brain region comparison (Rs cortex vs. hippocampus samples, **S3D Fig in S1 File**). Notably, there is a grouping tendency based on genotype, sex and BROI.

## Heatmaps/hierachical clustering - chartography of DEGs

To visualize overall transcriptome readouts and the transcript/expression profile of genes across the individual subgroups, we performed a heatmap presentation. The latter was complemented by hierarchical clustering to elicit similarities and dissimilarities of transcription profiles between the individual subgroups (**S4 Fig in S1 File**).

## DEGs in the RS cortex and hippocampus of APP/PS1 AD vs. WT mice of both sexes

**RS cortex of female APP/PS1 mice.** Using the Feature Extraction and GeneSpring RT software (both Agilent Technologies Germany GmbH & Co. KG, Germany), DEGs and l(i) ncRNAs were characterized based on one-way ANOVA testing with Tukey *post hoc* analysis and the application of the Benjamini-Hochberg procedure for FDR determination. As depicted in **S5 Table in S1 File**, a total number of 273 genes and l(i)ncRNAs were found including four candidates with a FC < -1.5 and 104 candidates with a FC > 1.5. The majority (85.35%) of DEGs was upregulated in this subgroup (233↑/40↓) (**Fig 1A**). The predominance of significantly upregulated gene candidates is also confirmed in the Volcano plot (**Fig 2A**).

The top 30 DEGs (FC > 1.5 and FC < -1.5, respectively) are displayed in **S5A Fig in S1 File**. The following three FC candidate genes were downregulated, i.e., *Ptpn6, Pisd-ps3 and LOC105247294*. The following high FC candidate genes (top 10) were upregulated, i.e., *Ccl6, Ifi27l2a, Cd52, Bcl2a1d, Prnp, Lyz2, C4b, Tyrobp, Slamf9, Lyz1*.

**Hippocampus of female APP/PS1 mice.** Using the Feature Extraction and GeneSpring RT software (both Agilent Technologies Germany GmbH & Co. KG, Germany), DEGs were defined based on one-way ANOVA testing including Tukey *post hoc* test and the application of the Benjamini-Hochberg procedure for control of FDR. As depicted in **S6 Table in S1 File**, a total number of 458 DEGs including l(i)ncRNAs were found including 15 candidates with a FC < -1.5 and 88 candidates with a FC > 1.5. However, the majority (53.06%) of DEGs was downregulated in this subgroup (215↑/243↓) (**Figs 1A and 2B**).

The top 30 DEGs (FC > 1.5 and FC < -1.5, respectively) are displayed in **S5B Fig in S1 File**. The following high FC candidate genes (top 10) were downregulated, i.e., *BC030499, Rims3, Pcsk1, Pdzph1, Etnppl, Pisd-ps3, Myh8, Spag5, Myh7b, Prr16*. The upregulated top 10 high FC candidate genes comprised: *Ccl6, Cd52, Lyz2, Prnp, Lyz1, Bcl2a1d, 5830408C22Rik, C4b, Slamf9, Aspg*.

**RS cortex of male APP/PS1 mice.** The Feature Extraction and GeneSpring RT software (both Agilent Technologies Germany GmbH & Co. KG, Germany) were used to characterize the DEGs via one-way ANOVA testing, Tukey *post hoc* analysis and subsequent Benjamini-Hochberg procedure for control of FDR. As depicted in **S7 Table in S1 File**, a total number of 236 gene candidates including l(i)ncRNAs were found with five candidates exhibiting a FC < -1.5 and 63 candidates displaying a FC > 1.5. The majority (60.17%) of DEGs was upregulated in this subgroup (142↑/94↓) (**Fig 1A**). The latter also becomes apparent in the Volcano plot (**Fig 2C**).

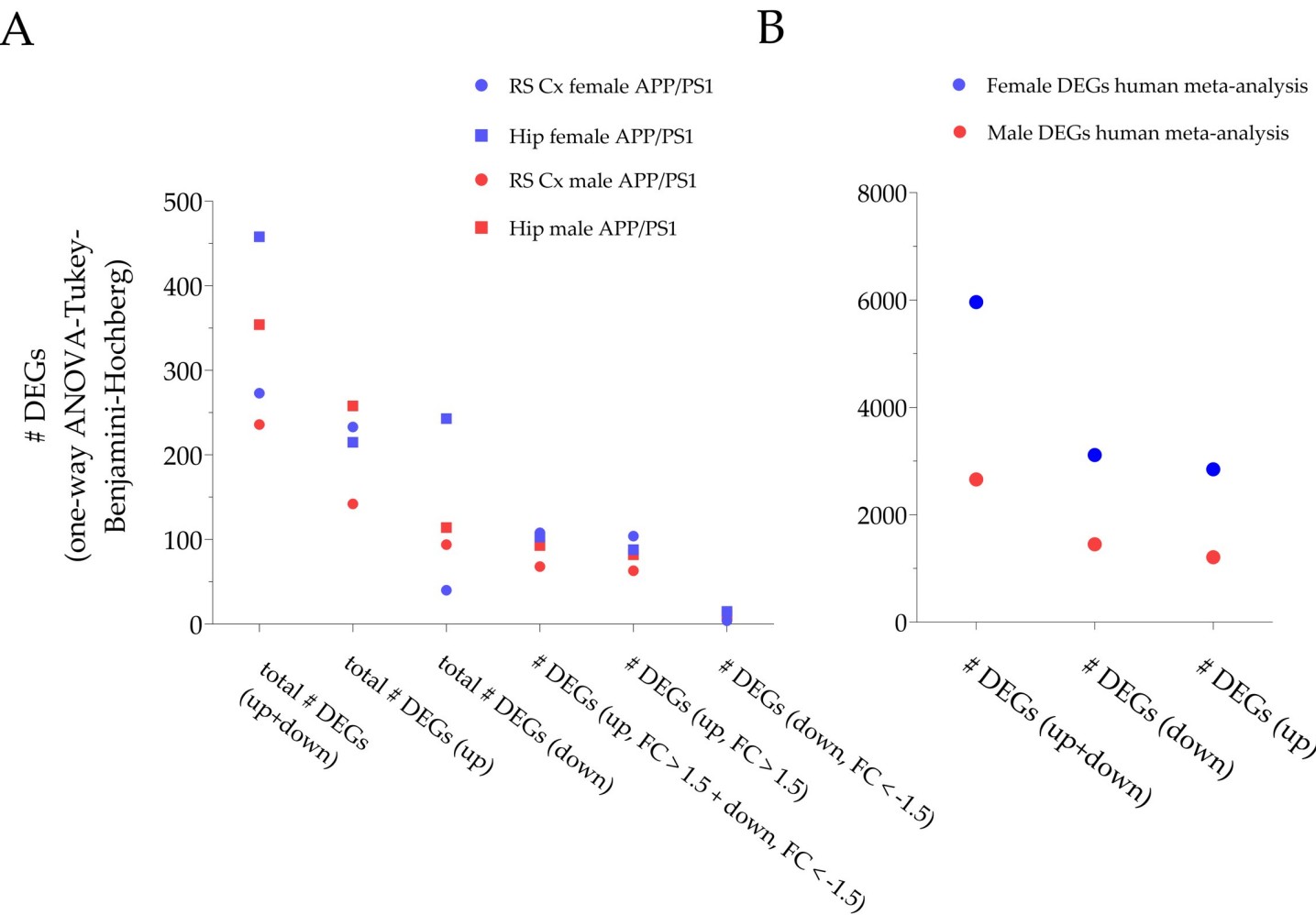

**Fig 1. Quantitative distribution of DEGs in sex- and brain-region specific subgroups of APP/PS1 mice.** (A) Illustration of the total number of DEGs in our four comparative subgroups, i.e., Rs Cx of female APP/PS1 mice, Hip of female APP/PS1 mice, RS Cx of male APP/PS1 animals and Hip of male APP/PS1 mice. All identified genes are based on a one-way ANOVA, Tukey and Benjamini-Hochberg procedure to identify significantly up- or downregulated genes. The total number of DEGs is higher in the hippocampus compared to the RS cortex. For the total number of upregulated genes, there seems to be no clear preference in BROI or sex. The same holds true for DEGs < -1.5 or > 1.5. Note that for most subgroups, the number of downregulated genes is much less compared to upregulated ones. (B) Illustration of the number of DEGs from a meta-analysis from 2114 *post mortem* AD patient samples performed by Wan et al. (2020). In contrast to our APP/PS1 study, the number of DEGs is persistently higher in *post mortem* probes from female AD subjects compared to probes from males. Overall, the numbers of DEGs are much higher compared to our APP/PS1 study. The latter might be due to the interindividual variability in the AD patients probes and the fact that Wan et al. (2020) included seven BROIs in their meta-analysis. For methodological details on their meta-analysis and statistics see [73].

The top 30 DEGs (FC > 1.5 and FC < -1.5, respectively) are displayed in **S5C Fig in S1 File**. The downregulated high FC candidate genes included: *6530402F18Rik*, *1700001L05Rik*, *Arpp21*, *Ctr9*, *Scyl2*. The following top 10 high FC candidate genes were upregulated in this subgroup: *Ccl6*, *Prnp*, *Cd52*, *Bcl2a1d*, *Lgals3bp*, *Tyrobp*, *Lyz1*, *C4b*, *Gpr84*, *Lyz2*.

**Hippocampus of male APP/PS1 mice.** The Feature Extraction and GeneSpring RT software (both Agilent Technologies Germany GmbH & Co. KG, Germany) was utilized to list DEGs based on one-way ANOVA and Tukey *post hoc* testing followed by the Benjamini-Hochberg procedure for control of FDR. As depicted in **S8 Table in S1 File**, a total number of 354 gene candidates including l(i)ncRNAs were found including eleven candidates with a FC < -1.5 and 64 candidates with a FC > 1.5. The majority (67.80%) of DEGs was upregulated in this subgroup (240↑/114↓) (**Figs 1A and 2D**).

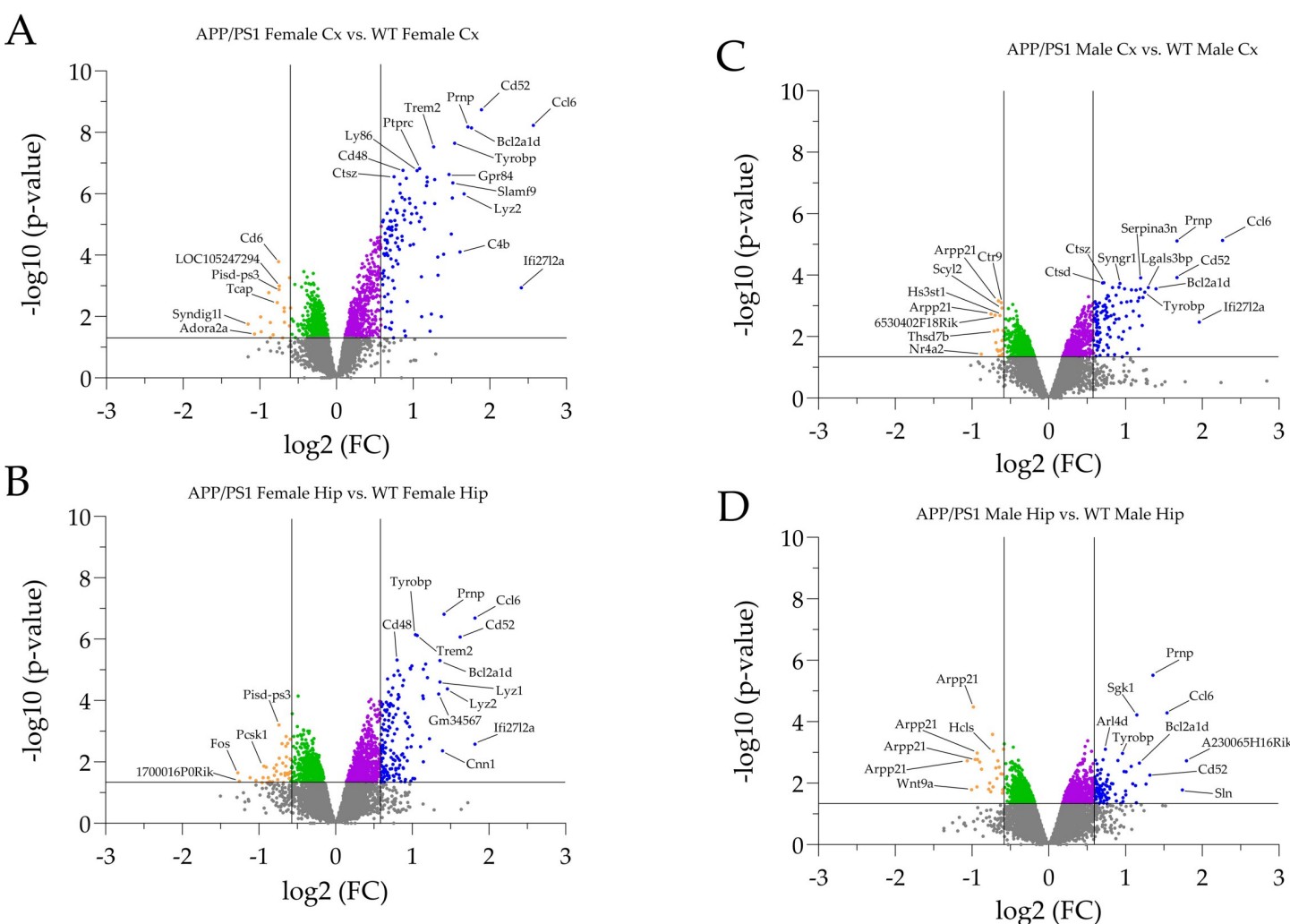

**Fig 2. Volcano plots illustrating statistical significance vs. magnitude of FC.** Profiles are depicted for the RS cortex of female APP/PS1 mice (A), the hippocampus of female APP/PS1 mice (B), the RS cortex of male APP/PS1 mice (C) and the hippocampus of male APP/PS1 animals (D). All genes exhibiting differential transcript levels were plotted with each dot representing one gene. The log2-FC in the individual APP/PS1 subgroups vs. WT is represented on the x-axis. The y-axis displays related log10 p-values (t-test procedure without further correction). The p-value of 0.05 and a FC of > 1.5 and < -1.5 are indicated by a horizontal line and vertical lines, respectively. Data points below the horizontal line represent gene candidates that were not significantly altered in transcription. Gene candidates in the different Volcano sectors (significantly up- or downregulated and FC > 1.5 or FC < -1.5) are color-coded. Note that downregulated gene candidates in the upper left sector (FC < -1.5, light orange) and upregulated genes in the upper right sector (FC > 1.5, blue) were further investigated in Venn and pathway studies. In most subgroups, the upregulated gene candidates predominate. Selected DEGs with high FC and statistical significance were individually labelled. Genes with significantly altered FC in the range from -1.5 to 1.5 (green and purple) were not further analyzed in our study.

The top 30 DEGs (FC > 1.5 and FC < -1.5, respectively) are displayed in **S5D Fig in S1 File**. The following high FC candidate genes were downregulated, i.e., *Arpp21*, *Wnt9a*, *Pla2g4e*, *4933407I18Rik*, *Hlcs*, *Shisa9*, *Hdac9*. The following top 10 high FC candidate genes were upregulated, i.e., *Ccl6*, *Prnp*, *Cd52*, *Phf11d*, *Bcl2a1d*, *Sgk1*, *Lyz1*, *Lyz2*, *Gfap*, *C4b*. Note that DEG numbers in **S5-S8 Tables in S1 File** can slightly differ from the number of high FC gene candidates in **S5 Fig in S1 File** due to, e.g., unspecified sequences or multiple transcript variants of individual genes.

For comparison with the aforementioned lists of DEGs in the individual APP/PS1 subgroups, **Fig 1B** illustrates the number of DEGs from a meta-analysis from 2114 *post mortem* AD patient samples performed by Wan et al. (2020) [73].

## GO and enrichment studies for upregulated genes (FC > 1.5) in the RS cortex and hippocampus of APP/PS1 AD mice compared to controls

**RS cortex of female APP/PS1 mice.** GO and enrichment analysis in the individual subgroups was performed using Metascape (metascape.org). Pathway and process enrichment analysis was carried out with the following ontology sources, i.e., GO Biological Processes, KEGG Pathway, Reactome Gene Sets, and WikiPathways. Importantly, all genes in the genome have been used as the enrichment background. Due to the number and proportion of the significantly up- and downregulated genes, the GO analysis and enrichment was performed for gene candidates with an absolute FC > 1.5. The top 20 enrichment clusters for upregulated DEGs in the RS cortex of female APP/PS1 mice compared to controls are depicted in **Fig 3A**. In addition, a detailed list of genes related to the top 10 identified clusters is given in **Table 1A**. The top three cluster terms/categories in this subgroup encompass (i) the microglia pathogen phagocytosis pathway (WP3626, Log(10)P: -23.06), (ii) the immune effector process (GO:0002252, Log(10)P: -18.88), and (iii) the innate immune response (GO:0045087, Log(10)P: -16.66). In addition, a PPI enrichment analysis was conducted using the following databases: STRING6, BioGrid7, OmniPath8, InWeb_IM9 (metascape org.). The top three PPI categories included (i) the microglia pathogen phagocytosis pathway (WP3626, Log(10)P: -25.5), (ii) the immune effector process (GO:0002252, Log(10)P: -17.3) and (iii) the neutrophil degranulation (R-MMU-6798695, Log(10)P: -15.4).

**Hippocampus of female APP/PS1 mice.** The top 20 enrichment clusters for upregulated DEGs (FC > 1.5) for the hippocampus of female APP/PS1 mice compared to WT controls are depicted in **Fig 3B**. In addition, a detailed list of genes related to the top 10 identified clusters is given in **Table 1B**. The top three cluster terms/categories in this subgroup encompass (i) the microglia pathogen phagocytosis pathway (WP3626, Log(10)P: -13.76), (ii) the innate immune response (GO:0045087, Log(10)P: -12.36), and (iii) the positive regulation of immune response (GO:0050778, Log(10)P: -11.08). In addition, a PPI enrichment analysis was performed using the following databases: STRING6, BioGrid7, OmniPath8, InWeb_IM9 (metascape org.). The top three PPI categories included: (i) the immune effector process (GO:0002252, Log(10)P: -14.8), (ii) the microglia pathogen phagocytosis pathway (WP3626, Log(10)P: -13.8) and (iii) the positive regulation of immune response (GO:0050778, Log(10)P: -13.1).

**RS cortex of male APP/PS1 mice.** GO analysis and enrichment was performed for gene candidates with a FC >1.5. The top 20 enrichment clusters for upregulated genes for the RS cortex of male APP/PS1 mice compared to WT controls are depicted in **Fig 3C**. In addition, a detailed list of genes related to the top 10 identified clusters is provided in **Table 1C**. The top three cluster terms/categories in this subgroup comprised (i) the microglia pathogen phagocytosis pathway (WP3626, Log(10)P: -19.26), (ii) the immune effector process (GO:0002252, Log(10)P: -12.84), and (iii) the neutrophil degranulation (R-MMU-6798695, Log(10)P: -12.50). The top three PPI enrichment categories included (i) the microglia pathogen phagocytosis pathway (WP3626, Log(10)P: -22.1), (ii) the positive regulation of immune response (GO:0050778, Log (10)P: -16.8) and (iii) the immune effector process (GO:0002252, Log(10)P: -13.7).

**Hippocampus of male APP/PS1 mice.** The top 20 enrichment clusters for upregulated (FC > 1.5) DEGs of the hippocampus of male APP/PS1 mice compared to WT controls are depicted in **Fig 3D**. In addition, a detailed list of genes related to the top 10 identified clusters is given in **Table 1D**. The top three cluster terms/categories in this subgroup comprised (i) the macrophage markers (WP2271, Log(10)P: -13.32), (ii) the immune effector process (GO:0002252, Log(10)P: -13.10), and (iii) the microglia pathogen phagocytosis pathway (WP3626, Log(10)P: -12.98). The top three PPI categories included (i) the immune effector

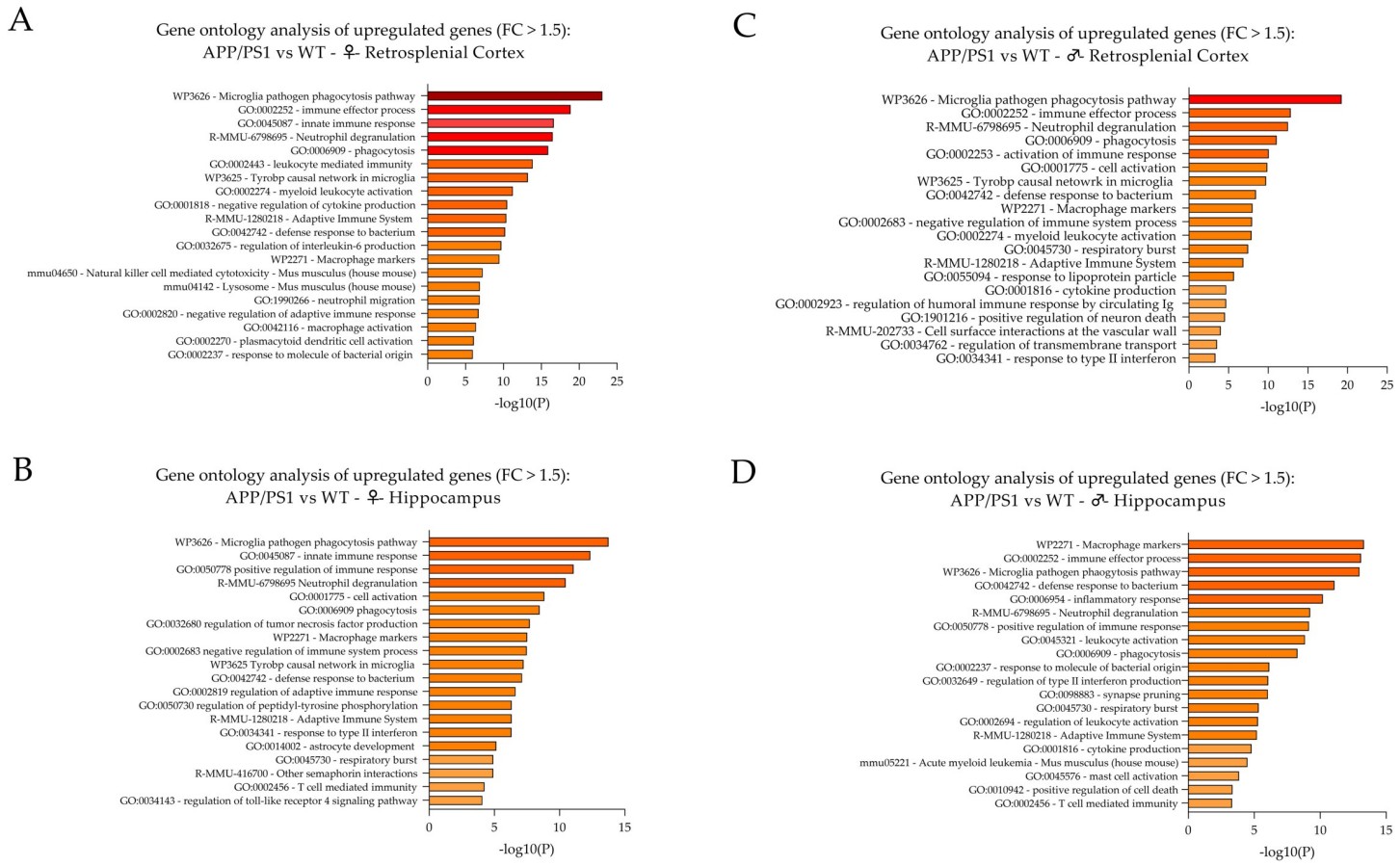

**Fig 3. GO analysis of DEGs (FC > 1.5, p < 0.05).** Bar graphs of enriched terms across the input gene list and related p-values were obtained from the Metascape gene list analysis report for (A) the RS cortex of female APP/PS1 AD vs. WT mice, (B) the hippocampus of female APP/PS1 AD vs. WT mice, (C) the RS cortex of male APP/PS1 AD vs. WT mice and (D) the hippocampus of male APP/PS1 AD vs. WT animals. The top 20 categories/clusters are itemized together with their related -Log(10)P values in a descending fashion.

process (GO:0002252, Log(10)P: -14.8), (ii) the microglia pathogen phagocytosis pathway (WP3626, Log(10)P: -14.6) and (iii) the leukocyte mediated immunity (GO:0002443, Log(10)P: -12.7).

**Comprehensive enrichment characteristics of upregulated DEGs of the RS cortex and hippocampus in male and female APP/PS1 mice compared to WT controls.** Obviously, some cluster terms turned out to be most relevant in all four subgroups, i.e., the microglia pathogen phagocytosis pathway (WP3626), neutrophil degranulation (R-MMU-6798695) and phagocytosis (GO:0006909). Other common categories include the immune effector process (GO:0002252) and the Tyrobp causal network in microglia (WP3625) in three subgroups and the leukocyte activation (GO:0045321), macrophage markers (WP2271) and negative regulation of immune system process (GO:0002683) in two subgroups. Individual, subgroup-specific categories comprised, e.g., the leukocyte mediated immunity (GO:0002443) in the RS cortex of female APP/PS1 mice and the regulation of tumor necrosis factor production (GO:0032680) in the hippocampus of female APP/PS1 mice or sex-specific categories such as the innate immune response (GO:0045087) in the cortex and hippocampal of female APP/PS1 probes.

Overall, the majority of DEGs is related to the activation of microglial cells, leucocytes and macrophages/the immune system which is in accordance with previous studies [74]. However,

**Table 1. Enrichment clusters and related upregulated DEGs.** Enriched upregulated gene candidates (FC > 1.5, indicated via gene symbols) are depicted for the top 10 related categories/cluster terms derived from GO analysis (see also Fig 2) for (A) the RS cortex of female APP/PS1 AD mice, (B) the hippocampus of female APP/PS1 AD mice, (C) the RS cortex of male APP/PS1 AD mice, and (D) the hippocampus of male APP/PS1 AD mice. In addition, the Log(10)P values for the individual enriched categories/cluster terms are provided.

| Description/Category/ Cluster term | Log(10) P | Gene symbols |
|---|---|---|
| **A)** | | |
| Microglia pathogen phagocytosis pathway (WP3626) | -23,06 | C1qa,C1qb,C1qc,Cyba,Fcer1g,Fcgr1,Ptpn6,Itgb2,Ncf2,Ncf4,Rac2, Tyrobp,Vav1,Trem2,Slc11a1,Cd14,Ctss,Fcgr2b,Fcgr3, Rab7b, Hvcn1,Psmb9,Psmb8,Tlr7,Prnp,Pycard,Arhgap9,Uba7,Ctsd,Igf1 |
| Immune effector process (GO:0002252 | -18,88 | Adora3,Bcl3,C1qa,C1qb,C1qc,C4b,Csf2rb,Fcer1g,Fcgr1,Fcgr2b, Fcgr3,Ptpn6,Irf8,Lag3,Myo1f,Slc11a1,Lcp1,Tyrobp,Apbb1ip, Pycard,Trem2,Slamf9,Havcr2,Aif1,Bcl2a1d,Cd48,Igf1,Itgb2,Ptprc, Vav1,Plek,Ifi27l2a |
| Innate immune response (GO:0045087) | -16,66 | C1qa,C1qb,C1qc,Capg,Cd14,Cyba,Fcer1g,Fcgr1,Ptpn6,Irf8,Lag3, Ly86,Mpeg1,Naip2,Slc11a1,Ccl6,Ccl9,Irf5,Pycard,Trem2, Sp110, Tlr7,Havcr2,Rab7b,Oas1a,Tlr12,Adora3,Bcl2a1d,C4b,Fcgr3,Itgb2, Ptprc,Tyrobp,Vav1,Aif1,Cd68,Fcgr2b,Il4i1,Laptm5, Rac2,Bcl3, Cd300c2,Igf1,Ctss,Myo1f |
| Neutrophil degranulation (R-MMU-6798695) | -16,51 | Cd14,Cd68,Ctsd,Ctss,Cyba,Fcer1g,Fcgr2b,Ptpn6,Hexb,Itgb2,Lyz2, Slc11a1,Ptprc,Syngr1,Tyrobp,Lair1,Ctsz,Pycard,Hvcn1,Gpr84, Gusb,Hk3,Arhgap9 |
| Phagocytosis (GO:0006909) | -15,90 | Aif1,Fcer1g,Fcgr1,Fcgr2b,Fcgr3,Irf8,Itgb2,Ncf2,Ncf4,Slc11a1, Tyrobp,Vav1,Trem2,Rab7b,P2ry6,Adora3,Il4i1,Ptpn6,Lag3,Ptprc, Rac2,Havcr2,C4b,Cyba,Pycard,Cd48,Samsn1,Ctss,Psmb9,Psmb8, Cd14,Ctsd,Lgals3bp,Siglech,Lyz2,Lyz1,Igf1,Laptm5,Syngr1,Cd68, Tbxas1,Apbb1ip |
| Leukocyte mediated immunity (GO:0002443) | -13,87 | Adora3,Bcl3,C1qa,C1qb,C1qc,C4b,Csf2rb,Fcer1g,Fcgr1,Fcgr2b, Fcgr3,Ptpn6,Lag3,Myo1f,Slc11a1,Slamf9,Cd14,Irf8,Itgb2,Pycard, Havcr2,Trem2,Tlr7,Oas1a,Ctss,Lcp1 |
| Tyrobp causal network in microglia (WP3625) | -13,22 | C1qc,Capg,Hcls1,Itgb2,Ncf2,Tyrobp,Apbb1ip,Plek,Pycard,Samsn1, Aif1,Myo1f,Lcp1,Rac2 |
| Myeloid leukocyte activation (GO:0002274) | -11,25 | Adora3,Aif1,C1qa,Cd48,Fcer1g,Fcgr3,Myo1f,Slc11a1,Tyrobp, Pycard,Trem2,Havcr2,Bcl3,Irf8,Lcp1,Apbb1ip,Fcgr2b,Rac2, Prnp, Igf1,Ptprc,P2ry6 |
| Negative regulation of cytokine production (GO:0001818) | -10,50 | Bcl3,Fcgr2b,Ptpn6,Igf1,Lag3,Laptm5,Slc11a1,Prnp,Ptprc,Tyrobp, Pycard,Trem2,Havcr2,Oas1a,C1qc,Cd68,Fcer1g,Il4i1,Samsn1, Rab7b,Adora3,Aif1,Itgb2,Rac2,Plek,Csf2rb,Ccl6,Ccl9,P2ry6, Naip2,Hcls1,Myo1f,Vav1,Apbb1ip,Mlxipl |
| Adaptive immune system (R-MMU-1280218) | -10,39 | Ctsd,Ctss,Cyba,Fcgr1,Fcgr2b,Ptpn6,Itgb2,Lag3,Psmb9,Psmb8, Ncf2,Ncf4, Ptprc,Vav1,Hcst,Lair1,Uba7,Trem2,Cd300c2 |
| **B)** | | |
| Microglia pathogen phagocytosis pathway (WP3626) | -13,76 | C1qa,C1qb,C1qc,Cyba,Fcer1g,Ptpn6,Itgb2,Tyrobp,Trem2,Bcl3, C4b,Ctsh,Icam1,Irf8,Myo1f,Slc11a1,Lcp1, Slamf9,Myo1g,Cd14, Prnp,Tubb6,Tlr7 |
| Innate immune response (GO:0045087) | -12,36 | C1qa,C1qb,C1qc,Capg,Cd14,Cyba,Fcer1g,Ptpn6,Irf8,Ly86,Mpeg1, Naip2,Slc11a1,Ccl6,Vim,Ifitm3,Trem2, Sp110,Tlr7,Rab7b,Cd68, Icam1,Itgb2,Tyrobp |
| Positive regulation of immune response (GO:0050778) | -11,08 | Bcl2a1d,C1qa,C1qb,C1qc,C4b,Cd14,Cyba,Fcer1g,Ptpn6,Itgb2, Slc11a1,Pla2g4a,Ptprc,Tyrobp,Fyb,Trem2,Tlr7, Myo1g,Ctsh, Psmb8 |
| Neutrophil degranulation (R-MMU-6798695) | -10,47 | Cd14,Cd68,Ctsh,Cyba,Fcer1g,Ptpn6,Itgb2,Lyz2,Slc11a1,Ptprc, Syngr1,Tyrobp,Ctsz,Hvcn1,Gpr84,Arhgap9,Icam1 |
| Cell activation (GO:0001775) | -8,83 | Bcl2a1d,Bcl3,C1qa,Cd48,Fcer1g,Ptpn6,Icam1,Irf8,Igf1,Itgb2, Myo1f,Slc11a1,Lcp1,Ptprc,Tyrobp,Trem2,Slamf9, Cd14,Laptm5, Prnp,Tlr7,Psmb8,Rab32 |

(*Continued*)

**Table 1.** (*Continued*)

| Description/Category/ Cluster term | Log(10) P | Gene symbols |
|---|---|---|
| Phagocytosis (GO:0006909) | -8,47 | *Fcer1g,Irf8,Itgb2,Slc11a1,Tyrobp,Trem2,Rab7b,P2ry6,Myo1g, Cd48,Ptpn6,Icam1,Hcst,C1qa,Myo1f,C4b,Laptm5, Ptprc,Tlr7, Il4i1,Igf1,Pla2g4a,Prnp,Samsn1,Cd68,Cd14,Ccl6,Slamf9,Bcl3,Fyb, Cyba* |
| Regulation of tumor necrosis factor production (GO:0032680) | -7,72 | *Bcl3,Cd14,Cyba,Fcer1g,Ptpn6,Igf1,Ptprc,Tyrobp,Trem2,Laptm5, Tlr7,Rab7b,Irf8,Slc11a1,Il4i1,Ly86,Myo1f,Pla2g4a* |
| Macrophage markers (WP2271) | -7,52 | *Cd14,Cd68,Lyz2,Cd52* |
| Negative regulation of immune system process (GO:0002683) | -7,48 | *C1qc,Cd68,Fcer1g,Il4i1,Ptpn6,Igf1,Laptm5,Prnp,Ptprc,Tyrobp, Samsn1,Trem2,Rab7b,Bcl3,Slc11a1,Icam1,Itgb2,Myo1f, Ifitm3* |
| Tyrobp causal network in microglia (WP3625) | -7,24 | *C1qc,Capg,Hcls1,Itgb2,Tyrobp,Samsn1,Laptm5* |
| **C)** | | |
| Microglia pathogen phagocytosis pathway (WP3626) | -19,26 | *C1qa,C1qb,C1qc,Cyba,Fcer1g,Fcgr1,Ptpn6,Itgb2,Ncf2,Tyrobp, Trem2,Bcl2a1d,C4b,Cd14,Fcgr3,Lag3,Slc11a1,Ptprc, Fyb,Tlr7, Capg,Irf8,Ly86,Mpeg1,Naip2,Ccl6,Cd68,Igf1,Cd300c2,Ctss,Myo1f, Il4i1* |
| Immune effector process (GO:0002252) | -12,84 | *C1qa,C1qb,C1qc,C4b,Fcer1g,Fcgr1,Fcgr3,Ptpn6,Irf8,Lag3,Myo1f, Slc11a1,Tyrobp,Trem2,Slamf9,Ctss* |
| Neutrophil degranulation (R-MMU-6798695) | -12,50 | *Cd14,Cd68,Ctsd,Ctss,Cyba,Fcer1g,Ptpn6,Itgb2,Lyz2,Slc11a1, Ptprc,Syngr1,Tyrobp,Ctsz,Hvcn1,Gpr84* |
| Phagocytosis (GO:0006909) | -11,07 | *Fcer1g,Fcgr1,Fcgr3,Irf8,Itgb2,Ncf2,Slc11a1,Tyrobp,Trem2,P2ry6, C4b,Cyba,Ptprc,Ptpn6,Lag3,Tlr7,Cd48,Il4i1, Samsn1,Lyz2,Lyz1, Ctss,Myo1f,Ccl6,Cd14,Lgals3bp,Siglech,Slamf9,Cd68,Igf1,Syngr1* |
| Activation of immune response (GO:0002253) | -10,06 | *Bcl2a1d,C1qa,C1qb,C1qc,C4b,Cd14,Fcer1g,Ptpn6,Ptprc,Tyrobp, Fyb,Tlr7* |
| Cell activation (GO:0001775) | -9,87 | *Bcl2a1d,C1qa,Cd48,Fcer1g,Fcgr3,Ptpn6,Irf8,Igf1,Itgb2,Lag3, Myo1f,Slc11a1,Ptprc,Tyrobp,Trem2,Slamf9* |
| Tyrobp causal network in microglia (WP3625) | -9,73 | *C1qc,Capg,Hcls1,Itgb2,Ncf2,Tyrobp,Samsn1* |
| Defense response to bacterium (GO:0042742) | -8,45 | *Fcer1g,Fcgr1,Irf8,Lyz2,Lyz1,Mpeg1,Myo1f,Naip2,Slc11a1,Trem2, Slamf9* |
| Macrophage markers (WP2271) | -8,01 | *Cd14,Cd68,Lyz2,Cd52,Irf8,Slc11a1,Prnp,Tlr7,Trem2,Ctss,Itgb2, Ly86,Ncf2,Igf1,P2ry6,Ptprc,Cyba,Tyrobp,Fcgr1* |
| Negative regulation of immune system process (GO:0002683) | -7,97 | *C1qc,Cd68,Fcer1g,Il4i1,Ptpn6,Igf1,Lag3,Prnp,Ptprc,Tyrobp, Samsn1,Trem2,Hcls1,Itgb2,Slc11a1,Myo1f,Bcl2a1d,Naip2* |
| **D)** | | |
| Macrophage markers (WP2271) | -13,32 | *Cd14,Cd68,Cd86,Lyz2,Rac2,Cd52* |
| Immune effector process (GO:0002252) | -13,10 | *B2m,Bcl3,C1qa,C1qc,C4b,Ctsh,Fcgr1,Fcgr2b,Fcgr3,Irf8,Myo1f, Ncf1,Slc11a1,Tyrobp,Trem2,Cd86,H2bc6,Cd14, Irf9,Tlr7* |
| Microglia pathogen phagocytosis pathway (WP3626) | -12,98 | *C1qa,C1qc,Fcgr1,Hck,Ncf1,Rac2,Tyrobp,Trem2,Slc11a1,Hvcn1, Prnp,Ccl6,Rhoh* |
| Defense response to bacterium (GO:0042742) | -11,08 | *B2m,Bcl3,Fcgr1,Hck,Irf8,Lyz2,Lyz1,Mpeg1,Myo1f,Naip2,Ncf1, Slc11a1,Trem2* |
| Inflammatory response (GO:0006954) | -10,20 | *C1qa,Cd14,Cd68,Fcgr1,Fcgr3,Hck,Ly86,Naip2,Ncf1,Slc11a1,Ccl6, Tyrobp,Trem2,Tlr7,B2m,C1qc,Irf8,Mpeg1,Clec5a* |
| Neutrophil degranulation (R-MMU-6798695) | -9,24 | *B2m,Cd14,Cd68,Ctsh,Ctss,Fcgr2b,Lyz2,Slc11a1,Syngr1,Tyrobp, Clec5a,Hvcn1,Gpr84* |
| Positive regulation of immune response (GO:0050778) | -9,15 | *B2m,Bcl2a1d,C1qa,C1qc,C4b,Cd14,Cd86,Fcgr1,Fcgr3,Slc11a1, Tyrobp,Fyb,Trem2,Tlr7,Bcl3,Irf8,Clec5a,Cd300c2,Ctsh, Ly86, Myo1f,Ncf1* |
| Leukocyte activation (GO:0045321) | -8,85 | *B2m,Bcl2a1d,Bcl3,C1qa,Cd48,Cd86,Fcgr2b,Fcgr3,Irf8,Myo1f, Slc11a1,Tyrobp,Rhoh,Trem2,Cebpa* |

(*Continued*)

**Table 1.** (Continued)

| Description/Category/ Cluster term | Log(10) P | Gene symbols |
|---|---|---|
| Phagocytosis (GO:0006909) | -8,28 | *Fcgr1,Fcgr2b,Fcgr3,Hck,Irf8,Slc11a1,Tyrobp,Trem2,B2m,Ctss, Ctsh,Lyz2,Lyz1,Ncf1,C4b,Cd86,Rac2,Tlr7,Irf9,Cd14, Cebpa,Ly86, Cd48,Samsn1,Myo1f,H2bc6,Cd68* |
| Response to molecule of bacterial origin (GO:0002237) | -6,14 | *B2m,Cd14,Cd68,Cd86,Fcgr2b,Irf8,Ly86,Slc11a1,Trem2,Sgk1,Ctss, Tlr7* |

there are additional, distinct sex- and brain-region related specificities in transcriptional profiles which are elaborated in detail below.

**GO and enrichment studies for downregulated DEGs (FC < -1.2) in the RS cortex and hippocampus of APP/PS1 AD mice compared to controls.** As outlines above, the overall number of downregulated DEGs with a FC < -1.5 was severely lower compared to the upregulated DEGs (FC > 1.5). To carry out GO and enrichment studies with a sufficient number of downregulated gene candidates, the cutoff was set to a FC < -1.2.

**RS cortex of female APP/PS1 mice.** Analysis of downregulated DEGs of the RS cortex of female APP/PS1 vs. WT mice revealed no GO categories and significant enrichment in this subgroup. In addition, PPI enrichment analysis also revealed no significant categories (**Fig 4A** and **Table 2A**).

**Hippocampus of female APP/PS1 mice.** The seven enrichment clusters for downregulated DEGs of the hippocampus of female APP/PS1 mice compared to WT controls are depicted in **Fig 4B**. In addition, a detailed list of DEGs related to the identified clusters is given in **Table 2B**. The top three cluster terms/categories in this subgroup encompass (i) the chemical synaptic transmission (GO:0007268, Log(10)P: -5.16), (ii) the peptide hormone processing (GO:0016486, Log(10)P: -3.59), and (iii) the homophilic cell adhesion via plasma membrane adhesion molecules (GO:0007156, Log(10)P: -2.83). PPI enrichment analysis revealed three top categories, i.e., (i) the synaptic vesicle cycle (GO:0099504, Log(10)P: -6.5), (ii) the vesicle-mediated transport in synapse (GO:0099003, Log(10)P: -6.2) and (iii) the regulation of synaptic vesicle fusion to presynaptic active zone membrane (GO:0031630, Log(10)P: -5.3).

**RS cortex of male APP/PS1 mice.** The six enrichment clusters for downregulated DEGs of the RS cortex of male APP/PS1 mice compared to WT controls are depicted in **Fig 4C**. In addition, a detailed list of DEGs related to the identified clusters is provided in **Table 2C**. The top three cluster terms/categories in this subgroup comprise (i) rRNA processing (GO:0006364, Log(10)P: -3.02), (ii) antigen processing: ubiquitination & proteasome degradation (R-MMU-983168, Log(10)P: -2.48), and (iii) mechanisms associated with pluripotency (WP1763, Log(10)P: -2.45). PPI enrichment analysis revealed three top categories, i.e., (i) DNA metabolic process (GO:0006259, Log(10)P: -3.5), (ii) cellular senescence (mmu04218, Log(10)P: -3.3) and (iii) regulation of heart contraction (GO:0008016, Log(10)P: -3.2).

**Hippocampus of male APP/PS1 mice.** The three enrichment clusters for downregulated gene candidates of the hippocampus of male APP/PS1 mice compared to WT animals are depicted in **Fig 4D**. In addition, a detailed list of genes related to the identified clusters is shown in **Table 2D**. The three cluster terms/categories in this subgroup encompass (i) the centrosome cycle (GO:0007098, Log(10)P: -4.20), (ii) the histone monoubiquitination (GO:0010390, Log(10)P: -4.19), and (iii) endocytosis (GO:0006897, Log(10)P: -2.54). PPI enrichment analysis revealed only one category, i.e., histone modification (GO:0016570, Log(10)P: -2.7).

A    Gene ontology analysis of downregulated genes (FC < -1.2):
APP/PS1 vs WT - ♀ Retrosplenial Cortex

No enrichment terms

B    Gene ontology analysis of downregulated genes (FC < -1.2):
APP/PS1 vs WT - ♀ Hippocampus

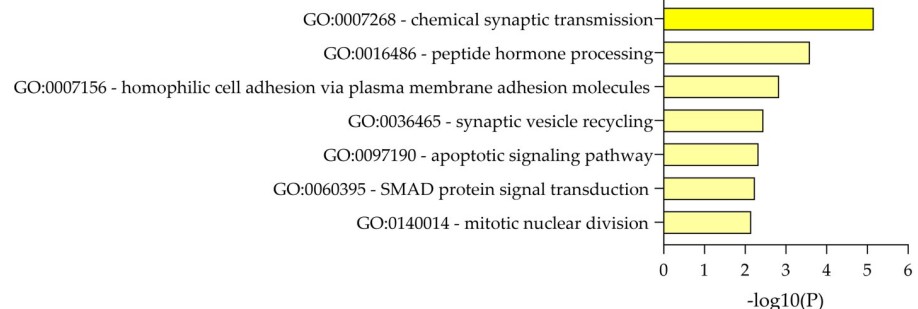

C    Gene ontology analysis of downregulated genes (FC < -1.2):
APP/PS1 vs WT - ♂ Retrosplenial Cortex

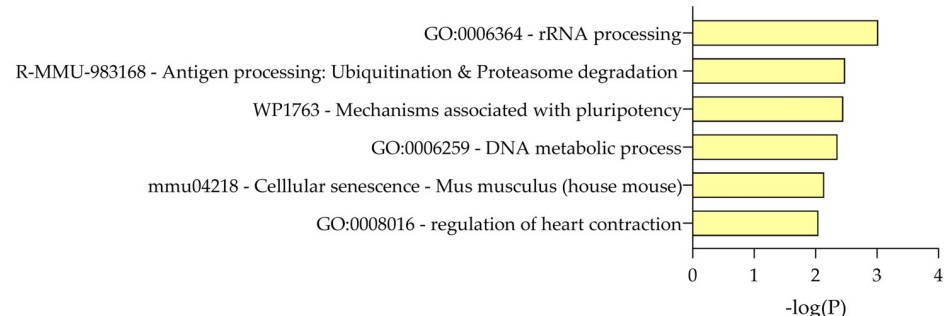

D    Gene ontology analysis of downregulated genes (FC < -1.2):
APP/PS1 vs WT - ♂ Hippocampus

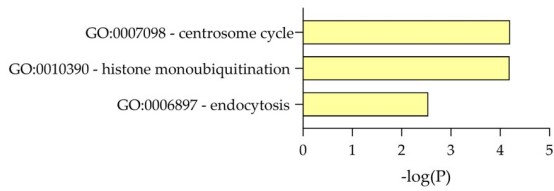

**Fig 4. GO analysis of DEGs (FC < -1.2, p < 0.05).** Bar graphs of enriched terms across the input gene list were obtained from the Metascape gene list analysis report for (A) the RS cortex of female APP/PS1 AD vs. WT mice, (B) the hippocampus of female APP/PS1 AD vs. WT mice, (C) the RS cortex of male APP/PS1 AD vs. WT mice and (D) the hippocampus of male APP/PS1 AD vs. WT mice. The individual categories are provided together with their related -Log(10)P values in a descending fashion. Note that no GO analysis results could be obtained in the RS cortex of female APP/PS1 AD mice vs. WT controls. Overall, GO analysis for downregulated gene candidates revealed a limited number of categories compared to upregulated gene candidates (see also Fig 2).

**Table 2. Enrichment clusters and related downregulated DEGs.** Enriched downregulated gene candidates (FC < -1.2, indicated via gene symbols) are depicted for the related categories/cluster terms derived from GO analysis (see also Fig 3) for (A) the RS cortex of female APP/PS1 AD mice, (B) the hippocampus of female APP/PS1 AD mice, (C) the RS cortex of male APP/PS1 AD mice, and (D) the hippocampus of male APP/PS1 AD mice. Note that GO analysis and enrichment did not reveal significant results for the RS cortex of female APP/PS1 AD animals. The Log(10)P values for the individual enriched categories/cluster terms are also provided. Importantly, the number of enriched clusters and related gene candidates is considerably reduced for downregulated compared to upregulated gene candidates (see also Table 1).

| Description/Category/Cluster term | Log (10)P | Gene symbols |
|---|---|---|
| **A)** | | |
| --- | ---- | --- |
| **B)** | | |
| Chemical synaptic transmission (GO:0007268) | -5,16 | Cplx1,Rab3a,Vamp1,Xbp1,Mpp2,Clstn1,Sncb,Clstn3, Rims3,Begain,Dnajc6,Syngr1,Rab3d,Pex26 |
| Peptide hormone processing (GO:0016486) | -3,59 | Pcsk1,Scg5,Pcsk1n,Cmklr2 |
| Homophilic cell adhesion via plasma membrane adhesion molecules (GO:0007156) | -2,83 | Ptprm,Nectin1,Clstn1,Clstn3,Elavl2,Dab2ip,Rims3 |
| Synaptic vesicle recycling (GO:0036465) | -2,45 | Rab3a,Dnajc6,Sncb |
| Apoptotic signaling pathway (GO:0097190) | -2,33 | Fgfr3,Inhbb,Xbp1,Dab2ip,Faim2,Gsk3a,Gpc5,Lypd6, Meg3,Abtb3,Frs3 |
| SMAD protein signal transduction (GO:0060395) | -2,24 | Inhbb,Meg3,Abtb3 |
| Mitotic nuclear division (GO:0140014) | -2,15 | Fgfr3,Zwint,Spag5,Phf13 |
| **C)** | | |
| rRNA processing (GO:0006364) | -3,02 | Gtf2h5,Exosc1,Nsun4,Mettl5,Ints9,Dnmt3b,Ctr9, Hnrnpu |
| Antigen processing: Ubiquitination & Proteasome degradation (R-MMU-983168) | -2,48 | Psmc2,Ube2e1,Fbxl20,Dcaf1 |
| Mechanisms associated with pluripotency (WP1763) | -2,45 | Dnmt3b,Rbbp4,Ctr9,Hnrnpu |
| DNA metabolic process (GO:0006259) | -2,36 | Dnmt3b,Rbbp4,Gtf2h5,Uvssa,Atr,Dcaf1 |
| Cellular senescence - Mus musculus (house mouse) (mmu04218) | -2,14 | Cacna1d,Rbbp4,Atr |
| Regulation of heart (GO:0008016) | -2,05 | Cacna1d,Cacna2d1,Sp4 |
| **D)** | | |
| Centrosome cycle (GO:0007098) | -4,20 | Brca2,Odf2,Pard6a,Ccdc57 |
| Histone mono-ubiquitination (GO:0010390) | -4,19 | Brca2,Ctr9,Ube2e1,Matr3,Hdac9 |
| Endocytosis (GO:0006897) | -2,54 | Ldlr,Pikfyve,Syt5,Fcho1,Atp6v1h |

**Comprehensive enrichment characteristics for downregulated genes of the RS cortex and hippocampus in male and female APP/PS1 mice.** Overall, the number of significantly downregulated DEGs was lower compared to the upregulated genes. Similarly, the number of significantly enriched categories was also reduced for downregulated gene candidates compared to the upregulated DEGs.

## Venn analysis of upregulated DEGs (FC > 1.5) of the individual subgroups

**Cross-regional and region-specific upregulation of DEGs in the RS cortex and hippocampus of female APP/PS1 mice vs. controls.** We first investigated DEGs (FC > 1.5) in both the RS cortex and hippocampus of female APP/PS1 vs. WT mice (**Fig 5A**). In total, 22 candidates were upregulated solely in the hippocampus, 36 genes were upregulated only in the RS cortex, but 55 candidates exhibited an increased, intersectional transcription profile in both RS cortex and hippocampus. The individual gene candidates are also listed in **S9A Table in**

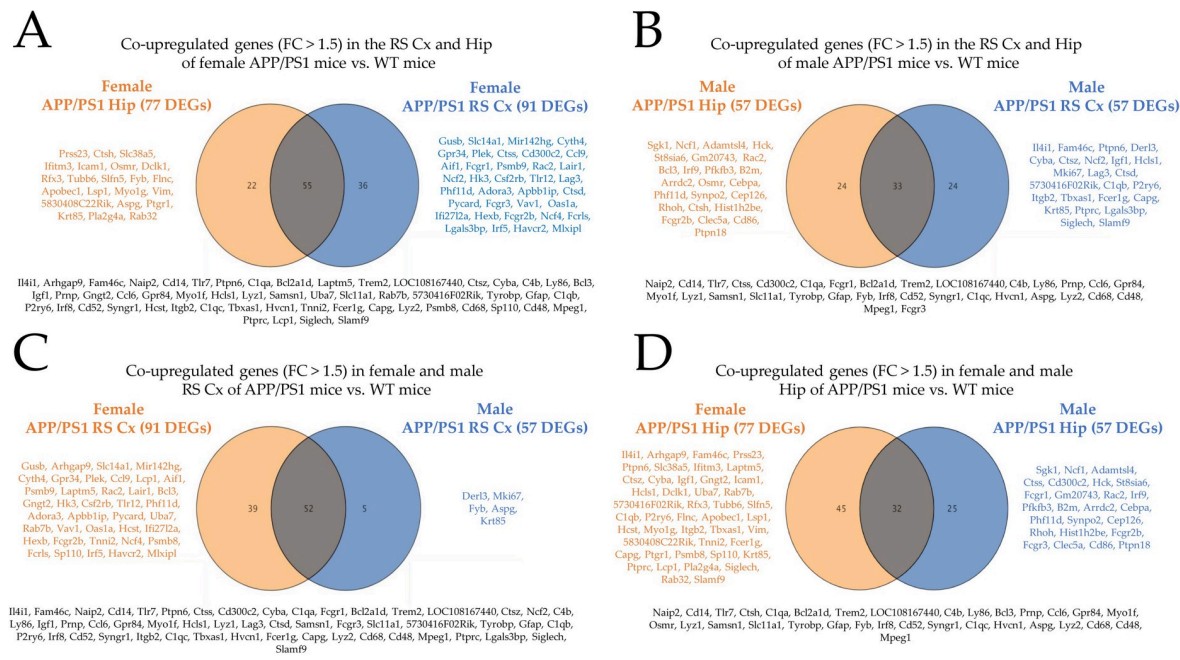

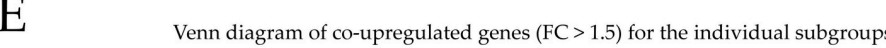

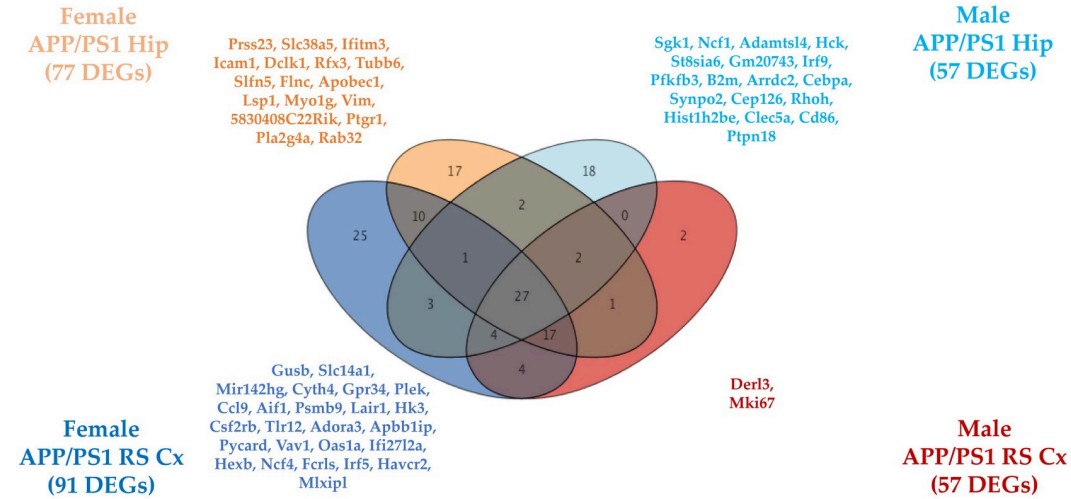

**Fig 5. Venn diagrams of upregulated DEGs (FC > 1.5) in APP/PS1 AD vs. WT control mice related to either both brain regions (RS cortex or hippocampus) or both sexes.** Co-upregulated (intersectional) DEGs in (A) the RS cortex and hippocampus of female APP/PS1 AD mice, (B) the RS cortex and hippocampus of male APP/PS1 AD mice, (C) the RS cortex of female and male APP/PS1 AD mice and (D) the hippocampus of female and male APP/PS1 AD mice. Intersectional gene symbols (in black) are listed below each diagram, the other gene symbols are depicted next to the diagram (in orange and blue). E) Venn diagram including all four subgroups. Note that there are specific DEGs (signature genes, listed here) that are selectively upregulated only in one of the four study groups and thus represent a unique profile for sex and BROI.

**S1 File**. The cross-area GO and enrichment profiling revealed the following top three categories for the 55 upregulated, intersectional gene candidates: (i) the microglia pathogen phagocytosis pathway (WP3626, Log(10)P: -15.07), (ii) the innate immune response (GO:0045087, Log(10)P: -12.94), and (iii) the immune effector process (GO:0002252, Log(10)P: -11.81). Notably, these co-upregulated candidates represent the majority of upregulated genes in both subgroups. Although it is interesting to point out gene candidates that are co-upregulated in both the RS cortex and hippocampus of female APP/PS1 mice, it is also important to consider the region-specific candidates that are exclusively upregulated in either the RS cortex or the hippocampus of female AD mice.

In the RS cortex, 36 candidates exhibited selectively increased transcript levels including *Gusb*, *Slc14a1*, *Mir142hg*, *Cyth4*, *Gpr34*, *Plek*, *Ctss*, *Cd300c2*, *Ccl9*, *Aif1*, *Fcgr1*, *Psmb9*, *Rac2*, *Lair1*, *Ncf2*, *Hk3*, *Csf2rb*, *Tlr12*, *Lag3*, *Phf11d*, *Adora3*, *Apbb1ip*, *Ctsd*, *Pycard*, *Fcgr3*, *Vav1*, *Oas1a*, *Ifi27l2a*, *Hexb*, *Fcgr2b*, *Ncf4*, *Fcrls*, *Lgals3bp*, *Irf5*, *Havcr2*, *Mlxipl* (process enrichment analysis: (i) positive regulation of response to external stimulus (GO:0032103, Log(10)P: -10.10); (ii) phagocytosis (GO:0006909, Log(10)P: -8.17); (iii) adaptive immune system (R-MMU-1280218, Log(10)P: -8.08); (iv) inflammatory response (GO:0006954, Log(10)P: -6.61); (v) neutrophil degranulation (R-MMU-6798695, Log(10)P: -5.62); (vi) Tyrobp causal network in microglia (WP3625, Log(10)P: -5.53). In the hippocampus, 22 candidates exclusively displayed elevated transcript levels, including *Prss23*, *Ctsh*, *Slc38a5*, *Ifitm3*, *Icam1*, *Osmr*, *Dclk1*, *Rfx3*, *Tubb6*, *Slfn5*, *Fyb*, *Flnc*, *Apobec1*, *Lsp1*, *Myo1g*, *Vim*, *Aspg*, *Ptgr1*, *Krt85*, *Pla2g4a*, *Rab32* (process enrichment analysis: (i) T cell mediated immunity (O:0002456, Log(10)P: -4.62); (ii) adaptive immune system (R-MMU-1280218, Log(10)P: -3.49); (iii) epithelial cell development (GO:0002064, Log(10)P: -2.94); (iv) supramolecular fiber organization (GO:0097435, Log(10)P: -2.76).

**Cross-regional and region-specific upregulation of DEGs in the RS cortex and hippocampus of male APP/PS1 mice vs. controls.** Next, we investigated DEGs (FC > 1.5) in the RS cortex and hippocampus of male APP/PS1 vs. WT mice (**Fig 5B**). In total, 24 candidates were upregulated solely in the hippocampus, 24 DEGs were upregulated only in the RS cortex, but 33 candidates displayed increased transcript levels in both RS cortex and hippocampus. The individual gene candidates are also listed in **S10A Table in S1 File**. The cross-area GO and enrichment profiling revealed the following top three categories for the upregulated, intersectional gene candidate transcripts: (i) inflammatory response (GO:0006954, Log(10)P: -10.95), (ii) positive regulation of immune response (GO:0050778, Log(10)P: -10.02), and (iii) immune effector process (GO:0002252, Log(10)P: -9.49). We further investigated the 24 gene candidate transcripts that were exclusively upregulated in the RS cortex of male APP/PS1, i.e., *Il4i1*, *Fam46c*, *Ptpn6*, *Derl3*, *Cyba*, *Ctsz*, *Ncf2*, *Igf1*, *Hcls1*, *Mki67*, *Lag3*, *Ctsd*, *C1qb*, *P2ry6*, *Itgb2*, *Tbxas1*, *Fcer1g*, *Capg*, *Krt85*, *Ptprc*, *Lgals3bp*, *Siglech*, *Slamf*9 (process enrichment analysis: (i) microglia pathogen phagocytosis pathway (WP3626, Log(10)P: -11.41); (ii) Tyrobp causal network in microglia (WP3625, Log(10)P: -6.34); (iii) regulation of leukocyte mediated immunity (GO:0002703, Log(10)P: -6.07); (iv) regulation of tumor necrosis factor production (GO:0032680, Log(10)P: -5.81); (v) leukocyte mediated immunity (GO:0002443, Log(10)P: -5.15); (vi) regulation of sequestering of calcium ion (GO:0051282, Log(10)P: -4.93); (vii) phagocytosis (GO:0006909, Log(10)P: -4.66). Similarly, the transcripts of 24 gene candidates were specifically upregulated in the hippocampus of male APP/PS1 mice, i.e., *Sgk1*, *Ncf1*, *Adamtsl4*, *Hck*, *St8sia6*, *Gm20743*, *Rac2*, *Bcl3*, *Irf9*, *Pfkfb3*, *B2m*, *Arrdc2*, *Osmr*, *Cebpa*, *Phf11d*, *Synpo2*, *Cep126*, *Rhoh*, *Ctsh*, *Hist1h2be*, *Fcgr2b*, *Clec5a*, *Cd86*, *Ptpn18* (GO characteristics: (i) Fc gamma R-mediated phagocytosis (mmu04666, Log(10)P: -5.59); (ii) leukocyte mediated immunity (GO:0002443, Log(10)P: -5.25); (iii) leukocyte mediated cytotoxicity (GO:0001909, Log(10)P: -4.37).

Interestingly, the number of upregulated, intersectional DEGs in both RS cortex and hippocampus of female APP/PS1 mice vs. controls was much higher than the related number in male APP/PS1 mice compared to WT animals (55 vs. 33).

**Cross-sex and sex-specific DEGs in the RS cortex of APP/PS1 mice vs. controls.** A central aspect of our study was the characterization of sex-specific DEGs genes in the RS cortex and hippocampus of APP/PS1 AD vs. WT mice. We first investigated DEGs (FC > 1.5) in both female and male RS cortex of APP/PS1 vs. WT mice (**Fig 5C**). In total, 39 candidates were upregulated solely in the female RS cortex, five were upregulated only in the male RS cortex, but a majority of 52 DEGs exhibited increased transcript levels in the RS cortex of both male and female APP/PS1 AD mice. The individual gene candidates are also listed in **S11A Table in S1 File**. The cross-sex GO and enrichment profiling revealed the following top three categories for intersectional upregulated transcripts of gene candidates: (i) microglia pathogen phagocytosis pathway (WP3626, Log(10)P: -19.75), (ii) immune effector process (GO:0002252, Log(10)P: -13.51), and (iii) neutrophil degranulation (R-MMU-6798695, Log(10)P: -13.21). A sex-specific elaboration of upregulated transcripts in the RS cortex of APP/PS1 mice revealed only five candidates in males, i.e., *Derl3*, *Mki67*, *Fyb*, *Aspg*, *Krt85* (with no enrichment results), but 39 signature candidates in female APP/PS1 animals: *Gusb*, *Arhgap9*, *Slc14a1*, *Mir142hg*, *Cyth4*, *Gpr34*, *Plek*, *Ccl9*, *Lcp1*, *Aif1*, *Psmb9*, *Laptm5*, *Rac2*, *Lair1*, *Bcl3*, *Gngt2*, *Hk3*, *Csf2rb*, *Tlr12*, *Phf11d*, *Adora3*, *Apbb1ip*, *Pycard*, *Uba7*, *Rab7b*, *Vav1*, *Oas1a*, *Hcst*, *Ifi27l2a*, *Hexb*, *Fcgr2b*, *Tnni2*, *Ncf4*, *Psmb8*, *Fcrls*, *Sp110*, *Irf5*, *Havcr2*, *Mlxipl* (process enrichment analysis: (i) immune effector process (GO:0002252, Log(10)P: -6.23); (ii) phagocytosis (GO:0006909, Log(10)P: -5.10); (iii) adaptive immune system (R-MMU-1280218, Log(10)P: -4.70); (iv) regulation of immune effector process (GO:0002697, Log(10)P: -4.70); (v) neutrophil degranulation (R-MMU-6798695, Log(10)P: -4.42); (vi) beta-catenin independent WNT signaling (R-MMU-3858494, Log(10)P: -4.10); and (vii) Tyrobp causal network in microglia (WP3625, Log(10)P: -3.79).

**Cross-sex and sex-specific DEGs in the hippocampus of APP/PS1 mice vs. controls.** Finally, we investigated the DEGs (FC > 1.5) in both female and male hippocampus of APP/PS1 vs. WT mice (**Fig 5D**). In total, 45 candidates were upregulated solely in the female hippocampus, 25 were upregulated only in the male hippocampus, but 32 candidates turned out to be co-upregulated in the hippocampus of both male and female APP/PS1 AD vs. WT mice (compared to 52 in the RS cortex, see above). The individual gene candidates are also listed in **S12A Table in S1 File**. The cross-sex (intersectional) GO and enrichment profiling in the hippocampus revealed the following top three categories for co-upregulated gene candidates: (i) immune effector process (GO:0002252, Log(10)P: -9.64), (ii) macrophage markers (WP2271, Log(10)P: -9.08), and (iii) defense response to bacterium (GO:0042742, Log(10)P: -8.63).

Notably, 45 upregulated transcripts of related candidates in the hippocampus were solely attributable to female APP/PS1 mice, i.e, *Il4i1*, *Arhgap9*, *Fam46c*, *Prss23*, *Ptpn6*, *Slc38a5*, *Ifitm3*, *Laptm5*, *Ctsz*, *Cyba*, *Igf1*, *Gngt2*, *Icam1*, *Hcls1*, *Dclk1*, *Uba7*, *Rab7b*, *Rfx3*, *Tubb6*, *Slfn5*, *C1qb*, *P2ry6*, *Flnc*, *Apobec1*, *Lsp1*, *Hcst*, *Myo1g*, *Itgb2*, *Tbxas1*, *Vim*, *5830408C22Rik*, *Tnni2*, *Fcer1g*, *Capg*, *Ptgr1*, *Psmb8*, *Sp110*, *Krt85*, *Ptprc*, *Lcp1*, *Pla2g4a*, *Siglech*, *Rab32*, *Slamf9* (process enrichment analysis: (i) microglia pathogen phagocytosis pathway (WP3626, Log(10)P: -7.73); (ii) IL-5 signaling pathway (WP151, Log(10)P: -6.61); (iii) adaptive immune system (R-MMU-1280218, Log(10)P: -6.27); (iv) regulation of leukocyte cell-cell adhesion (GO:1903037, Log(10)P: -5.01); (v) innate immune response (GO:0045087, Log(10)P: -4.98); (vi) leukocyte mediated immunity (GO:0002443, Log(10)P: -4.96); phagocytosis (GO:0006909, Log(10)P: -4.88); (vii) positive regulation of phosphatidylinositol 3-kinase signaling (GO:0014068, Log(10)P: -4.67); (viii) eicosanoid metabolism via cyclooxygenases (COX) (WP4347, Log(10)P: -4.48); (ix) supramolecular fiber organization (GO:0097435, Log(10)P: -3.95); (x) regulation of

sequestering of calcium ion (GO:0051282, Log(10)P: -3.87); (xi) phagosome (mmu04145, Log(10)P: -3.31). In the hippocampus of male APP/PS1, transcripts of 25 candidates were selectively upregulated, i.e., *Sgk1, Ncf1, Adamtsl4, Ctss, Cd300c2, Hck, St8sia6, Fcgr1, Gm20743, Rac2, Irf9, Pfkfb3, B2m, Arrdc2, Cebpa, Phf11d, Synpo2, Cep126, Rhoh, Hist1h2be, Fcgr2b, Fcgr3, Clec5a, Cd86, Ptpn18* (process enrichment analysis: (i) antigen processing and presentation of exogenous peptide antigen (GO:0002478, Log(10)P: -9.92); (ii) microglia pathogen phagocytosis pathway (WP3626, Log(10)P: -6.95); (iii) antigen processing and presentation of exogenous peptide antigen via MHC class II (GO:0019886, Log(10)P: -5.81) (iv) regulation of leukocyte activation (GO:0002694, Log(10)P: -4.26).

**Sex- and region-specific transcriptome fingerprints / signature genes in APP/PS1 AD mice vs. controls.** A Venn analysis including DEGs in all four APP/PS1 subgroups is depicted in Fig 5E. Note that there are sets of DEGs that are exclusively upregulated in only one of these APP/PS1 subgroups (vs. controls) and can thus serve as molecular/transcriptome fingerprint genes/signature genes for BROI- and sex-specific alterations in this APP/PS1 AD model. In the RS cortex of female APP/PS1 this includes 25 out of 91 DEGs, 17 out of 77 DEGs in the hippocampus of female APP/PS1, only two out of 57 DEGs in the RS cortex of male APP/PS1 mice and finally, 18 out of 57 DEGs in the hippocampus of male APP/PS1 animals.

Enrichment analysis of the 25 upregulated signature genes in the female APP/PS1 RS cortex (Fig 5E) revealed the following categories: (i) inflammatory response (GO:0006954, Log(10)P: -5.58), (ii) Tyrobp causal network in microglia (WP3625, Log(10)P: -4.36), (iii) neutrophil degranulation (R-MMU-6798695, Log(10)P: -3.48), (iv) phagocytosis (GO:0006909, Log(10)P: -3.12), (v) RAF/MAP kinase cascade (R-MMU-5673001, Log(10)P: -2.44).

Additional enrichment studies for the 17 upregulated signature genes in the female APP/PS1 hippocampus (Fig 5E) revealed the following categories: (i) epithelial cell development (GO:0002064, Log(10)P: -3.34); (ii) membrane trafficking (R-MMU-199991, Log(10)P: -2.22), (iii) supramolecular fiber organization (GO:0097435, Log(10)P: -2.17), (iv) adaptive immune system (R-MMU-1280218, Log(10)P: -2.01).

The two upregulated signature genes in the male APP/PS1 RS cortex (*Derl3* and *Mki67*, Fig 5E) revealed no significant enrichment terms.

Enrichment analysis for the 18 upregulated signature genes in the male APP/PS1 hippocampus (Fig 5E) revealed the following categories: (i) positive regulation of leukocyte activation (GO:0002696, Log(10)P: -3.67), (ii) signaling by receptor tyrosine kinases (R-MMU-9006934, Log(10)P: -2.50), (iii) regulation of cytoskeleton organization (GO:0051493, Log(10)P: -2.15), and (iv) supramolecular fiber organization (GO:0097435, Log(10)P: -2.08).

In addition, we performed pathway studies using the *Reactome Pathway Database* (for details, please refer to "reactome.org"). We first analyzed intersectional gene sets for upregulated DEGs (FC > 1.5). Notably, for all intersectional genes sets (RS cortex and hippocampus of female APP/PS1 mice, RS cortex and hippocampus of male APP/PS1 mice, male and female RS cortex of APP/PS1 mice, male and female hippocampus of APP/PS1 mice, see also Fig 5A–5D), the top three pathways were related to (i) the innate immune system, (ii) the immune system and (iii) neutrophil degranulation (**S6 Fig** in **S1 File**). Thus, the transcriptional overlay between the individual subgroups is very much conserved. We then investigated pathway characteristics for the significantly upregulated signature genes of the individual subgroups (Fig 5E). The signature gene sets again turned out to exhibit pathway specificities: In the female APP/PS1 hippocampus, the top three pathways included (i) interleukin-4 and interleukin-13 signaling, (ii) interferon signaling, and (iii) aggrephagy. In the female APP/PS1 RS cortex, the top three pathways were related to (i) hyaluronan uptake and degradation, (ii) hyaluronan metabolism, and (iii) the immune system. The most relevant signature pathways in the male APPPS1

hippocampus included (i) the transcriptional regulation of granulopoiesis, (ii) cytokine signaling in the immune system, and (iii) interferon gamma signaling. In the male APPPS1 RS cortex, the latter comprised (i) defective CFTR causes cystic fibrosis, (ii) ABC transporter disorders, and (iii) ABC-family proteins mediated transport (**S6 Fig in S1 File**).

## Venn analysis of downregulated DEGs (FC < -1.5) in the individual subgroups

We first investigated DEGs (FC < -1.5) in both the RS cortex and hippocampus of female APP/PS1 vs. WT mice (**Fig 6A**). In total, 14 candidates were exclusively downregulated in the female hippocampus of APP/PS1 mice, two were exclusively downregulated in the cortex of female APP/PS1 mice, but only one candidate was co-downregulated in both the cortex and hippocampus of female APP/PS1 AD vs. WT mice, i.e., *Pisd-ps3* (**S9B Table in S1 File**). Candidate gene transcripts that were selectively downregulated in the RS cortex of female APP/PS1 mice included *LOC105247294* and *Ptpn6*. The DEGs that were selectively downregulated in the hippocampus of female APP/PS1 mice comprised *Asic4*, *Pcsk1*, *Etnppl*, *Myh7b*, *Spag5*, *Arpp21*, *Gpr1*, *Myh8*, *Ccdc184*, *Pdzph1*, *Trim66*, *BC030499*, *Rims3*, *Prr16*.

Secondly, we investigated the DEGs (FC < -1.5) in both the RS cortex and hippocampus of male APP/PS1 vs. WT mice (**Fig 6B**). Six candidates were exclusively downregulated in the hippocampus of male APP/PS1 mice, four were exclusively downregulated in the cortex of male APP/PS1 mice, but only one candidate was co-downregulated in both regions, i.e., *Arpp21* (**S10B Table in S1 File**). Candidate gene transcripts that were selectively downregulated in the RS cortex of male APP/PS1 mice included *Scyl2*, *Ctr9*, *Scyl2*, and *1700001L05Rik*. Candidate gene transcripts that were selectively downregulated in the hippocampus of male APP/PS1 mice comprised *4933407I18Rik*, *Hlcs*, *Hdac9*, *Pla2g4e*, *Wnt9a*, *Shisa9*.

Next, we investigated the DEGs (FC < -1.5) in the RS cortex of both male and female APP/PS1 vs. WT mice (**Fig 6C**). Three candidates were exclusively downregulated in the RS cortex of female APP/PS1 mice, five were exclusively downregulated in the cortex of male APP/PS1 mice, but no intersectional gene candidates were detected in the RS cortex of both sexes (**S11B Table in S1 File**). In the RS cortex of male APP/PS1 mice only the following genes were downregulated: *6530402F18Rik*, *Ctr9*, *Arpp21*, *Scyl2*, *1700001L05Rik*. In the RS cortex of females, only the following genes were downregulated: *LOC105247294*, *Ptpn6*, *Pisd-ps3*.

Finally, we elaborated the DEGs (FC < -1.5) in the hippocampus of both male and female APP/PS1 vs. WT mice (**Fig 6D**). In total, 14 candidates were exclusively downregulated in the hippocampus of female APP/PS1 mice, 6 were exclusively downregulated in the hippocampus of male APP/PS1 mice, but only one candidate was co-downregulated in the hippocampus of both sexes. i.e., *Arpp21*. (**S12B Table in S1 File**). The following candidate transcripts were selectively downregulated in the male hippocampus of APP/PS1: *4933407I18Rik*, *Hlcs*, *Hdac9*, *Pla2g4e*, *Wnt9a*, *Shisa9*. Those DEGs that were selectively downregulated in the female hippocampus of APP/PS1 comprised: *Asic4*, *Pcsk1*, *Etnppl*, *Myh7b*, *Spag5*, *Gpr1*, *Myh8*, *Ccdc184*, *Pdzph1*, *Pisd-ps3*, *Trim66*, *BC030499*, *Rims3*, *Prr16*.

A Venn analysis including intersectional and signature DEGs in all four APP/PS1 subgroups compared to WT animals is depicted in **Fig 6E**. Note that there are again sets of DEGs that are exclusively downregulated in only one of these subgroups and can thus serve as molecular/transcriptome fingerprints/signature gene sets for BROI- and sex-specific alterations in this APP/PS1 AD model. In the RS cortex of female APP/PS1 mice, this includes two out of three DEGs, 13 out of 15 DEGs in the hippocampus of female APP/PS1, four out of five DEGs in the RS cortex of male APP/PS1 mice and finally, six out of seven DEGs in the hippocampus of male APP/PS1 animals. Interestingly, although the number of downregulated DEGs is

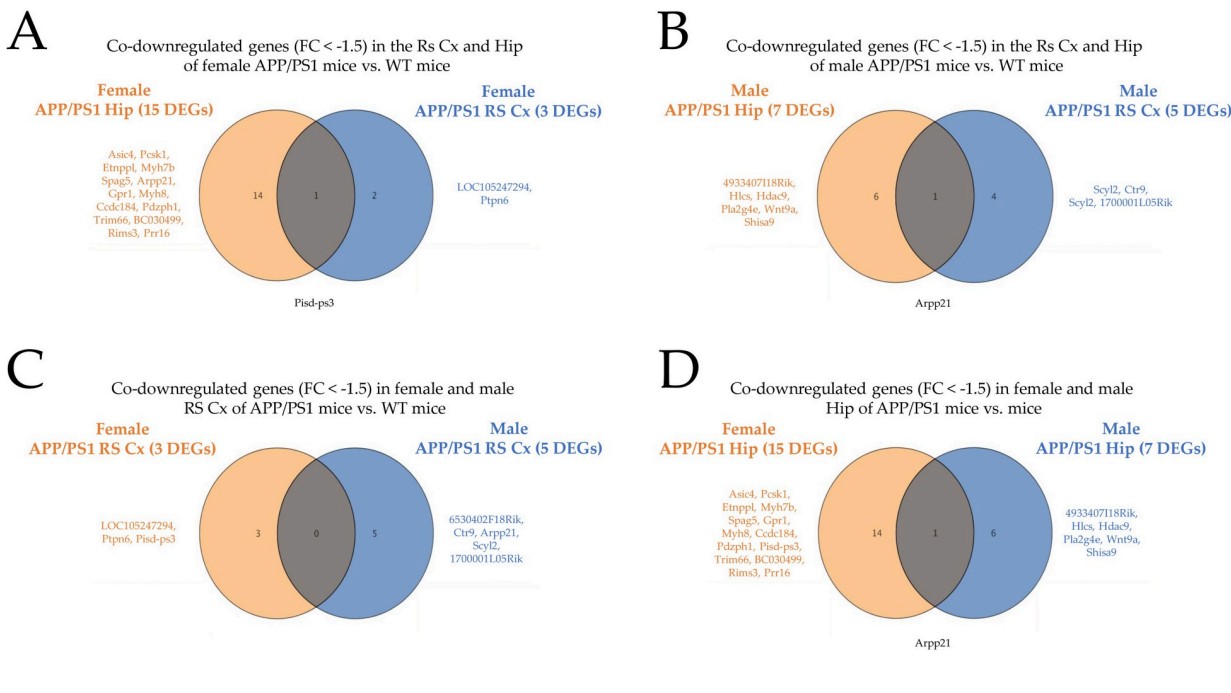

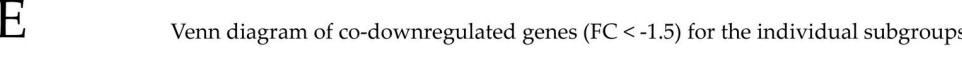

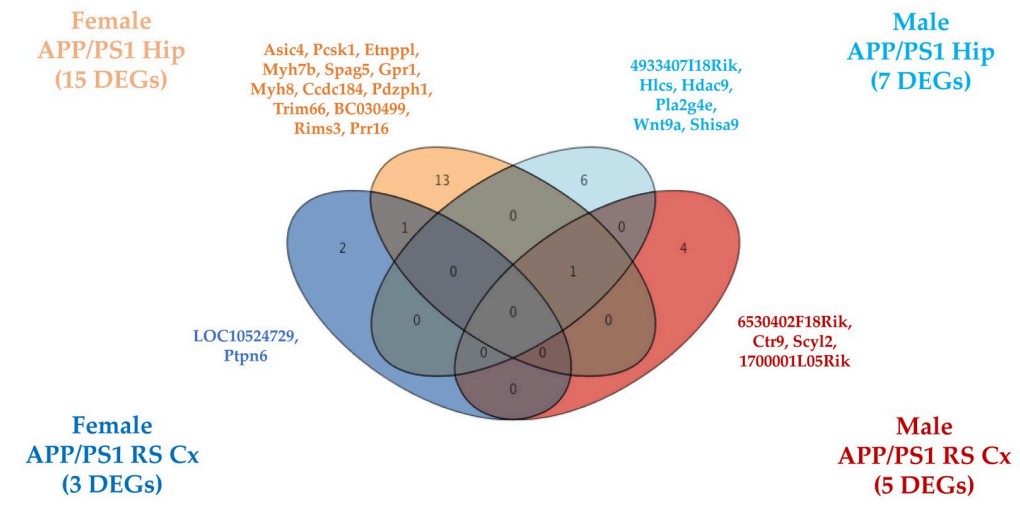

**Fig 6. Venn diagrams of downregulated DEGs (FC < -1.5) in APP/PS1 AD vs. WT control mice related to either both brain regions (RS cortex and hippocampus) or both sexes.** Intersectional DEGs in (A) the RS cortex and hippocampus of female APP/PS1 AD mice, (B) the RS cortex and hippocampus of male APP/PS1 AD mice, (C) the RS cortex of female and male APP/PS1 AD mice and (D) the hippocampus of female and male APP/PS1 mice. Intersectional gene symbols (in black) are listed below each diagram, the other gene symbols are depicted next to the diagram (in orange and blue). E) Venn-diagram including all four study groups. Note that there are specific DEGs (signature genes, listed here) that are selectively downregulated only in one of the four study groups and thus represent a unique transcriptome profile for sex and BROI. Strikingly, the number of downregulated gene candidates is severely lower compared to the upregulated ones.

significantly lower compared to the upregulated candidates, the majority of downregulated DEGs contributes to their related BROI- and sex-specific transcriptome fingerprints. Gene ontology and enrichment studies using Metascape did not reveal significantly enriched terms/ categories.

Due to the minimal or lacking number of intersectional downregulated genes, pathway analysis was not applicable here (**Fig 6A–6D**). The downregulated signature gene sets however, revealed interesting specificities (**Fig 6E**): In the female hippocampus of APP/PS1 mice, the top three signature pathways included prostanoid ligand receptors, peptide hormone biosynthesis, and the synthesis, secretion, and inactivation of Glucose-dependent insulinotropic polypeptide (GIP). In the female RS cortex of APP/PS1 mice, the top signature pathways encompassed interleukin-37 signaling, interleukin-1 family signaling, and PECAM1 interactions. In the hippocampus of male APP/PS1 mice, authenticator genes were related to the following pathways: defective HLCS related multiple carboxylase deficiency, defects in biotin metabolism, and hydrolysis of LPC. Finally, top signature pathways in the RS cortex of male APP/PS1 mice were related to RNA polymerase II transcription elongation, formation of RNA Pol II elongation complex, and E3 ubiquitin ligases ubiquitinate target proteins (**S6 Fig in S1 File**).

## Differentially regulated l(i)ncRNAs in the cortex and hippocampus of male and female APP/PS1 mice compared to WT controls

Long (intergenic/intervening) non-coding RNAs (l(i)ncRNAs) represent a class of non protein-coding RNAs (> 200 nt) that serve important regulatory functions in transcriptional silencing or activation, chromosomal modification and intranuclear trafficking. L(i)ncRNAs are predominantly classified as antisense l(i)ncRNAs, intronic transcripts, large intergenic noncoding RNAs, promoter-associated l(i)ncRNAs and UTR associated l(i)ncRNAs.

Using the Feature Extraction tool (Agilent Technologies Germany GmbH & Co. KG, Germany), we selectively evaluated differentially regulated l(i)ncRNA entities in all four subgroups (**S13 Table in S1 File** illustrates the significantly altered l(i)ncRNAs, the related FCs, p-values and sequence characteristics). Statistics are based on one-way ANOVA, Tukey *post hoc* analysis and the Benjamini-Hochberg procedure. With only one exception (hippocampus of female APP/PS1 mice), all l(i)ncRNA related FCs were in the range of -1.5 to 1.5. Notably, the majority of differentially regulated l(i)ncRNAs was downregulated compared to WT controls. In RS cortex probes of female APP/PS1 mice, only two lincRNA candidates were identified (**S13A Table in S1 File**), whereas 14 l(i)ncRNAs were found in hippocampal probes of female APP/PS1 (**S13B Table in S1 File**). 13 l(i)ncRNAs were found in the RS cortex of male APP/PS1 (**S13C Table in S1 File**) and 13 l(i)ncRNAs in the hippocampus of male APP/PS1 mice (**S13D Table in S1 File**).

The *in silico* analysis of l(i)ncRNAs was done using RNAcentral („https://rnacentral.org/") [75], Rfam ("https://rfam.org/") [76], the Coding-Potential Assessment Tool (CPAT, "http://rna-cpat.sourceforge.net") [77], and various databases, e.g., NONCODE ("http://www.noncode.org/") [78].

All l(i)ncRNAs differentially transcribed in the individual subgroups in our study turned out to be annotated by the NONCODE database. One l(i)ncRNA sequence was annotated by five databases with known presence in the hippocampus and reports about involvement in epigenetic effects [79]. In most cases, database information about l(i)ncRNA tissue distribution nicely paralleled our findings. However, detailed functional implications of up- or downregulated l(i)ncRNAs in the pathophysiology of the individual APP/PS1 subgroups remain to be determined in the future.

## qPCR analysis of selected gene candidates relevant in AD etiopathogenesis

**Comparison of selected DEGs between APP/PS1 and 5XFAD mice.** Our own previous radiotelemetric EEG studies from the cortex and hippocampus of 5XFAD mice, another model of AD disease, had revealed complex alterations in CNS rhythmicity [31, 32]. Similarly, our electrophysiological studies in APP/PS1 mice had showed age- and sex-dependent differences in the relative power of the theta and gamma frequency range in the cortex and hippocampus [13, 14]. Our own previous transcriptome studies in 5XFAD mice had revealed some gene candidates that might be functionally relevant for the electrophysiological alterations observed and some had been validated using qPCR [31, 32]. As APP/PS1 mice exhibit similar phenotypical features as 5XFAD animals, we analyzed the same set of critical 5XFAD related genes via qPCR in the APP/PS1 model, i.e., *Cacna1c*, *Cacna1d*, *Plcd4*, *Casp8*, *Chrm1*, *Chrm3*, *and Chrm5*, to elaborate whether there are common, relevant DEGs in both AD mouse models (first qPCR approach). In female APP/PS1 mice, Caspase 8 (encoded by *Casp8*) was significantly upregulated compared to WT controls (FC: 1.3674, $p = 0.028$), but not the other candidates (**S14A Table and S7A and S7B Fig in S1 File**). Interestingly, Caspase 8 did not exhibit altered expression in male APP/PS1 mice (FC: 1.0892, $p = 0.685$). In contrast, the high-voltage activated L-type $Ca^{2+}$ channel $Ca_v1.3$ (encoded by *cacna1d*) was significantly downregulated in male APP/PS1 mice (FC: -1.2122, $p = 0.0285$) but not in female APP/PS1 animals (FC: 1.1681, $p = 0.114$) (**S7E and S7F Fig in S1 File**). This sex-specific difference in $Ca_v1.3$ transcript levels in APP/PS1 mice was statistically confirmed (FC: 1.4113, $p = 0.0285$) (**S7E and S7F Fig in S1 File**). For the high-voltage activated L-type $Ca^{2+}$ channel $Ca_v1.2$ (encoded by cacna1c), no genotype- and sex-specific difference in transcript levels was detected. A significant difference in transcript levels between male and female APP/PS1 mice was also observed for *Chrm1*, the gene encoding the muscarinic receptor type 1 (M1) (**S7I and S7J Fig in S1 File**). These findings further stress the fundamental relevance of sex-specific analysis as carried out in our transcriptome studies.

Our transcriptome data revealed no significant up- or downregulation of *Casp8*, *Plcd*, *Chrm1*, *Chrm3* and *Chrm5*. In the RS cortex and hippocampus of female APP/PS1 mice and the RS cortex of male APP/PS1 mice, PLCgamma was altered (FC: 1.3, FC: 1.4 and FC: 1.3, respectively). In the hippocampus of male APP/PS1 mice, PlCd3 was downregulated (FC: -1.2). Importantly, *cacna1d* encoding the $Ca_v1.3$ HVA $Ca^{2+}$ channel was listed as a DEG in the RS cortex of male APP/PS1 mice, but not in the hippocampus. This phenomenon is due to the applied strict statistical procedure including one-way ANOVA, Tukey *post hoc* test and the application of the Benjamini-Hochberg procedure to decrease the FDR. The DEG extraction without application of *post-hoc* correction again included *cacna1d* as a differentially regulated gene candidate. These qPCR findings underline that different AD mouse lines, i.e., 5XFAD and APP/PS1, can clearly exhibit different transcriptome characteristics.

**Validation of selected DEGs in APP/PS1 mice.** In the second qPCR approach, we validated a selected number of DEGs from our APP/PS1 transcriptome study, i.e., four upregulated (*Siglech*, *Ptpn6*, *Laptm5*, *Plek*, **Fig 7A–7H**) and two downregulated (*Arpp21*, *Shisa9*, **Fig 7I–7L**) genes. These DEGs were analyzed in the RS cortex of both male and female APP/PS1 animals (same mice (RNA) as utilized for our transcriptome studies). As qPCR analysis of all relevant DEGs, including intersectional and signature genes, would be beyond the scope of our study, we focused on a selected number of candidates in the RS cortex. The latter is early and severely affected in AD pathogenesis. Importantly, the four DEGs *Siglech*, *Ptpn6*, *Laptm5*, *and Plek* which were suggested to be significantly upregulated in female APP/PS1 mice in our transcriptome study, were also significantly upregulated in the qPCRs (**Fig 7B, 7D, 7F and 7H and S14B Table in S1 File**). The same held true for the qPCR validation of *Siglech*, *Ptpn6*,

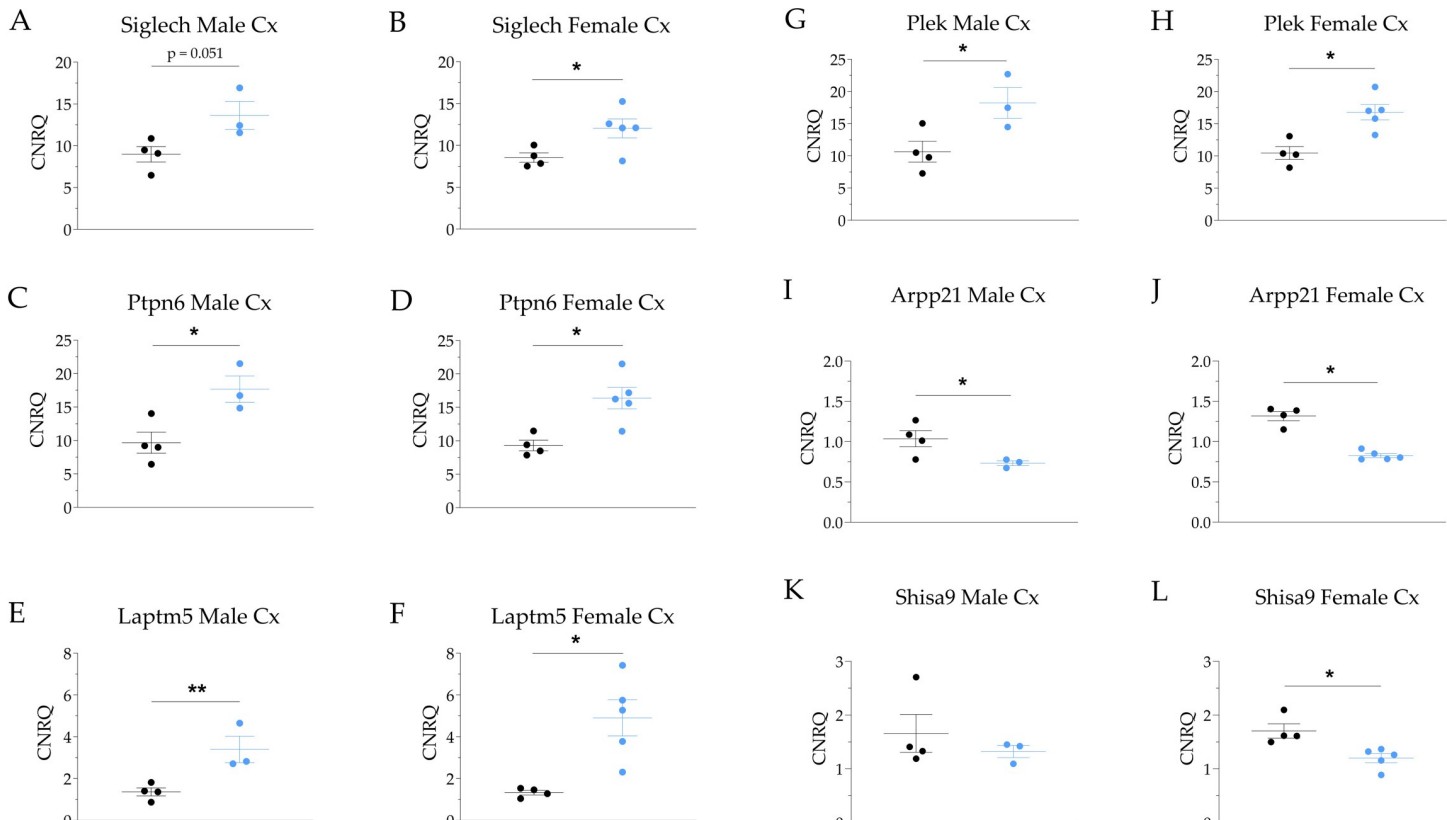

**Fig 7. Comparative qPCR analysis of selected DEGs from the RS cortex of female and male APP/PS1 AD.** Transcript levels (CNRQ) for *Siglech* (A, B), *Ptpn6* (C, D), *Laptm5* (E, F), *Plek* (G, H), *Arpp21* (I, J), *Shisa9* (K, L) were obtained from eight APP/PS1 AD mice (four ♂, four ♀) and eight WT control animals (three ♂, five ♀). Results are depicted using scatter plots including mean ± SEM. RNA was taken from the same animals analyzed in the transcriptome approach. Control mice are shown in black, APP/PS1 animals are highlighted in light blue.

*Laptm5, and Plek* in the <u>male</u> RS cortex of APP/PS1 animals (**Fig 7A, 7C and 7E**) with *Siglech* exhibiting a statistical trend (**S14B Table in S1 File**). In addition, the FC-values from the transcriptome study and the qPCR validation nicely correlate. Note that many DEGs (such as *Ptpn6*) exhibit different transcript variants which can be up- or downregulated. For *Ptpn6*, the upregulated transcript variants were validated.

The two DEGs that were suggested to be downregulated in our microarray analyses (*Arpp21, Shisa9*), were also confirmed in our qPCR approach in the RS cortex of <u>female</u> APP/PS1 mice (**Fig 7J and 7L** and **S14B Table in S1 File**). In the RS cortex of <u>male</u> transgenics however, significant downregulation of *Shisa9* could not be confirmed (**S14B Table in S1 File**). Overall, our qPCR findings underline the validity of our transcriptome studies and the detected DEGs.

## Discussion

### The role of APP/PS1 mice in AD research and transcriptomics–previous results and open challenges addressed in our study

The APP/PS1 AD mouse model investigated in our study exhibits moderate severity and disease progression that well resembles AD in humans based on the categories of homology, isomorphism, and predictability [19, 22]. We previously performed complex electrophysiological

*in vivo* analysis of APP/PS1 mice using implantable EEG radiotelemetry to elucidate defined alterations in hippocampal and cortical rhythmicity via FFT based frequency analysis [13, 14]. We detected region-, age- and sex-specific differences in EEG characteristics that stress - together with findings of other groups - that functional alterations in AD pathophysiology clearly require subgroup-specific analyses [13, 14, 80]. Although transcriptome studies have previously been carried out in APP/PS1 AD mice, these studies exhibit some limitations in design that affect their informative value. The latter is mainly related to the combined analysis of sexes [37, 38, 81], the selective analysis of males only [41–43], an unspecified sex distribution [39, 82], analysis of only one brain region [37, 39, 40, 43, 81], investigation of no distinct brain region (e.g., cortex in total [41, 42] or the analysis of entire brain probes [38] (**S15 Table in S1 File**). Based on our previous findings on distinct electrophysiological alterations in the APP/ PS1 AD model related to sex, age and brain region [13, 14], we performed transcriptome studies in the RS cortex and hippocampus of both sexes in WT controls and APP/PS1 mice.

## Role of the RS cortex and hippocampus in AD etiopathogenesis and disease progression

The cortex is known to be early affected in AD pathophysiology both on the structural and functional level [52, 83]. Interestingly, progression of AD-specific lesions severely differs between individual cortical regions. Whereas the (pre)frontal cortex is early involved, alterations in the motor cortex seem to be delayed [84, 85]. We have chosen the RS cortex for our transcriptome studies, as its early and severely affected in AD pathology and symptomatology [50, 86]. As another predilection area in the etiopathogenesis of AD, the hippocampus was chosen as a second BROI for our transcriptome analysis. The hippocampus is judged to be a key interface structure in the brain and involved in the consolidation of memory engrams and their ecphoria [47, 87]. Impaired and dysfunctional hippocampal activity together with complex structural alterations related to extracellularly localized Aβ plaque formation and intracellular τ-deposits in APP/PS1 AD mice are typically accompanied by sophisticated deficits in memory formation, and cognitive/behavioral tests [88, 89].

## Functional relevance of selected DEGs in APP/PS1 AD mice

Our data illustrate, that key gene candidates in AD etiopathogenesis are differentially regulated in brain regions of male and female APP/PS1 mice compared to WT controls (**Tables 3 and 4**). We identified some gene candidates that had already been related to AD, whereas others have not yet been attributed in detail or at all to its pathophysiology. An overall discussion of the entire set of high fold change DEGs in the individual subgroups is beyond the scope of our study. However, we have pointed out some functional aspects of selected DEGs and how they might influence the etiopathogenesis of AD with a focus on sex- and region-specific distribution. As outlined below, DEGs are predominantly related to the neuroinflammatory response, the activation of microglia, macrophages and neurophil granulocytes [37, 90]. Many DEGs further contribute to innate immune reactions and immune effector processes that are largely triggered by Aβ [90, 91].

**Intersectional DEGs in microglial/neutrophilic activation.** Microglial activity seems to play a critical role in the etiopathogenesis of AD and the devastating structural and functional alterations that encompass progression of the disease [92–94]. Microglia represent the resident immune cells within the CNS and share some phenotypic markers with myeloid cells of the macrophage/monocyte lineage [95]. Microglia have been functionally segregated into pro-inflammatory M1 and anti-inflammatory M2 subgroups based on their molecular and functional profile [96, 97], though the entire activation spectrum between the maxima seems to a

**Table 3. Overview of the sex- and region-specific upregulation of DEGs in the four APP/PS1 subgroups.** This table illustrates all upregulated DEGs (FC > 1.5) from APP/PS1 microarray analysis. DEGs that are upregulated only in one APP/PS1 subgroup are highlighted in red, those exhibiting enhanced transcript levels in two subgroups are marked in green. Increased transcript levels in three subgroups are marked in ocher, those in all four subgroups are highlighted in yellow. The table elicits the complex sex- and brain region specific profile of upregulated DEGs.

| APP/PS1 mice DEGs (FC > 1.5) | ♀ Cx | ♀ Hip | ♂ Cx | ♂ Hip |
|---|---|---|---|---|
| Adora3, Aif1, Apbb1ip, Ccl9, Csf2rb, Cyth4, Fcrls, Gpr34, Gusb, Havcr2, Hexb, Hk3, Ifi27l2a, Irf5, Lair1, Mir142hg, Mlxipl, Ncf4, Oas1a, Plek, Psmb9, Pycard, Slc14a1, Tlr12, Vav1 | ■ (red) | | | |
| 5830408C22Rik, Apobec1, Dclk1, Flnc, Icam1, Ifitm3, Lsp1, Myo1g, Pla2g4a, Prss23, Ptgr1, Rab32, Rfx3, Slc38a5, Slfn5, Tubb6, Vim | | ■ (red) | | |
| Derl3, Mki67 | | | ■ (red) | |
| Adamtsl4, Arrdc2, B2m, Cd86, Cebpa, Cep126, Clec5a, Gm20743, Hck, Hist1h2be, Irf9, Ncf1, Pfkfb3, Ptpn18, Rhoh, Sgk1, St8sia6, Synpo2 | | | | ■ (red) |
| Arhgap9, Gngt2, Hcst, Laptm5, Lcp1, Psmb8, Rab7b, Sp110, Tnni2, Uba7 | ■ (green) | | ■ (green) | |
| Ctsd, Lag3, Lgals3bp, Ncf2 | ■ (green) | ■ (green) | | |
| Ctsh, Osmr | | ■ (green) | | ■ (green) |
| Fcgr2b, Phf11d, Rac2 | ■ (green) | | | ■ (green) |
| 5730416F02Rik, Capg, Ctsz, Cyba, Fam46c, Fcer1g, Hcls1, Igf1, Il4i1, Itgb2, P2ry6, Ptpn6, Ptprc, Siglech, Slamf9, Tbxas1 | ■ (blue) | ■ (blue) | ■ (blue) | ■ (blue) |
| Bcl3 | ■ (blue) | ■ (blue) | | ■ (blue) |
| Aspg, Fyb | | ■ (blue) | ■ (blue) | ■ (blue) |
| Cd300c2, Ctss, Fcgr1, Fcgr3 | ■ (blue) | | ■ (blue) | ■ (blue) |
| Bcl2a1d, C1qa, C1qc, C4b, Ccl6, Cd14, Cd48, Cd52, Cd68, Gfap, Gpr84, Hvcn1, Irf8, LOC108167440, Ly86, Lyz1, Lyz2, Mpeg1, Myo1f, Naip2, Prnp, Samsn1, Slc11a1, Syngr1, Tlr7, Trem2, Tyrobp | ■ (yellow) | ■ (yellow) | ■ (yellow) | ■ (yellow) |

continuum [74, 98]. Strong efforts have been made to decipher the phenotypic diversity of microglial cells in the healthy, aging, and AD brains of mice and humans. AD-dysprogrammed microglia are often referred to as disease-associated microglia (DAM) [99], microglia associated with neurodegenerative disease (MGnD) [100], or activated response microglia (ARM) [101]. The delineation of transcriptome profiles in these microglia has led to the specification of gene candidates related to the homeostatic signature of microglia, differentiation of microglia vs. myeloid populations, microglia vs. macrophages, the microglial surfaceome, and the microglial sensome [74, 102]. Detailed comprehensive lists of microglia associated transcripts are provided by Boche and Gordon (2020) [74] and Crotti A et al. (2016) [74, 103].

**Table 4. Overview of the sex- and region-specific downregulation of DEGs in the four APP/PS1 subgroups.** This table illustrates all downregulated DEGs (FC < -1.5) from APP/PS1 microarray analysis. DEGs that are downregulated only in one APP/PS1 subgroup are highlighted in red, those exhibiting enhanced transcript levels in two subgroups are marked in green. Increased transcript levels in three subgroups are marked in ocher, those in all four subgroups are highlighted in yellow. The table elicits the complex sex- and brain region specific profile of downregulated DEGs.

| APP/PS1 DEGs (FC < -1.5) | ♀ Cx | ♀ Hip | ♂ Cx | ♂ Hip |
|---|---|---|---|---|
| LOC105247294, Ptpn6 | ■ (red) | | | |
| Asic4, BC030499, Ccdc184, Etnppl, Gpr1, Myh7b, Myh8, csk1, Pdzph1, Prr16, Rims3, Trim66 | | ■ (red) | | |
| 1700001L05Rik, 6530402F18Rik, Ctr9, Scyl2 | | | ■ (red) | |
| 4933407I18Rik, Hdac9, Hlcs, Pla2g4e, Shisa9, Wnt9a | | | | ■ (red) |
| Pisd-ps3 | ■ (green) | ■ (green) | | |
| Spag5 | ■ (green) | | | ■ (green) |
| Arpp21 | ■ (yellow) | ■ (yellow) | ■ (yellow) | ■ (yellow) |

Recent advances in single-cell-omic methodologies have dramatically increased the availability of gene expression data from microglia in healthy and aged brains, and AD [74]. Previous studies by Mukherjee et al. (2019) described DEGs in microglia related to neurodegenerative diseases, including *Tyrobp*, *Fcer1g*, *Itgb2*, *Myo1f*, *Ptprc*, *Trem2* and *C1qa* [104]. We were able to confirm many of these gene candidates in our experimental subgroups and most important, in both sexes.

The *Fcer1g* gene for example, plays an important role in IgE-specific FC gamma receptor mediated phagocytosis and microglial activation [105]. Importantly, upon activation of *Trem2*, *Syk* is recruited by *Fcer1g* and *Tyrobp* [106, 107]. Notably, *Tyrobp* and *Trem2* were upregulated in all four APP/PS1 subgroups in our study, similar to *Fcer1g* which however, remained unaltered in the hippocampus of male APP/PS1 mice.

Some DEGs that were upregulated in our study are functionally related to leucocyte migration, e.g., *Itgb2* [108]. *Myo1f* is important in neutrophil degranulation and regulation of cell migration as well [108]. *Rac2* encodes a protein with GTPase and protein kinase regulator activity and is involved in the regulation of leukocyte activation and leukocyte chemotaxis. It belongs to the Rho family GTPases which are master regulators of actin cytoskeletal organization and associated with a number of neuropsychiatric and neurodegenerative diseases, including AD [109].

We further identified several cluster of differentiation (Cd) encoding genes to be upregulated in APP/PS1 AD mice. Most of these Cd genes, i.e., *Cd48*, *Cd52* and *Cd68*, are involved in antigen binding, signaling and receptor activity. Some, like *Cd68*, were reported to be involved in aging and cellular response to organic substances with critical localization in lysosomes within the brain. The gene product of *Trem2* forms a receptor signaling complex that activates myeloid cells, including dendritic cells and microglia. In humans and mice, mutations in *Trem2* may serve as risk factors for the development of AD [110].

Further relevant genes are related to the interleukin, interferon, and cytokine system, e.g., the C-C motif chemokine 6, encoded by *Ccl6*, one of the top FC candidates in our study. The latter displays expression in microglia and astrocytes. Interestingly, treatment of neurons with CCL6 revealed protective effects against glutamate neurotoxicity. This neuroprotective effect of CCL6 is mediated via the chemokine receptor CCR1, and downstream by PI3K and likely to be another protective mechanism in AD pathophysiology [111]. For additional description of the DEGs, please refer also to **S16 Table in S1 File**.

**Intersectional genes in phagocytic activation.**   Phagocytosis is an important field in AD pathogenesis [112] and numerous DEGs in our study groups are related to this process. *Capg* (encoding for capping actin protein, gelsolin like), for example, is a member of the gelsolin/villin family of actin-regulatory proteins and was reported to play an important role in phagocytic vesicles and actin-related intracellular processes [113]. Others, like *Fcgr* (FC gamma receptor) are also involved in endocytosis similar to *Fcgr2b* [114]. The latter is located in dendritic spines and involved in phagocytosis, cytotoxicity and cellular response to Aβ [114]. Studies in both mice and humans suggest that FcγR mediates a pro-inflammatory response which could also be responsible for vascular side effects following application of therapeutic antibodies against Aβ [115]. Increasing evidence has emerged that FcγR expression in microglia and neurons is enhanced upon aging and involved in the etiopathogenesis of AD [115]. Glial fibrillary acid protein encoded by *Gfap* is another example of a structural constituent of the cytoskeleton that is involved in chaperone-mediated autophagy and intracellular protein transport. It was reported to be involved in long-term potentiation (LTP) and neurogenesis. The *Gfap* gene product is currently under discussion for its utility as a plasma biomarker for AD-related pathologies [116]. *Itg2* (coding for integrin beta2), *Laptm5* (encoding for lysosomal-associated protein transmembrane 5) and *Lcp1* (encoding for lymphocyte cytosolic protein 1) are

involved in cargo receptor processes and endocytosis, lysosomal transport and filament bundle assembly located in phagocytic cups [117]. Notably, we were able to validate *Laptm5* transcript upregulation via qPCR in the RS cortex of both male and female APP/PS1 mice.

The prion protein encoded by *Prnp* exhibited one of the highest fold-changes in our study. It serves several functions, including Aβ binding activity, regulation of protein localization to membranes, protein phosphorylation, aspartic-type endopeptidase inhibitor activity, negative regulation of apoptotic processes and modulation of phagocytosis [118]. There is likely to be an association between AD and cellular prion protein (PrPC) levels. PrPC exerts high affinity for oligomeric Aβ (particularly Aβ42) and mediates Aβ-induced neurotoxicity in AD pathogenesis [119]. Synaptotoxic Aβ assemblies further mediate long-term depression (LTD) via action on PrPC [120]. PrPC is decreased in the hippocampus and temporal cortex in aging and sporadic AD but not in familial Alzheimer's disease (FAD) [121]. Another study also suggested a reduced expression of PrPC in AD patients compared to healthy individuals, which pointed to a potential protective role of PrPC expression in AD [122], although this remains controversial [123]. However, with symptomatic clinical manifestation, PrPC expression seemed to decrease again [123]. There is clearly an ambiguous picture of PrPC expression levels in AD that needs future clarification [118]. Interestingly, *Prnp* was strongly co-upregulated in the RS cortex and hippocampus of both female and male APP/PS1 mice in our study.

One of the highest FC candidates in our study turned out to be glycosidase lysozyme 2 (*Lyz2*). Alzheimer's disease patients were reported to exhibit increased lysozyme levels in the cerebrospinal fluid. In addition, lysozyme 2 co-localizes with Aβ and exerts protective effects against Aβ associated neurodegenerative action. Importantly, lysozyme was proposed as a potential biomarker and therapeutic target in AD [124].

The gene product of *Siglech* (sialic acid binding Ig-like lectin H, possibly an ortholog of CD33 in humans) has cargo receptor activity and acts upstream of or within receptor-mediated endocytosis and was reported to serve as a risk factor for late-onset AD [125]. Importantly, *Siglech* was upregulated in each APP/PS1 subgroup in our study despite the hippocampus of male AD animals. Our qPCR experiments further validated the upregulation of *Siglech* in the cortex of APP/PS1 mice. For additional DEG profile information see also **S16 Table in S1 File**.

**Intersectional genes encoding for endopeptidases in APP/PS1 AD mice.** Endopeptidases play an important role in AD pathology and are potential drug target candidates and biomarkers with high predictive efficacy [126]. We identified several endopeptidases to be upregulated in the different APP/PS1 subgroups. The latter include *Bcl2a1d* (encoding for B cell leukemia/lymphoma 2 related protein A1d), a cysteine-type endopeptidase involved in apoptotic processes and *Ctsd*, an aspartic-type endopeptidase (cathepsin D) localized in the extracellular space and lysosomes and involved in autophagosome assembly [127]. *Bcl2a1d* turned out to be a top fold-change candidate in our study. Other upregulated endopeptidases include *Ctsh*, a cysteine protease (cathepsin H) that can form both aminopeptidase and endopeptidase activity depending on its posttranslational processing and assembly. *Ctss* represents another upregulated peptidase C1 family member of cysteine protease (cathepsin S, like *Ctsh*). *Naip2* (NLR family, apoptosis inhibitory protein 2) encodes a protein with ATP binding activity and cysteine-type endopeptidase inhibitor activity involved in apoptotic process [128]. *Naip2* was upregulated in all four APP/PS1 subgroups in our study.

Again, our study revealed sex-specific characteristics in differential regulation: the top fold-change gene candidate *Bcl2a1d* was upregulated in all APP/PS1 groups, whereas *Ctsd* and *Ctss* were not upregulated in the female hippocampus, and *Ctsh* was not upregulated in the RS cortex of male and female APP/PS1 animals. For further details please refer to **S16 Table in S1 File**.

**Oxidative stress and reactive oxygen species in APP/PS1 AD mice.** Reactive oxygen species (ROS) play an important role in AD pathogenesis and microglia-associated neuroinflammation [129]. DEGs such as *Cyba* and *Ncf* are part of the NADPH oxidase complex. They are involved in ROS metabolic processes, $Ca^{2+}$ homeostasis and AD pathogenesis [130]. The *Nrf2* encoded NF-E2-related-factor 2 was reported to suppress inflammation and oxidative stress in an APP knock-in AD mouse models and could thus exert protective effects [131]. For additional information see also **S16 Table in S1 File**.

## Sex- and region-specific alterations in gene transcript levels in APP/PS1 mice

In some AD mouse models and humans, sex-specific differences in transcriptome profiles have been partially documented. Transcriptional human-mouse intersections are mainly related to Aβ/τ pathology and microglia and exhibit sex-dependent spatiotemporal signatures upon disease progression. The results from RNA-seq of human *post mortem* brain tissue stressed the relevance of microglial and inflammatory mechanisms in AD pathogenesis, among other pathways [73, 132–134]. Notably, different from transcriptome profiles in human brains, mouse models allow more controlled experimental conditions that can be used to investigate causation, isolated effects of specific molecular or pharmacological interventions, and elaborate spatiotemporal alterations. Although transcriptional overlaps have been identified between AD mouse models and human brains in some studies [73, 135–137], other reports have questioned the overall degree of conservation [113, 138]. Regardless of these discrepancies, sex/gender has been suggested as a strong modulator in AD etiopathogenesis [73, 139, 140].

Most studies carried out in APP/PS1 mice and other AD models do not provide sex-specific analysis of transcriptome data (see **S15 Table in S1 File**). In the following, we have elaborated the sex-specific profile of the major DEGs detected in our study. Importantly, these gene candidates turned out to be signature genes (**Fig 5E**). In addition, many of them contribute to the devastating AD pathophysiology. However, some (as outline below) also exert counteracting, autoprotective effects.

**Functional relevance of signature genes in the hippocampus of female APP/PS1 mice.** In our study design, 17 gene candidates turned out to be exclusively upregulated in the hippocampus of female APP/PS1 AD mice, i.e., *Prss23, Slc38a5, Ifitm3, Icam1, Dclk1, Rfx3, Tubb6, Slfn5, Flnc, Apobec1, Lsp1, Myo1g, Vim, 5830408C22Rik, Ptgr1, Pla2g4a, Rab32*. The latter are related to the following GO categories: <u>epithelial cell development, membrane trafficking, supramolecular fiber organization, and the adaptive immune system.</u> Two related signature genes are highlighted below:

*Intercellular adhesion molecule-1 (Icam).* An interesting candidate in our study turned out to be *Icam1*, encoding for the intercellular adhesion molecule-1, a transmembrane glycoprotein that is reported to be overexpressed in many neuropathological settings. *Icam1*, which is known to play an important role in the immune response, is expressed in the CNS, particularly in astrocytes, microglial and endothelial cells in the white and gray matter of the human forebrain. *Icam1* has been attributed to the etiopathogenesis of neurodegenerative, e.g., AD and neuropsychiatric disorders, due to its functional involvement in blood–brain barrier function and as a marker for inflammation [141]. Interestingly, the *Icam1* gene product was suggested to be neuroprotective and a potential candidate in cytokine-mediated therapy of AD [141]. Studies in humans revealed that the *Icam1* gene product was increased in the cerebrospinal fluid (CSF) in preclinical, prodromal, and dementia stages of AD, associated with cortical thinning and cognitive deterioration and could act as a biomarker for early neuroinflammation and an AD risk assessment factor [142]. Although *Icam1* was previously suggested to serve as a

CSF biomarker for AD, we found *Icam1* to be significantly upregulated (FC > 1.5) only in the hippocampus of female APP/PS1 mice and it turned out to be one of the differentially expressed fingerprint genes in this subgroup.

*Tubulin beta 6 (Tubb6). Tubb6* encodes for a tubulin isotype, i.e., tubulin beta 6 class V with GTP binding activity, a constituent of the cytoskeleton being involved in microtubule and actin dynamics. Microtubule dysfunction is a prominent feature in many neurodegenerative diseases including AD [143]. *Tubb6* mRNA was previously detected in the hippocampus of C57BL/6 mice and mRNA levels were reported to be low and to further decrease in early age [144]. Proteome studies indicated an overrepresentation of tubulin beta 6 in mouse models of AD [145]. Importantly, in our transcriptome studies, *Tubb6* was only upregulated in the hippocampus of female APP/PS1 mice, not in the RS cortex and not in male APP/PS1 mice.

**Functional relevance of signature genes in the RS cortex of female APP/PS1 mice.** In the RS cortex of female APP/PS1 AD mice, 25 candidates exhibited signature character, i.e., *Gusb, Slc14a1, Mir142hg, Cyth4, Gpr34, Plek, Ccl9, Aif1, Psmb9, Lair1, Hk3, Csf2rb, Tlr12, Adora3, Apbb1ip, Pycard, Vav1, Oas1a, Ifi27l2a, Hexb, Ncf4, Fcrls, Irf5, Havcr2, Mlxipl*. These specific DEGs turned out be related to the fields of <u>inflammatory response, Tyrobp causal network in microglia, neutrophil degranulation, phagocytosis and the RAF/MAP kinase cascade</u>. Some of the related gene candidates and their implications in AD are further illuminated below:

*Pleckstrin (Plek).* A most interesting DEG selectively upregulated in the RS cortex of female APP/PS1 mice turned out to be *Plek*. Pleckstrin is an important protein in cytoskeleton reorganization and neurite outgrowth and is supposed to be of functional relevance in AD pathophysiology as the structural and functional integrity/dynamics of the cytoskeleton are a prerequisite for neuronal function and survival [146, 147]. Pleckstrin (*Plek*) turned to be a negatively regulated target of miR-409-5p, a miRNA that impairs neurite outgrowth. Interestingly, $A\beta_{1-42}$ peptides significantly decrease cortical expression of miR-409-5p, resulting in less decrease of neuronal viability, and reduction in subsequent $A\beta_{1-42}$-induced pathologies [147]. In contrast, overexpression of miR-409-5p exerts neurotoxic effects in neuronal cells and impairs differentiation. Overexpression of *Plek*, on the other hand, harbors the capability to rescue neurite outgrowth from this neurotoxicity [147]. It has previously reported that miR-409-5p is early downregulated in APP/PS1 mice and that *Plek* is upregulated at 9 and 12, but not 4 and 6 months-old APP/PS1 animals [147]. Our study reveals that *Plek* is upregulated in 8 months old animals, and according to our transcriptome results, a signature gene in the RS cortex of female APP/PS1 mice.

Strikingly, the previously reported early downregulation of miR-409-5p in combination with the observed upregulation of *Plek* in AD progression might thus point to a counteracting, compensatory and autoprotective reaction to alleviate the structural and functional synaptic impairment induced by Aβ. *Plek* and its gene product Pleckstrin could thus serve as potential early biomarker in AD.

*Adenosine receptor type 3 (Adora 3).* Some upregulated genes, such as *Adora 3*, are not only involved in microglia activation but also neutrophil degranulation and leucocyte migration/chemotaxis. *Adora3* encodes a G-protein coupled adenosine receptor type 3, which is supposed to mediate inhibition of neutrophil degranulation and therefore potentially prevents cell damage. However, it was also reported to exhibit ambiguous effects in neuroprotection and neurodegeneration [108, 148, 149] stressing that modulation of the adenosinergic system in AD, in terms of pathology and therapeutics, is a sophisticated issue [150]. Despite this potential intractability and the lack of medicines targeting adenosinergic receptors for AD treatment, substantial efforts are made in drug research and development. Importantly, *Adora3* turned out to be a unique fingerprint candidate for the RS cortex of female APP/PS1 mice in our study. This sex-

specific and potentially neuroprotective profile of *Adora3* warrants future attention also in drug development and potential individualized pharmaceutical treatment in AD.

*Vav guanine nucleotide exchange factor 1 (Vav1).* *Vav1* encodes a guanine nucleotide exchange factor (GEF) for Rho family GTPases that activate pathways leading to actin cytoskeletal rearrangements and transcriptional alterations. It serves as a functional regulator of *Rac2*. It acts upstream of or within several processes, including integrin-mediated signaling pathway and neutrophil chemotaxis [151]. *Vav1* also seems to promote learning by activating HIF-1 and GLUT-1 and thereby contributing to glucose distribution to the brain, another important factor in AD [152]. Interestingly, *Vav1* turned out to be a unique fingerprint DEG for the RS cortex of female APP/PS1 mice.

*Apoptosis inducing factor (Aif).* The apoptosis inducing factor (*Aif*) is responsible for caspase-independent programmed cell death [153]. *Aif* was reported to mediate cell death in the TgCRND8 AD mouse model in a region-specific manner with aging-related cell death in the cortex but not the hippocampus [153]. Notably, this holds true for our findings in APP/PS1 AD mice as well. Indeed, *Aif* turned out to be a unique fingerprint gene in the cortex of APP/PS1 mice, however, only in females and not in males–a phenomenon that has not been specified before [153].

*Hexosaminidase subunit beta (Hexb).* The gene product of *Hexb* (hexosaminidase subunit beta) has been associated with AD based on genome-wide association studies (GWAS) [154]. In addition, β-hexosaminidase can lead to a reduction of Aβ complexes, the aggregation and accumulation of which has been accelerated by exposure to gangliosides [155]. Importantly, *Hexb* was unique for the RS cortex of female APP/PS1 mice.

*Proteasome 20S subunit beta 9 (Psmb9).* The gene products of *Psmb9* and *Psmb8* (encoding for proteasome 20S subunit beta 9 and 8 respectively) form part of a proteasome (prosome, macropain) core complex which plays an important role in AD [156]. The ubiquitin-proteasome system (UPS) is a major regulator of protein homeostasis. In contrast, the immunoproteasomes constitute a specialized form of proteasomes, which modulate inflammatory processes in AD via clearance of oxidant-damaged proteins. In APP/PS1 mice, Aβ-deposition was reported to parallel upregulation of immunoproteasomes. The latter turned out to be upregulated by pro-inflammatory cytokines, e.g., type I and type II interferons (IFNs). Wagner et al. (2017) suggested that immunoproteasomes are upregulated in the innate immune response towards extracellular Aβ-accumulation, although previous rodent studies revealed inconsistent data with decreased, unchanged, and increased proteasomal activity with aging [156–158]. Wagner et al. (2017) used both males and females. However, they did not analyze for sex-specific differences [156]. Strikingly, *Psmb8* was upregulated only in the RS cortex and hippocampus of females and not in males in our study. In contrast, *Psmb9* evolved as a unique fingerprint gene for the RS cortex of female APP/PS1 mice. The potential neuroprotective upregulation of *Psmb9* thus seems to be attributable to the predilected brain region and sex in AD.

*PYD and CARD domain containing gene (Pycard).* *Pycard* (PYD and CARD domain containing gene, encoding for adapter protein apoptosis-associated speck-like protein containing a CARD (ASC)) codes for an inflammasome component and is involved in regulation of autophagy. Previous reports demonstrated that released ASC specks can bind to Aβ, enhance its aggregation, and increase its toxicity [159]. In our study, *Pycard* turned out to be a unique fingerprint for the RS cortex of female APP/PS1 mice. As *Pycard* was recently suggested to be a promising diagnostic target in early AD patients [160], sex-specific characteristics require further attention.

*G protein-coupled receptor 34 (Gpr34).* The G-protein-coupled receptor 34 (*Gpr34*) was found to be highly expressed in the hippocampus of APP/PS1 mice [161] and to be involved in microglia phagocytic activity, complement activation and synaptic pruning [174]. *In vitro* and *in vivo*

Gpr34 knockdown approaches resulted in decreased levels of TNF-α, IL-1β, IL-6 and iNOS and suppression of ERK/NF-κB signal activation. Similarly, *Gpr34*-overexpression resulted in activation of ERK/NF-κB signalling and upregulation of TNF-α, IL-1β, IL-6 and iNOS. Systemically, *Gpr34* knockdown relieved cognitive deficits in APP/PS1 mice and limited neuroinflammation and microglial activation, most probably via the ERK/NF-κB pathway [161].

**Functional implications of signature genes in the hippocampus and RS cortex of male APP/PS1 mice.** In the hippocampus of male APP/PS1 AD mice, exclusive upregulation was observed in 18 gene candidates, i.e., *Sgk1*, *Ncf1*, *Adamtsl4*, *Hck*, *St8sia6*, *Gm20743*, *Irf9*, *Pfkfb3*, *B2m*, *Arrdc2*, *Cebpa*, *Synpo2*, *Cep126*, *Rhoh*, *Hist1h2be*, *Clec5a*, *Cd86*, *Ptpn18*. Some are discussed below in more detail:

*Hematopoietic cell kinase (Hck).* The hematopoietic cell kinase encoded by *HcK* serves as a factor of the innate immune system. It has previously been demonstrated that the *Hck* pathway exerts important functions in the regulation of microglial neuroprotective functions during the early stage of AD [162]. Strikingly, hematopoietic cell kinase deficiency aggravated cognitive impairment along with elevated Aβ levels and plaque formation. It further attenuated Aβ phagocytosis and enhanced iNOS expression in microglia [163]. Thus, *Hck* is likely to exert a prominent neuroprotective function via modulation of microglial function and could attenuate early AD development.

*Serum- and glucocorticoid-inducible kinase 1 (Sgk1).* Serum- and glucocorticoid-regulated kinase 1 (encoded by *Sgk1*) is a serine/threonine protein kinase that activates certain $K^+$, $Na^+$, and $Cl^-$ channels, involved in cell survival and neuronal excitability [164]. SGK1 is engaged in various neurodegenerative pathways related to, e.g., apoptosis, autophagy, neuroinflammation, and ion channel regulation. Interestingly, hippocampal overexpression of *Sgk1* improved spatial memory, reduced the devastating impact of Aβ accumulation and rescued actin cytoskeleton polymerization in middle-aged APP/PS1 mice [164, 165]. It has been speculated that Sgk1 could exert a protective role against oxidative stress and play an antiapoptotic role [166]. Whether pharmacological stimulation of serum- and glucocorticoid-regulated kinase 1 is protective in AD as well needs to be determined in the future.

*Synaptopodin (Synpo2).* Synaptopodin serves as an actin-binding protein that is tightly associated with the spine apparatus and plays an important role in synaptic plasticity [167]. Synaptopodin forms clusters in spines and regulates $Ca^{2+}$ release from internal stores via ryanodin receptors in dendritic spines [167, 168]. Deficiency of synaptopodin resulted in LTP deficits, impaired spatial memory and a lack of synaptic plasticity [167, 169, 170]. Interestingly, synaptopodin expression was reported to be downregulated in the hippocampus of patients suffering from dementia with Lewy bodies, Parkinson's disease, mild cognitive impairments, and AD [170, 171]. However, we observed a selective upregulation of synaptopodin in the hippocampus of male APP/PS1 mice. The later might point to a potential autoprotective role of *Synpo2* in this experimental setting.

It should be emphasized that impairment of the structural and functional integrity of synapses is a hallmark in AD pathophysiology. Aβ causes disruption of NMDA and AMPA receptors, $Ca^{2+}$ dyshomeostasis, reduced synaptic plasticity, suppression of LTP and aggravated LTD. Microglia/astrocyte activation, cytokine release, mitochondrial disruption, impairment of the cytoskeleton organization and deficits in energy metabolism further enhance synaptic dysfunction [120]. Mechanistically, a number of synapse-related target molecules and signaling pathways are related to these phenomena, e.g., Wnt/β-catenin, IKK/NF-κB, JAK2/STAT3, JNK, Akt, MAPK, caspase-3, GSK-3β and CDK-5 [120]. Many DEGs identified in our study (see above) enhance synaptic dysfunction, e.g. PrPC, whereas others, such as synaptopodin might exert synaptoprotective effects.

Interestingly, only two candidates, *Derl3*, and *Mki67*, exhibited unique upregulation in the RS cortex of male APP/PS1 AD animals (**Fig 5E**).

*Derlin (Derl3)*. Derlin 3 (encoded by Derl3) resides in the endoplasmic reticulum (ER) and is involved in the degradation of mis-folded glycoproteins in the ER. Dysfunction of Derlin 1 and 2 have previously been related to neurodegenerative diseases [75].

*Ki-67 (Mki67)*. Mki67 encoding for Ki-67 serves as cell proliferation marker and is maximally expressed in G2 phase and mitosis. Interestingly, previous studies identified MKi67 (and also *Top2a*, *Mcm2*, *Tubb5*) to be enriched in DNA replication and chromatin rearrangement in a specific subgroup of microglial cells, i.e., cycling/proliferating microglia (CPMs), that contributes only 0.3%–1.2% of the total microglial population [76]. It has further been suggested that Ki-67 is involved in the pathogenesis of neurofibrillary degeneration in AD [172].

As outlined previously, the number of downregulated genes in the individual subgroups turned out to be significantly lower compared to the upregulated ones (**Fig 6**). However, it was possible to characterize unique profiles of downregulated genes in the individual subgroups. In the hippocampus of female APP/PS1 AD mice, these signature genes included 13 candidates (*Asic4*, *Pcsk1*, *Etnppl*, *Myh7b*, *Spag5*, *Gpr1*, *Myh8*, *Ccdc184*, *Pdzph1*, *Trim66*, *BC030499*, *Rims3*, *Prr16*), only two candidates (*LOC105247294*, *Ptpn6*) in the RS cortex of female APP/PS1 AD mice, six gene candidates in the hippocampus of male APP/PS1 AD mice (*4933407I18Rik*, *Hlcs*, *Hdac9*, *Pla2g4e*, *Wnt9a*. *Shisa9*) and four candidates in the RS cortex of male APP/PS1 AD animals (*6530402F18Rik*, *Ctr9*, *Scyl2*, *1700001L05Rik*) (**Fig 6E**).

Notably, the number of intersectional, upregulated genes was much higher in the RS cortex of APP/PS1 mice compared to the hippocampus (52 versus 32, **Fig 5C and 5D**). Many of these additional, intersectional candidates in the RS cortex (**Fig 5C**) are recruited from gene candidates that turned out to be selectively expressed in the male or female hippocampus (**Fig 5D**). Consequently, more gene candidates are specifically upregulated in the female hippocampus compared to the RS cortex (45 vs. 39) and the male hippocampus compared to the RS cortex (25 vs. 5). Thus, sex-specific differences in DEGs are more overt in the hippocampus than in the RS cortex.

It turned out that several DEGs are region-specific as well. It has been shown for example, that microglia in different brain regions can differ in gene expression, particularly in genes related to bioenergetic and immunoregulatory pathways [173]. Interestingly, the number of co-upregulated DEGs in the RS cortex and hippocampus of females was much higher than the related number in male APP/PS1 mice (55 vs. 33, **Fig 5**).

These findings are supported by symptomatic sex dimorphism profiling in 12 months old male and female APP/PS1 AD mice performed by Jiao et al. (2016) [139]. Indeed, there is increasing evidence, that sex-biased and BROI-specific patterns may originate from differences in vulnerabilities/susceptibilities and/or resilience in different brain regions at different AD progression states. Female APP/PS1 mice exhibited a higher parenchymal Aβ load compared to male APP/PS1 animals. The latter was most prominent in the hippocampus [139]. In addition, cerebral amyloid angiopathy and related microhemorrhage was more frequent in female APP/PS1 mice. Although top-ranked, intersectional genes associated with neuroinflammation, microglia activation and immune response are found in both sexes and both the RS cortex and hippocampus, specific gene sets exert fingerprint character in the individual subgroups. Notably, related parameters, including levels of phosphorylated τ, proinflammatory cytokines, microgliosis, astrocytosis, and synaptic/neuronal disintegration were particularly enhanced in female APP/PS1 animals [139]. Some DEGs that predominate in females are related to mitochondria and ROS. In females, mitochondrial function seems to be more resistant against Aβ mediated neurotoxicity. This phenomenon is probably due to a reduction of ROS and suppression of apoptogenic signals via estrogen [174]. This may be one reason why

females suffer more from AD with decreasing estrogen levels following menopause. Brain-region und sex-related specificities in DEGs, as observed in our study, might also be due to differences in brain region connectivity/architecture between males and females. The latter suggests that network characteristics in females favor a more rapid spread of neurofibrillary tangles in the CNS [175]. In addition, different BROIs are disproportionally modulated via complex spatiotemporal activity of sex hormones [176–178].

In summary, the determination of sex- and brain region-specific transcriptional profiles as presented here, is highly necessary for future characterization and validation of DEGs as potential biomarkers and personalized medicinal approaches.

**Disease aggravating versus neuroprotective DEGs in APP/PS1 mice.** Our transcriptome studies in APP/PS1 AD mice have revealed a high number of significantly upregulated and downregulated gene candidates. Whereas many of them turned out to be intersectional, some exhibited signature characteristics. It's interesting to note that not all DEGs contribute to the devastating complex pathophysiology and pathobiochemistry of AD. Indeed, some candidates, such as *Plek*, *Adora3* and *Psmb9* in the female RS cortex, *Icam1* in the female hippocampus, and *Hck* and *Sgk1* in the male hippocampus of APP/PS1 mice seem to exert counteracting, autoprotective functions in the individual subgroups. Some intersectional genes such as, *Ccl6*, *Lyz2* and *Nrf2* also exhibit self-protective effects. They clearly deserve special attention in drug research and development in the future.

## L(i)ncRNAs in APP/PS1 mice

In the mammalian genome, tens of thousands of l(i)ncRNAs were identified, with up to 40% of these l(i)ncRNAs being specifically expressed in the brain [179, 180]. These l(i)ncRNAs without apparent protein-coding influence metabolism, proliferation, differentiation, and survival of neuronal cells [180]. L(i)ncRNA expression has been associated with many neurodegenerative diseases [181–183] in patients or animal models and therefore, could serve as biomarkers or potential treatment targets, e.g., in AD. There is evidence that l(i)ncRNAs are aberrantly expressed in a complex spatiotemporal pattern during AD disease progression. L(i)ncRNAs are involved in the regulation of Aβ production, τ hyperphosphorylation, oxidative stress, mitochondrial dysfunction, impairment of synaptic transmission and neuroinflammation, cell death etc. and are likely to affect clinical diagnosis, disease monitoring and therapy [57]. These regulatory actions of l(i)ncRNAs are mediated on the transcriptional and posttranscriptional levels. L(i)ncRNAs can affect many cellular processes, including chromatin and DNA modification, RNA transcription, pre-RNA splicing, mRNA stability, and translation [180, 184, 185].

Several hundred differentially expressed l(i)ncRNAs were detected in 3xTg-AD model mice, compared to controls [186]. In the hippocampus of APP/PS1 mice, 99 downregulated l(i)ncRNAs and 150 upregulated l(i)ncRNAs were observed compared to WT mice [40].

The SurePrint G3 Mouse Gene Expression v2 8x60K Microarray which we used in our study is not specifically designated to analyze l(i)ncRNAs. We identified a smaller number of l(i)ncRNAs in our individual experimental subgroups, but we were able to characterize, e.g., two l(i)ncRNAs differentially transcribed in the hippocampus of male APP/PS1 mice (l(i)ncRNA: chr18:38776580–38841080 reverse strand) and the hippocampus of female APP/PS1 animals (l(i)ncRNA: chr8:122920901–123008463 forward strand). This sex-specificity in l(i)ncRNA transcription has not been reported before [40]. Our transcriptome study is the first to reveal sex- and region-specific l(i)ncRNAs in the individual subgroups. Although these l(i)ncRNA candidates are annotated in related databases (**S13 Table in S1 File**), further studies are necessary to elucidate and validate their detailed functional implications in AD.

**Social hierarchy and potential inference with cortical and hippocampal transcriptome data.** In the interpretation of transcriptome data, the functional interdependence with social hierarchy is often not addressed, although this aspect must be taken into account as a potential confounding factor. It has been demonstrated that behaviors and brain gene expression in WT, e.g., C57BL/6 male mice, is severely affected by different social network sizes and hierarchy in the home cage [187, 188].

Alterations in gene expression mainly affected the serotonergic system in dominant and subordinate mice. In addition, subordinate mice exhibited significantly higher corticosterone concentration than dominant males revealing increased stress in subordinate males. Increased chronic stress can lead to downregulation of *Bdnf* and increased expression of the BDNF receptor gene, *Trkb*, in the hippocampus [188]. Interestingly, these candidates were not detected as DEGs in our transcriptome studies, neither in females nor in males, which might indicate only limited interference of our results with social hierarchy phenomena.

## Influence of APP overproduction and transcriptome profiles

Overexpression of WT APP and mutant APP variants has been used to establish many well-characterized transgenic AD mouse lines, including APP/PS1 [189]. It's noteworthy that APP overexpression in these first-generation transgenic mouse models can cause overproduction of different APP fragments in addition to Aβ. The latter can make it challenging to distinguish the functional impact of these fragments and might affect the translational relevance of such models [190, 191]. Potentials limitations of mutant APP- and APP/PS-overexpressing mouse models, such as APP/PS1 include, e.g., the lack of non-coding regions of the APP gene that affect splicing of APP mRNA and transcriptional regulation, unphysiological interaction of overexpressed APP and overproduced non Aβ fragments with cellular proteins such as kinesin via JIP-1, non-specific ER impairment upon APP/PS-overexpression, appearance of Aβ entities that are different from those identified in clinical AD brain, different region specificity of Aβ pathology, and inconsistent drug effects [192]. Notably, second-generation mouse models are likely to overcome these intrinsic limitations of the APP overexpression paradigm as they utilize an APP knock-in strategy in order to selectively overexpress, e.g., Aβ$_{42}$, but not APP. The latter was achieved, e.g., in single APP knock-in mouse models with a murine Aβ sequence carrying three humanized amino acids that differ between mice and humans [192]. However, these models are facing numerous limitations as well [192]. Thus, it's important to be aware of the potential interference of APP- or APP/PS-overexpression with AD disease pathology in first generation AD mouse models [192].

## Conclusions

Our APP/PS1 AD model-based data support the hypothesis that a major response in the brain to Aβ accumulation is related to microglia activation, immune response and neuroinflammation. Early and profound neuroinflammatory/immune response seems to be a fundamental driving force in AD pathogenesis. Many DEGs were claimed to be involved in these processes, however, sex- and region-specific differentiation of their functional relevance is rare. Our data clearly illustrate that sex-specific differences in the transcriptome profiles are of major relevance and that some DEGs, i.e., signature genes, are exclusively upregulated or downregulated in males or females (**Tables 3 and 4**). So far, numerous defective genes, transcriptional and translational alterations were shown to contribute to AD pathogenesis, further influenced by diverse environmental factors and epigenetic phenomena. The consideration of sex, age, comorbidity factors, disease progression state and most important, the individual genomic profile, are mandatory for a valid, promising, personalized pharmacotherapeutic approach in

the future. The latter will require distinct protocols and strategies in drug research and development, the planning of clinical trials to optimize therapeutics and the establishment of new diagnostic approaches. The analysis of the patient's individual genome including gene sets associated with disease pathogenesis, mechanisms of drug action, drug metabolism, drug transporters and multifaceted cascades and metabolic pathways, will dramatically facilitate the personalized therapeutic approach in AD.

## Supporting information

**S1 File.** **S1 Fig:** Genotyping of APP/PS1 AD mice and WT control animals. **S2 Fig:** 3D image of the murine brain including the RS cortex and hippocampus (BROIs) used for transcriptome analysis in our study. **S3 Fig:** PCA of transcriptomes from the RS cortex and hippocampus of WT controls and APP/PS1 AD mice of both sexes. **S4 Fig:** Hierarchical clustering of transcriptome data from the RS cortex and hippocampus of WT control and APP/PS1 AD mice of both sexes. **S5 Fig:** Bar diagrams of the top 30 candidates of DEGs with highest significant FCs (FC > 1.5 and FC < -1.5, p < 0.05). **S6 Fig:** Pathway analysis of intersectional and signature gene sets in APP/PS1 subgroups. **S7 Fig:** Comparative qPCR analysis of selected gene transcript levels from the hippocampus of female and male APP/PS1 AD with 5XFAD mice. **S1 Table:** PCR reaction set-up using PCR Mastermix and genomic DNA. **S2 Table:** Materials used for one-color microarray-based gene expression data collection. **S3 Table:** Software used for one-color microarray-based gene expression data collection. **S4 Table:** Details on genes, forward and reverse primer sequences and annealing temperatures relevant for qPCR experimentation. **S5 Table:** Characteristics of DEGs in the RS cortex of female APP/PS1 AD mice. **S6 Table:** Characteristics of DEGs in the hippocampus of female APP/PS1 AD mice. **S7 Table:** Characteristics of DEGs in the RS cortex of male APP/PS1 AD mice. **S8 Table:** Characteristics of DEGs in the hippocampus of male APP/PS1 AD mice. **S9 Table:** Venn analysis of DEGs in the RS cortex and hippocampus of female APP/PS1 AD mice. **S10 Table:** Venn analysis of DEGs genes in the RS cortex and hippocampus of male APP/PS1 AD mice. **S11 Table:** Venn analysis of DEGs in the RS cortex of male and female APP/PS1 AD mice. **S12 Table:** Venn analysis of DEGs in the hippocampus of male and female APP/PS1 AD mice. **S13 Table:** Differentially regulated l(i)ncRNAs in APP/PS1 AD vs. WT mice. **S14 Table:** qPCR-based FC analysis of selected genes in the hippocampus of APP/PS1 AD vs. WT control mice. **S15 Table:** Design of transcriptome studies carried out in APP/PS1 AD mice. **S16 Table:** Functional implications of DEGs in AD.
(ZIP)

## Acknowledgments

The authors are grateful to Dr. Christina Kolb (German Center for Neurodegenerative Diseases, DZNE) and Dr. Robert Stark (DZNE) for assistance in animal breeding and animal health care.

## Author Contributions

**Conceptualization:** Anna Papazoglou, Christina Henseler, Sandra Weickhardt, Johanna Daubner, Teresa Schiffer, Agapios Sachinidis, Marco Weiergräber.

**Data curation:** Panagiota Papazoglou, Marco Weiergräber.

**Formal analysis:** Anna Papazoglou, Christina Henseler, Damian Krings, Marco Weiergräber.

**Funding acquisition:** Marco Weiergräber.

**Investigation:** Anna Papazoglou, Christina Henseler, Sandra Weickhardt, Jenni Teipelke, Panagiota Papazoglou, Teresa Schiffer, Marco Weiergräber.

**Methodology:** Anna Papazoglou, Christina Henseler, Sandra Weickhardt, Jenni Teipelke, Panagiota Papazoglou, Agapios Sachinidis, Marco Weiergräber.

**Project administration:** Marco Weiergräber.

**Resources:** Marco Weiergräber.

**Software:** Anna Papazoglou, Christina Henseler, Sandra Weickhardt, Panagiota Papazoglou, Agapios Sachinidis, Marco Weiergräber.

**Supervision:** Anna Papazoglou, Christina Henseler, Johanna Daubner, Marco Weiergräber.

**Validation:** Anna Papazoglou, Christina Henseler, Marco Weiergräber.

**Visualization:** Anna Papazoglou, Christina Henseler, Teresa Schiffer, Marco Weiergräber.

**Writing – original draft:** Marco Weiergräber.

**Writing – review & editing:** Anna Papazoglou, Christina Henseler, Sandra Weickhardt, Jenni Teipelke, Panagiota Papazoglou, Johanna Daubner, Teresa Schiffer, Damian Krings, Karl Broich, Jürgen Hescheler, Agapios Sachinidis, Dan Ehninger, Catharina Scholl, Britta Haenisch, Marco Weiergräber.

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
