## [Decision Letter · Decision Letter 0]

17 Oct 2023

PONE-D-23-30186Sex- and region-specific cortical and hippocampal whole genome transcriptome profiles from control and APP/PS1 Alzheimer’s disease mice.PLOS ONE

Dear Dr. Weiergräber,

Thank you for submitting your manuscript to PLOS ONE. After careful consideration by 2 Reviewers and an Academic Editor, all of the critiques of both Reviewers must be addressed in detail in a revision to determine publication status. If you are prepared to undertake the work required, I would be pleased to reconsider my decision, but revision of the original submission without directly addressing the critiques of the Reviewers does not guarantee acceptance for publication in PLOS ONE. If the authors do not feel that the queries can be addressed, please consider submitting to another publication medium. A revised submission will be sent out for re-review. The authors are urged to have the manuscript given a hard copyedit for syntax and grammar.

Please submit your revised manuscript by Dec 01 2023 11:59PM. If you will need more time than this to complete your revisions, please reply to this message or contact the journal office at plosone@plos.org. Please include the following items when submitting your revised manuscript:A rebuttal letter that responds to each point raised by the academic editor and reviewer(s). You should upload this letter as a separate file labeled 'Response to Reviewers'.A marked-up copy of your manuscript that highlights changes made to the original version. You should upload this as a separate file labeled 'Revised Manuscript with Track Changes'.An unmarked version of your revised paper without tracked changes. You should upload this as a separate file labeled 'Manuscript'.

We look forward to receiving your revised manuscript.

Kind regards,

Stephen D. Ginsberg, Ph.D.

Section Editor

PLOS ONE

3.Thank you for stating the following financial disclosure: "The study was internally funded by the Federal Institute for Drugs and Medical Devices (Bundesinstitut für Arzneimittel und Medizinprodukte, BfArM, Bonn, Germany)"

4. We noted in your submission details that a portion of your manuscript may have been presented or published elsewhere. "For transparency reasons, the raw, non-analyzed transcriptome data were uploaded to the Mendeley Data repository and described in Data in Brief. The data description in Data in Brief was uploaded as a copy. We are confident that this approach is most important to maximize data transparency and enable to readers to reperform our analyses or carry out further studies themselves in the future."

5. Please note that in order to use the direct billing option the corresponding author must be affiliated with the chosen institute. Please either amend your manuscript to change the affiliation or corresponding author, or email us at plosone@plos.org with a request to remove this option.

6. Please amend either the title on the online submission form (via Edit Submission) or the title in the manuscript so that they are identical.

**Comments to the Author**

1. Is the manuscript technically sound, and do the data support the conclusions?

Reviewer #1: Yes

Reviewer #2: Yes

2. Has the statistical analysis been performed appropriately and rigorously? 

Reviewer #1: Yes

Reviewer #2: Yes

3. Have the authors made all data underlying the findings in their manuscript fully available?

Reviewer #1: Yes

Reviewer #2: Yes

4. Is the manuscript presented in an intelligible fashion and written in standard English?

Reviewer #1: Yes

Reviewer #2: Yes

5. Review Comments to the Author

Reviewer #1: The authors of this manuscript has performed microarray studies in two specific regions of the brain in WT and APP/PS1 mouse model of Alzheimer’s Disease. Overall, changes were detected in the brain regions with unique and overlapping DEGs. As expected, many of the categories of genes that we altered belong to previously characterized pathways (e.g. inflammatory, phagocytosis). Nevertheless, the authors highlighted several unique genes and gene pathways that may be attributed to region and sex-specific differences in AD. However, there are no follow ups to these observations, or additional analysis or verification of these identified target genes. As such, much of what is novel in the manuscript is not adequately highlighted or discussed. Please see attached reviewer concerns for the manuscript.

Reviewer #2: In this article entitled “Sex- and region-specific cortical and hippocampal whole genome transcriptome profiles from control and APP/PS1 Alzheimer’s disease mice”, Papazoglou et al have showed differentially expressed gene signatures from retrosplenial cortex and hippocampus of age-matched male and female APP/PS1 AD mice and control animals. They found genes for activation of microglial, astrocytic and neutrophilic cells, innate immune response/immune effector response and neuroinflammation, phagosome and proteasome activation, and synaptic transmission were upregulated, while the genes for synaptic vesicle docking/fusion machinery, synaptic transmission, rRNA processing, ubiquitination and proteasome degradation, histone modification and cellular senescence were downregulated.

Although sex dependent transcriptome profiling is not very novel in hippocampus, and there are a few studies which have done transcriptome profiling in a sex dependent manner. Although the study is not completely novel it is interesting in terms of the systematic profiling of both regions cortex and hippocampus in a sex dependent fashion.

Please address these concerns

1) Line 111 suggest Sudden death in APP/PS1 mice peaks around 3–4 months of age, This is not commonly seen at 3-4 months there are many studies that reported is the time when the synaptic deficits starts to appear.

2) Please discuss the impact of the study based on an region and sex specific changes in transcriptome on the pathogenesis and prevention.

3) The role of immune mechanisms in the etiology of AD is well studied, so discuss in terms of comparisons of region and sex.

4) Are these upregulated genes pathogenic or is there any neuroprotective genes?

5) In synaptic plasticity upregulation of certain genes can lead to increase in synaptic transmission, while upregulation of some genes can lead to downregulation of synaptic plasticity. Please discuss.

6) APP/PS1 is a model where pathology also involves APP over production, so how would you negate its effects?

7) As some DEGs are exclusively upregulated or downregulated in males or females, this could be a target for gender specific medicine.Please discuss.

6. PLOS authors have the option to publish the peer review history of their article (what does this mean?). If published, this will include your full peer review and any attached files.

**Do you want your identity to be public for this peer review?** For information about this choice, including consent withdrawal, please see our Privacy Policy.

Reviewer #1: No

Reviewer #2: No

---

## [Author Response · Author response to Decision Letter 0]

14 Dec 2023

Response to Reviewers

General points:

General point 1: The authors are urged to have the manuscript given a hard copyedit for syntax and grammar.

Response: We thank the editor for this comment. We have carefully revised the manuscript once again for syntax and grammar.

General point 2: If applicable, we recommend that you deposit your laboratory protocols in protocols.io to enhance the reproducibility of your results. Protocols.io assigns your protocol its own identifier (DOI) so that it can be cited independently in the future. For instructions see: https://journals.plos.org/plosone/s/submission-guidelines#loc-laboratory-protocols. Additionally, PLOS ONE offers an option for publishing peer-reviewed Lab Protocol articles, which describe protocols hosted on protocols.io. Read more information on sharing protocols at https://plos.org/protocols?utm_medium=editorial-email&utm_source=authorletters&utm_campaign=protocols.

Response: We thank the editor for this comment. Concerning our experimental protocols, the Material and Methods section provides all information to reperform our experiments. We would also like to point out that the raw data (Agilent readout txt.-files) were uploaded to Mendeley Data, a public and freely available data repository (see below).

Journal requirement 1: 1. Please ensure that your manuscript meets PLOS ONE's style requirements, including those for file naming. The PLOS ONE style templates can be found at https://journals.plos.org/plosone/s/file?id=wjVg/PLOSOne_formatting_sample_main_body.pdf and 

Response: The revised manuscript meets the PLOS One style as requested by the editor.

Journal requirement 2: Did you know that depositing data in a repository is associated with up to a 25% citation advantage (https://doi.org/10.1371/journal.pone.0230416)? If you’ve not already done so, consider depositing your raw data in a repository to ensure your work is read, appreciated and cited by the largest possible audience. You’ll also earn an Accessible Data icon on your published paper if you deposit your data in any participating repository (https://plos.org/open-science/open-data/#accessible-data).

Response: We very much appreciate this recommendation. We are glad to inform you that we have uploaded and published the raw readout files of the SurePrint microarrays used in our study in the Mendeley database, a public repository with free access to the scientific audience. I would like to stress that we only uploaded the raw microarray readout files (as txt.- and csv.-files) and no additional processed or analyzed data.

The raw data can be freely accessed via: Weiergräber, Marco (2023), “Whole genome transcriptome data from the retrosplenial cortex and hippocampus of female and male control and APP/PS1 Alzheimer’s disease mice.”, Mendeley Data, V2, doi: 10.17632/z9264694b4.2.

These raw data are described in the Journal Data in Brief:

Papazoglou A, Henseler C, Weickhardt S, Daubner J, Schiffer T, Broich K, et al. Whole genome transcriptome data from the WT cortex and hippocampus of female and male control and APP/PS1 Alzheimer's disease mice. Data Brief. 2023;50:109594. (1)

We incorporated this information in the revised version of our manuscript:

“Please note that for transparency reasons, the raw read out data files were extracted as txt.-files/csv.-files and are freely available at the Mendeley Data repository (Weiergräber, Marco (2023), “Whole genome transcriptome data from the retrosplenial cortex and hippocampus of female and male control and APP/PS1 Alzheimer’s disease mice.”, Mendeley Data, V2, doi: 10.17632/z9264694b4.2 (59).”

Please note that only the raw data were published to guarantee maximum transparency. We would like to stress that none of the analyses presented in our manuscript submitted to PLOS One here has been published before.

Journal requirement 3: Thank you for stating the following financial disclosure: "The study was internally funded by the Federal Institute for Drugs and Medical Devices (Bundesinstitut für Arzneimittel und Medizinprodukte, BfArM, Bonn, Germany)". 

Response: We appreciate the editor’s comment. Indeed, the funders had no role in study design, data collection and analysis, decision to publish, or preparation of the manuscript. We added this information to the financial disclosure section in the revised manuscript, in the submission process and included it into the cover letter:

“Financial Disclosure Statement

The study was internally funded by the Federal Institute for Drugs and Medical Devices (Bundesinstitut für Arzneimittel und Medizinprodukte, BfArM, Bonn, Germany). The funder had no role in study design, data collection and analysis, decision to publish, or preparation of the manuscript.”

Journal requirement 4: We noted in your submission details that a portion of your manuscript may have been presented or published elsewhere. "For transparency reasons, the raw, non-analyzed transcriptome data were uploaded to the Mendeley Data repository and described in Data in Brief. The data description in Data in Brief was uploaded as a copy. We are confident that this approach is most important to maximize data transparency and enable to readers to reperform our analyses or carry out further studies themselves in the future."

Response: It is very important for us to stress that none of the information which has been uploaded to the Mendeley database and described in the related peer-reviewed Data in Brief article is part of our current submission to PLOS One. Our submission to PLOS One contains all the complex analyses presented, whereas the upload to the Medeley database and the related description of the raw data files in Data in Brief is soley related to the raw Agilent readout data files (as txt.- and csv.-files). This is verifiably not a dual publication/submission.

As request in Point 2 (see above), the original raw readout files were uploaded to the Mendeley database as a public data repository. These raw readout files were published in Data in Brief (1). We have published these raw data to enable maximum transparency. In addition, this allows every reader to either reperform our analyses our to carry out further user-specific analysis if desired.

We incorporated this information in the revised version of our manuscript:

“Please note that for transparency reasons, the raw data files were extracted as txt.-files/csv.-files and are, in addition, freely available at the Mendeley database/repository (Weiergräber, Marco (2023), “Whole genome transcriptome data from the retrosplenial cortex and hippocampus of female and male control and APP/PS1 Alzheimer’s disease mice.”, Mendeley Data, V2, doi: 10.17632/z9264694b4.2 (1).”

We have included the related information in the cover letter of our revised manuscript as well.

Journal requirement 5: Please note that in order to use the direct billing option the corresponding author must be affiliated with the chosen institute. Please either amend your manuscript to change the affiliation or corresponding author, or email us at plosone@plos.org with a request to remove this option.

Response: We apologize for the mistake and made the necessary amendment.

Point 6: Please amend either the title on the online submission form (via Edit Submission) or the title in the manuscript so that they are identical.

Response: We apologize for the mistake. We have harmonized the title of our manuscript in the online submission form and the revised manuscript.

 

Comments to the Author

1. Is the manuscript technically sound, and do the data support the conclusions?

Reviewer #1: Yes

Reviewer #2: Yes

Response: We thank the reviewers for their positive comment.

2. Has the statistical analysis been performed appropriately and rigorously?

Reviewer #1: Yes

Reviewer #2: Yes

Response: We thank the reviewers for their positive comment.

3. Have the authors made all data underlying the findings in their manuscript fully available?

Reviewer #1: Yes

Reviewer #2: Yes

Response: We thank the reviewers for their positive comments.

4. Is the manuscript presented in an intelligible fashion and written in standard English?

Reviewer #1: Yes

Reviewer #2: Yes

Response: We thank the reviewers for their positive comments.

5. Review Comments to the Author

Reviewer #1: The authors of this manuscript has performed microarray studies in two specific regions of the brain in WT and APP/PS1 mouse model of Alzheimer’s Disease. Overall, changes were detected in the brain regions with unique and overlapping DEGs. As expected, many of the categories of genes that we altered belong to previously characterized pathways (e.g. inflammatory, phagocytosis). Nevertheless, the authors highlighted several unique genes and gene pathways that may be attributed to region and sex-specific differences in AD. However, there are no follow ups to these observations, or additional analysis or verification of these identified target genes. As such, much of what is novel in the manuscript is not adequately highlighted or discussed. Please see attached reviewer concerns for the manuscript.

Response: We thank the reviewer for these comments. Please see our detailed response below in the point-by-point rebuttal.

6. PLOS authors have the option to publish the peer review history of their article (what does this mean?). If published, this will include your full peer review and any attached files.

If you choose “no”, your identity will remain anonymous, but your review may still be made public.

Do you want your identity to be public for this peer review? For information about this choice, including consent withdrawal, please see our Privacy Policy.

Reviewer #1: No

Reviewer #2: No

Response: The authors fully agree that the peer review history is published.

Response: We followed this recommendation and uploaded / checked all figures to PACE to ensure that they meet the PLOS requirements.

 

Response to reviewers

We thank the editor and the reviewers for the thorough appraisal of our work. We would like to respond to the reviewers’ comments as follows:

Reviewer 1:

Summary

The authors of this manuscript have performed bulk microarray studies in two specific regions of the brain in WT and APP/PS1 mouse model of Alzheimer’s Disease. Overall, DEGs and GO analysis indicate upregulation of different functional responses that largely belong to the inflammatory pathways. Nevertheless, the authors highlight several unique genes and gene pathways that may be attributed to region and sex-specific differences. Some of the findings are unique and interesting and should further be highlighted in the results and discussion.

Response: We thank the reviewer for the thorough appraisal of our work. Indeed, we have acquired a mass of microarray data that have been analyzed in our study. As outlined by the reviewer, the sex- and brain regions-specific signature genes identified in our study represent one of our major outcomes. These unique gene candidates could be groundbreaking for future gender- and personalized therapeutic approaches in Alzheimer’s disease in humans. As elaborated below, we further highlighted these most important aspects in the revised Results and Discussion section.

Point 1: Please note that I am unable to open the supplementary files as they are corrupted.

Response: We regret that the reviewer was unable to open the supplementary files. Given the fact that our files passed the editorial check and that we did not receive information about problems with opening of the supplementary files from the editor and the other reviewer, we regret that we cannot comment on the reasons. 

However, we have summarized, zip-compressed and uploaded the supplementary files once again. Please note that the supplementary files needed to be zip-compressed due to the large amount of data (data size).

Point 2: Figure 1A mislabeled axes typo.

Response: We thank the reviewer for this important comment, and we apologize for the mistake in labelling. The Y-axis was corrected to “# DEGs (one-way ANOVA-Tukey-Benjamini-Hochberg)” (see below).

Point 3: Figure 2 volcano plots axes should not be compressed at the top with individual DEGs labeled in the graph. At least the top targets. Otherwise, the graphs are not informative. Some of the labels are truncated and not assembled properly.

Response: We thank the reviewer for this helpful comment. Indeed, Volcano plots provide interesting information about the distribution of DEGs, their fold changes and significances in different subgroups. Our microarray data were obtained using the Agilent platform/system. The related analysis software (GeneSpring) uses an automatic Y-axis cut-off. We contacted the GeneSpring customer support and learned that it is not possible to eliminate the Y-axis cut-off in the original GeneSpring Volcano plots. To visualize the top DEGs in the Volcano plots, we thus transferred the DEG lists for all four APP/PS1 vs. WT settings/comparisons into GraphPad Prism. The DEGs were color-labelled (significant/non-significant/FC <-1.5, FC > 1.5) and the top DEGs with highest FC were individually labelled using their gene abbreviations. Please note that due to space limitations, it was not possible to highlight all genes. Please also note that in some cases more than one variant of a specific gene can be highlighted, due to detection of e.g., isoforms, splice variants, etc.

We regret that some labels of the x-axis were truncated or removed in the original submission. The latter occurred unintentionally while integrating the figures into the text document. In the revised version of Figure 2, the labels are now also properly displayed.

The legend of the revised Figure 2 was also adapted accordingly:

“Fig 2: Volcano plots illustrating statistical significance vs. magnitude of FC.

Profiles are depicted for the RS cortex of female APP/PS1 mice (A), the hippocampus of female APP/PS1 mice (B), the RS cortex of male APP/PS1 mice (C) and the hippocampus of male APP/PS1 animals (D). All genes exhibiting differential transcript levels were plotted with each dot representing one gene. The log2-FC in the individual APP/PS1 subgroups vs. WT is represented on the x-axis. The y-axis displays related log10 p-values (t-test procedure without further correction). The p-value of 0.05 and a FC of > 1.5 and < -1.5 are indicated by a horizontal line and vertical lines, respectively. Data points below the horizontal line represent gene candidates that were not significantly altered in transcription. Gene candidates in the different Volcano sectors (significantly up- or downregulated and FC > 1.5 or FC < -1.5) are color-coded. Note that downregulated gene candidates in the upper left sector (FC < -1.5, light orange) and upregulated genes in the upper right sector (FC > 1.5, blue) were further investigated in Venn and pathway studies. In most subgroups, the upregulated gene candidates predominate. Selected DEGs with high FC and statistical significance were individually labelled. Genes with significantly altered FC in the range from -1.5 to 1.5 (green and purple) were not further analyzed in our study.”

Point 4: Pg. 14 text and also throughout the entire results section. The target genes do not have to be completely listed in the text. It makes it incredibly laborious to read. Also, these targets are then shown in the table once again. Both sources of information appear redundant. Much better if added as supplementary tables and the results sections is written in a more concise manner to highlight major differences that are of interest instead of just regurgitating all the outputs from the figures which the readers can already see for themselves. In addition, figures look disorganized and cut and pasted directly from screenshots or generated by other software. They are pixelated and of different sizes. The entire organization of the paper makes it challenging to read.

Response: We very much appreciate the reviewer’s comment. We followed the reviewer’s suggestion and limited the number of DEGs of the individual subgroups in the text to 10 (instead of 30) (see Results section). This allows the reader to get an impression of the highest FC DEGs and switch to the supporting information (with entire lists of DEGs) if desired.

For the presentation of the GO analysis results, we completely removed the DEGs from the text, as these DEGs are also listed in Table 1 and Table 2. For the Venn analysis, we just left the signature gene lists insight the manuscript, as all the other DEGs are depicted in Figure 5 and Figure 6.

This substantially improved the readability of our revised manuscript.

Concerning the figures, we would like to point out that in the original submission, the figures were integrated into the text file and were thus of limited resolution to avoid excessive file size. None of our original figures were taken from screen shots, but exported from, e.g., GeneSpring (Agilent) analysis or Metascape. To guarantee maximum resolution in the revised version of our submission, we have exported raw data for Volcano plots into GraphPad Prism and generated a new Figure 2 (see above). In addition, we also exported the GO analysis results from Metascape to GraphPad Prism, in order to harmonize the individual image and text formats.

As requested in Point 10 (see below), we have also reorganized and significantly condensed Table 3 and Table 4 according to the reviewer’s suggestions. The latter has further increased the readability of our manuscript.

Point 5: Pg. 29. Line 865: There is no Figure 4E.

Response: We thank the reviewer for this important comment. In section 3.6.5. we referred to Figure 5E. We have made the related change in the revised version of our manuscript:

“A Venn analysis including DEGs in all four APP/PS1 subgroups is depicted in Figure 5E.”

Point 6: Fig. 5E: Any reason why DEGs are entities?

Response: We thank the reviewer for this comment. We wanted to stress that the sex- and region-specific candidates are of individual value in the interpretation of our data and have a signature character. To avoid any potential misunderstanding, we changed “entities” to “DEGs” in Figure 5E. Please note that we also made the related change in Figure 6E.

Point 7: Pg. 37. Line1014: Citation of online resources should be edited. Notations do not look accurate.

Response: We appreciate the reviewer’s comment. We double checked the citation resources and all three turned out to be valid. We edited the citation for Rfam to harmonize the notations. Please note that we also added the following references for the individual databases:

Reference for Noncode: “NONCODEV6: an updated database dedicated to long non-coding RNA annotation in both animals and plants. Zhao L et al., 2021. Nucleic Acids Research.”

Reference for Rfam: “Non‐coding RNA analysis using the Rfam database. Kalvari I et al., 2018. Current Protocols in Bioinformatics.”

Reference for RNA central: “RNAcentral: a comprehensive database of non-coding RNA sequences. The RNAcentral Consortium, 2016. Nucleic Acids Research.”

Reference for CPAT: “CPAT: Coding-Potential Assessment Tool using an alignment-free logistic regression model. Wang, L et al., 2013. Nucleic Acids Research.”

The related passage in the revised manuscript was amended as follows:

“The in silico analysis of l(i)ncRNAs was done using RNAcentral („https://rnacentral.org/“) (2), Rfam (“https://rfam.org/”) (3), the Coding-Potential Assessment Tool (CPAT, “http://rna-cpat.sourceforge.net”) (4), and various databases, e.g., NONCODE (“http://www.noncode.org/”) (5).”

Point 8: Section 3.8. It is not clear to me at all what the results are. It is merely stating what are on supplementary information without elaborating at all, the significance of the findings. What is the major conclusion for lincRNA?

Response: We thank the reviewer for this comment. Using our transcriptome approach, we have identified l(i)ncRNAs that were significantly up- or downregulated. We have screened various RNA databases as outlined in response to Point 7 to figure out whether there is any information available about those candidates identified in our subgroups. It turned out that many of the l(i)ncRNAs found in our study are listed in databases and in some cases, lincRNAs were reported to be transcribed in the CNS tissue. However, it turned out that there is hardly any information available about the potential functional implications of these lincRNAs in Alzheimer’s disease pathophysiology. Thus, one has to confess, that we are currently lacking more detailed functional data about most of the identified l(i)ncRNAs.

In our revised manuscript, we have highlighted the anticipated roles of lincRNAs in AD pathophysiology. We rephrased the entire Discussion section for our l(i)ncRNA findings and pointed out the sex- and region specificity of our l(i)ncRNA findings. Clearly, future studies are necessary to define the distinct mechanistic roles of the l(i)ncRNAs in the future. The l(i)ncRNA chapter of the Discussion section was rephrased as follows:

“L(i)ncRNAs in APP/PS1 mice.

In the mammalian genome, tens of thousands of l(i)ncRNAs were identified, with up to 40% of these l(i)ncRNAs being specifically expressed in the brain (174, 175). These l(i)ncRNAs without apparent protein-coding influence metabolism, proliferation, differentiation, and survival of neuronal cells (175). L(i)ncRNA expression has been associated with many neurodegenerative diseases (176-178) in patients or animal models and therefore, could serve as biomarkers or potential treatment targets, e.g., in AD. There is evidence that l(i)ncRNAs are aberrantly expressed in a complex spatiotemporal pattern during AD disease progression. L(i)ncRNAs are involved in the regulation of Aβ production, � hyperphosphorylation, oxidative stress, mitochondrial dysfunction, impairment of synaptic transmission and neuroinflammation, cell death etc. and are likely to affect clinical diagnosis, disease monitoring and therapy (57). These regulatory actions of l(i)ncRNAs are mediated on the transcriptional and posttranscriptional levels. L(i)ncRNAs can affect many cellular processes, including chromatin and DNA modification, RNA transcription, pre-RNA splicing, mRNA stability, and translation (175, 179, 180).

Several hundred differentially expressed l(i)ncRNAs were detected in 3xTg-AD model mice, compared to controls (181). In the hippocampus of APP/PS1 mice, 99 downregulated l(i)ncRNAs and 150 upregulated l(i)ncRNAs were observed compared to WT mice (40).

The SurePrint G3 Mouse Gene Expression v2 8x60K Microarray which we used in our study is not specifically designated to analyze l(i)ncRNAs. We identified a smaller number of l(i)ncRNAs in our individual experimental subgroups, but we were able to characterize, e.g., two l(i)ncRNAs differentially transcribed in the hippocampus of male APP/PS1 mice (l(i)ncRNA: chr18:38776580-38841080 reverse strand) and the hippocampus of female APP/PS1 animals (l(i)ncRNA: chr8:122920901-123008463 forward strand). This sex-specificity in l(i)ncRNA transcription has not been reported before (40). Our transcriptome study is the first to reveal sex- and region-specific l(i)ncRNAs in the individual subgroups. Although these l(i)ncRNA candidates are annotated in related databases (S Table 13), further studies are necessary to elucidate and validate their detailed functional implications in AD.”

Point 9: Section 3.9 The rationale for choosing these gene targets for qPCR is not well explained. Why are these target genes chosen? Is this based on previous results. You mentioned in the text “similar phenotypic features”. What are these features that have been shown that is relevant for your target genes? Why not look at more interesting sex-specific differences in gene expression?

Response: We thank the reviewer for the comment, and we apologize if the rationale for choosing the genes for qPCR has left some open questions. Within the last decade, our group has performed detailed electrophysiological studies (epidural and deep EEG recordings with subsequent time-frequency analysis) in two AD mouse lines, i.e., the 5XFAD and the APP/PS1 lines. Some years ago, we carried out microarray analysis in the hippocampus of 5XFAD mice and identified / analyzed a set of DEGs of relevance in AD. Therefore, in a first approach, we wanted to figure out whether those DEGs genes, identified in the 5XFAD hippocampus, are also up- or downregulated in the APP/PS1 mice or not. We have elaborated and clarified this issue in the revised version of our manuscript (see below).

In addition, we now also carried out new qPCR experiments for six DEGs (Siglech, Ptpn6, Laptm5, Plek, Arpp21, Shisa9). (Note that different isoforms of the Tyrosine-protein phosphatase non-receptor type 6 (encoded by Ptpn6) exist, which are differentially regulated. Therefore, Ptpn6 can be found in lists of both the up- and downregulated DEGs).

Our new qPCR experiments were carried out with the same RNA as has been used for our microarray studies. Please also note that the selection of the aforementioned gene candidates was based on, i.a., primer suitability testing and limitations of the original RNA from experimental animals. A qPCR validation of all signature genes would be beyond the scope of our study.

We have integrated the new information about our previous qPCR study (to compare gene candidates between the APP/PS1 and 5XFAD AD model) and the new qPCRs (to validate selected DEGs from our present transcriptome study in APP/PS1 mice) in the revised version of our manuscript.

In the revised Materials and methods section, the following changes were made:

“Quantitative real-time PCR (qPCR).

Two quantitative real-time PCR (qPCR) approaches were carried out: the first was used to unravel whether selected DEGs in another AD mouse model, i.e., 5XFAD, are also differentially up- or downregulated in the APP/PS1 model (16-19). This approach focusses on AD line-specific and cross-line DEGs. For this purpose, qPCRs were carried out in eight WT controls (four ♂, age 32.71 ± 0.37 wks, four ♀, age 32.93 ± 0.21 wks) and eight APP/PS1 AD mice (four ♂, age 32.53 ± 0.32 wks, four ♀, age 33,29 ± 0.00 wks). Note, that the aforementioned animals were exclusively used for this approach and not for our microarray studies. The second qPCR approach was carried out to validate a selected number of cortical DEGs (Siglech, Ptpn6, Laptm5, Plek, Arpp21, Shisa9) from our transcriptome results. Here, RNA was used from the same animals as utilized for our microarray studies (eight APP/PS1 AD mice (four ♂, four ♀, blue) and eight WT control animals (three ♂, five ♀, black), see above).

The cDNA synthesis from all RNA samples was performed using anchored-oligo(dt)18 and hexamer primer in a two-step reverse transcriptase (RT) PCR approach (Transcriptor First Strand cDNA Synthesis Kit, Qiagen GmbH, Germany). The qPCR reaction protocol was based on LightCycler 480 SYBR Green I Master (Roche, Roche Life Science, Germany). For details on primers for the specific gene candidates (Cacna1c, Cacna1d, Plcd4, Casp8, Chrm1, Chrm3, Chrm5, Siglech, Ptpn6, Laptm5, Plek, Arpp21, Shisa9) please refer to Siwek ME et al. (2015) (17) and Suppl Tab 4. The qPCR was performed in a Light Cycler 480 System (Roche Life Science, Germany) thermocycler. The following cycler protocol was carried out for all primer pairs: 95°C (10 min, pre-incubation step); 95°C (10 s, melting step); 60°C (20 s, annealing step); and 72°C (30 s, extension step), 35 cycles. Melting curve analysis was performed to evaluate the specificity of the amplification. Product detection and characterization was based on SYBR Green I Master (Roche Life Science, Germany), a ready-to-use hot start reaction mix containing FastStart Taq DNA Polymerase and DNA double-strand-specific SYBR Green I dye. Deionized, nuclease-free water (no cDNA) and total RNA samples (RT excluded) were used as negative controls. The Hprt was used as an internal reference gene (positive control). The Ct (cycle threshold) values were calculated using the LightCycler 480 System software (Roche Life Science, Germany). The FCs of Cacna1c, Cacna1d, Plcd4, Casp8, Chrm1, Chrm3, and Chrm5 gene expression for the first qPCR approach and Siglech, Ptpn6, Laptm5, Plek, Arpp21, Shisa9 for the second qPCR approach in APP/PS1 transgenic mice related to WT controls were calculated according to Schmittgen & Livak, 2008 (20). The cp values provided by the LightCycler 480 software (Roche, Germany) were imported to qBase+ software (Biogazelle, Belgium) and analyzed based on a delta-Cq quantification model with PCR efficiency correction, reference gene normalization considering the reference target stability of the selected housekeeping gene (Hprt) and inter-run calibration. The results were calculated as Calibrated Normalized Relative Quantity (CNRQ) and statistically investigated by the Mann-Whitney test.”

The Results section was amended as follows:

“qPCR analysis of selected gene candidates relevant in AD etiopathogenesis.

Comparison of selected DEGs between APP/PS1 and 5XFAD mice.

Our own previous radiotelemetric EEG studies from the cortex and hippocampus of 5XFAD mice, another model of AD disease, had revealed complex alterations in CNS rhythmicity (31, 32). Similarly, our electrophysiological studies in APP/PS1 mice had showed age- and sex-dependent differences in the relative power of the theta and gamma frequency range in the cortex and hippocampus (13, 14). Our own previous transcriptome studies in 5XFAD mice had revealed some gene candidates that might be functionally relevant for the electrophysiological alterations observed and some had been validated using qPCR (31, 32). As APP/PS1 mice exhibit similar phenotypical features as 5XFAD animals, we analyzed the same set of critical 5XFAD related genes via qPCR in the APP/PS1 model, i.e., Cacna1c, Cacna1d, Plcd4, Casp8, Chrm1, Chrm3, and Chrm5, to elaborate whether there are common, relevant DEGs in both AD mouse models (first qPCR approach). In female APP/PS1 mice, Caspase 8 (encoded by Casp8) was significantly upregulated compared to WT controls (FC: 1.3674, p = 0.028), but not the other candidates (S Table 14A, S Fig 7A, B). Interestingly, Caspase 8 did not exhibit altered expression in male APP/PS1 mice (FC: 1.0892, p = 0.685). In contrast, the high-voltage activated L-type Ca2+ channel Cav1.3 (encoded by cacna1d) was significantly downregulated in male APP/PS1 mice (FC: -1.2122, p = 0.0285) but not in female APP/PS1 animals (FC: 1.1681, p = 0.114) (S Fig 7E, F). This sex-specific difference in Cav1.3 transcript levels in APP/PS1 mice was statistically confirmed (FC: 1.4113, p = 0.0285) (S Fig 7E, F). For the high-voltage activated L-type Ca2+ channel Cav1.2 (encoded by cacna1c), no genotype- and sex-specific difference in transcript levels was detected. A significant difference in transcript levels between male and female APP/PS1 mice was also observed for Chrm1, the gene encoding the muscarinic receptor type 1 (M1) (S Fig 7I, J). These findings further stress the fundamental relevance of sex-specific analysis as carried out in our transcriptome studies.

Our transcriptome data revealed no significant up- or downregulation of Casp8, Plcd, Chrm1, Chrm3 and Chrm5. In the RS cortex and hippocampus of female APP/PS1 mice and the RS cortex of male APP/PS1 mice, PLCgamma was altered (FC: 1.3, FC: 1.4 and FC: 1.3, respectively). In the hippocampus of male APP/PS1 mice, PlCd3 was downregulated (FC: -1.2). Importantly, cacna1d encoding the Cav1.3 HVA Ca2+ channel was listed as a DEG in the RS cortex of male APP/PS1 mice, but not in the hippocampus. This phenomenon is due to the applied strict statistical procedure including one-way ANOVA, Tukey post hoc test and the application of the Benjamini-Hochberg procedure to decrease the FDR. The DEG extraction without application of post-hoc correction again included cacna1d as a differentially regulated gene candidate. These qPCR findings underline that different AD mouse lines, i.e., 5XFAD and APP/PS1, can clearly exhibit different transcriptome characteristics.

Validation of selected DEGs in APP/PS1 mice.

In the second qPCR approach, we validated a selected number of DEGs from our APP/PS1 transcriptome study, i.e., four upregulated (Siglech, Ptpn6, Laptm5, Plek, Fig 7A-H) and two downregulated (Arpp21, Shisa9, Fig 7I-L) genes. These DEGs were analyzed in the RS cortex of both male and female APP/PS1 animals (same mice (RNA) as utilized for our transcriptome studies). As qPCR analysis of all relevant DEGs, including intersectional and signature genes, would be beyond the scope of our study, we focused on a selected number of candidates in the RS cortex. The latter is early and severely affected in AD pathogenesis. Importantly, the four DEGs Siglech, Ptpn6, Laptm5, and Plek which were suggested to be significantly upregulated in female APP/PS1 mice in our transcriptome study, were also significantly upregulated in the qPCRs (Fig 7B, D, F, H, S Table 14B). The same held true for the qPCR validation of Siglech, Ptpn6, Laptm5, and Plek in the male RS cortex of APP/PS1 animals (Fig 7A, C, E) with Siglech exhibiting a statistical trend (S Table 14B). In addition, the FC-values from the transcriptome study and the qPCR validation nicely correlate. Note that many DEGs (such as Ptpn6) exhibit different transcript variants which can be up- or downregulated. For Ptpn6, the upregulated transcript variants were validated.

The two DEGs that were suggested to be downregulated in our microarray analyses (Arpp21, Shisa9), were also confirmed in our qPCR approach in the RS cortex of female APP/PS1 mice (Fig 7J, L, S Table 14B). In the RS cortex of male transgenics however, significant downregulation of Shisa9 could not be confirmed (S Table 14B). Overall, our qPCR findings underline the validity of our transcriptome studies and the detected DEGs.”

A new Fig 7 (scatter plot of qPCRs of validated DEGs in the RS cortex) was added to the revised manuscript.

A legend for the new Fig 7 was also added to the revised manuscript:

“Fig. 7. Comparative qPCR analysis of selected DEGs from the RS cortex of female and male APP/PS1 AD.

Transcript levels (CNRQ) for Siglech (A, B), Ptpn6 (C, D), Laptm5 (E, F), Plek (G, H), Arpp21 (I, J), Shisa9 (K, L) were obtained from eight APP/PS1 AD mice (four ♂, four ♀) and eight WT control animals (three ♂, five ♀). Results are depicted using scatter plots including mean ± SEM. RNA was taken from the same animals analyzed in the transcriptome approach. Control mice are shown in black, APP/PS1 animals are highlighted in light blue.”

We also added additional information about primers of the new qPCR to Suppl. Tab 4:

“Suppl Tab 4: Details on genes, forward and reverse primer sequences and annealing temperatures relevant for qPCR experimentation. The following genes, i.e., Cacna1d, Cacna1c, Plcd4, Casp8, Chrm1, Chrm3, Chrm5, Siglech, Ptpn6, Laptm5, Plek, Arpp21, Shisa9, were analyzed for quantitative transcriptional alterations using qPCR. Both forward and reverse primer sequences, annealing temperatures and information about the primer sources are provided. Hprt was used as an internal reference gene (positive control).”

Finally, we also added additional information about the FC obtained from the new qPCR experiments to Suppl. Tab 14:

“Suppl Tab 14: qPCR-based fold change analysis of selected genes in the cortex and hippocampus of APP/PS1 AD vs. WT control mice. (A) qPCR-based fold change analysis of selected genes in the hippocampus of APP/PS1 mice that were previously identified in the 5XFAD AD model (33,34). (B) qPCR-based FC analysis of selected DEGs (from our transcriptome study) in the RS cortex of APP/PS1 mice vs. WT control animals. FCs from qPCR studies are listed first, those from our microarrays are given in brackets. Note that negative FCs indicate downregulation, positive FCs indicate upregulation of related gene candidates. Significant values (p < 0.05) and statistical trends (0.05 < p < 0.1) are highlighted in bold.”

Point 10: Table 3: Consider condensing the table. It does not need to take up 3 pages. It is also hard to compare the data when it is scattered across multiple pages. Consider reordering the gene list by degree of overlap across genotypes. Alphabetical ordering contributes nothing to hierarchy.

Response: We thank the reviewer for this helpful comment and followed his/her suggestions for Table 3 (DEGs with FC > 1.5). We significantly condensed the table and in addition, we reordered the gene list by the pattern and degree of overlap.

We have included this information also in the revised legend of Table 3:

“Table 3: Overview of the sex- and region-specific upregulation of DEGs in the four APP/PS1 subgroups.

This table illustrates all upregulated DEGs (FC > 1.5) from APP/PS1 microarray analysis. DEGs that are upregulated only in one APP/PS1 subgroup are highlighted in red, those exhibiting enhanced transcript levels in two subgroups are marked in green. Increased transcript levels in three subgroups are marked in ocher, those in all four subgroups are highlighted in yellow. The table elicits the complex sex- and brain region specific profile of upregulated DEGs.”

Please note that we have applied the same approach to Table 4 (for the downregulated DEGs with a FC < -1.5).

We have included this information also in the revised legend of Table 4:

“Table 4: Overview of the sex- and region-specific downregulation of DEGs in the four APP/PS1 subgroups.

This table illustrates all downregulated DEGs (FC < -1.5) from APP/PS1 microarray analysis. DEGs that are downregulated only in one APP/PS1 subgroup are highlighted in red, those exhibiting enhanced transcript levels in two subgroups are marked in green. Increased transcript levels in three subgroups are marked in ocher, those in all four subgroups are highlighted in yellow. The table elicits the complex sex- and brain region specific profile of downregulated DEGs.”

Point 11: Consider reorganizing the discussion section. At the moment, the discussion reads like a literature review of the target genes. There are interesting discoveries your studies have revealed, for example, specific sex-related genes that appear to be region specific, like ICAM. Those should be highlighted and stressed. Perhaps even performed qPCR for verification. In any case, these interesting findings are embedded in a wall of text. That microglial/phagocytotic/neutrophil pathways are shared and activated is not surprising and should not be reiterated.

Response: We very much appreciate the reviewer’s comment. We have reorganized the Discussion section in a way that we have focused on the sex- and region-specific DEGs. We have focused the Discussion of the revised manuscript on these genes and selectively highlighted them. In addition, we have strongly reduced the amount of information about the microglial/phagocytotic/neutrophilic pathways and transferred this information to the supplementary information section (see also Suppl Tab 16) in case the reader likes to look up related information.

According to the reviewer’s suggestion, the entire Discussion section was changed as follows:

The initial section of the Discussion section elaborates the role of APP/PS1 mice in AD research and transcriptomics and focuses on AD mouse lines other than the APP/PS1 model used in our study. This section (see below) was removed to avoid any distraction:

“Previously, various transcriptome studies have been carried out in humans and animal models of AD (21, 22). Castillo E et al. (2017) performed an inter-species comparative gene expression profiling between AD patient brains and the AppNL-G-F/NL-G-F and 3xTg-AD-H mouse models based on microarrays of hippocampal and cortical tissue demonstrating a close link between amyloidosis and neuroinflammation (23, 24). Some studies specifically focused on the gene expression profiles of 10,000 individual microglial cells isolated from the cortex and hippocampus of male and female AppNL-G-F mice at the age of 3, 6 and 12 months. The authors focused on the role of age, sex and genes that modulate the microglial response to A� plaque formation (25). Early and sustained differential gene expression analysis in other AD mouse models, e.g., 3xTg AD mice, was carried out by Gatta V et al. (2014) (26). Further transcriptome studies were carried out in 5XFAD mice (27), Tg2576 mice (28) and TgCRND8 (29).”

The justification to choose both the RS cortex and hippocampus as target brain regions in our study was shortened in the revised Discussion section:

“Role of the RS cortex and hippocampus in AD etiopathogenesis and disease progression.

The cortex is known to be early affected in AD pathophysiology both on the structural and functional level (52, 83). Interestingly, progression of AD-specific lesions severely differs between individual cortical regions. Whereas the (pre)frontal cortex is early involved, alterations in the motor cortex seem to be delayed (84, 85). We have chosen the RS cortex for our transcriptome studies, as its early and severely affected in AD pathology and symptomatology (50, 86). As another predilection area in the etiopathogenesis of AD, the hippocampus was chosen as a second BROI for our transcriptome analysis. The hippocampus is judged to be a key interface structure in the brain and involved in the consolidation of memory engrams and their ecphoria (47, 87). Impaired and dysfunctional hippocampal activity together with complex structural alterations related to extracellularly localized A� plaque formation and intracellular �-deposits in APP/PS1 AD mice are typically accompanied by sophisticated deficits in memory formation, and cognitive/behavioral tests (88, 89).”

As requested, we have limited the amount of information regarding microglial / phagocytotic / neutrophil pathways:

“Functional relevance of selected DEGs in APP/PS1 AD mice.

Our data illustrate, that key gene candidates in AD etiopathogenesis are differentially regulated in brain regions of male and female APP/PS1 mice compared to WT controls (Table 3, 4). We identified some gene candidates that had already been related to AD, whereas others have not yet been attributed in detail or at all to its pathophysiology. An overall discussion of the entire set of high fold change DEGs in the individual subgroups is beyond the scope of our study. However, we have pointed out some functional aspects of selected DEGs and how they might influence the etiopathogenesis of AD with a focus on sex- and region-specific distribution. As outlined below, DEGs are predominantly related to the neuroinflammatory response, the activation of microglia, macrophages and neurophil granulocytes (37, 90). Many DEGs further contribute to innate immune reactions and immune effector processes that are largely triggered by A� (90, 91).

Intersectional DEGs in microglial/neutrophilic activation.

Microglial activity seems to play a critical role in the etiopathogenesis of AD and the devastating structural and functional alterations that encompass progression of the disease (92-94). Microglia represent the resident immune cells within the CNS and share some phenotypic markers with myeloid cells of the macrophage/monocyte lineage (95). Microglia have been functionally segregated into pro-inflammatory M1 and anti-inflammatory M2 subgroups based on their molecular and functional profile (96, 97), though the entire activation spectrum between the maxima seems to a continuum (74, 98). Strong efforts have been made to decipher the phenotypic diversity of microglial cells in the healthy, aging, and AD brains of mice and humans. AD-dysprogrammed microglia are often referred to as disease-associated microglia (DAM) (99), microglia associated with neurodegenerative disease (MGnD) (100), or activated response microglia (ARM) (101). The delineation of transcriptome profiles in these microglia has led to the specification of gene candidates related to the homeostatic signature of microglia, differentiation of microglia vs. myeloid populations, microglia vs. macrophages, the microglial surfaceome, and the microglial sensome (74, 102). Detailed comprehensive lists of microglia associated transcripts are provided by Boche and Gordon (2020) and Crotti A et al. (2016) (74, 103).

Recent advances in single-cell-omic methodologies have dramatically increased the availability of gene expression data from microglia in healthy and aged brains, and AD (74). Previous studies by Mukherjee et al. (2019) described DEGs in microglia related to neurodegenerative diseases, including Tyrobp, Fcer1g, Itgb2, Myo1f, Ptprc, Trem2 and C1qa (104). We were able to confirm many of these gene candidates in our experimental subgroups and most important, in both sexes.

The Fcer1g gene for example, plays an important role in IgE-specific FC gamma receptor mediated phagocytosis and microglial activation (105). Importantly, upon activation of Trem2, Syk is recruited by Fcer1g and Tyrobp (106, 107). Notably, Tyrobp and Trem2 were upregulated in all four APP/PS1 subgroups in our study, similar to Fcer1g which however, remained unaltered in the hippocampus of male APP/PS1 mice.

Some DEGs that were upregulated in our study are functionally related to leucocyte migration, e.g., Itgb2 (108). Myo1f is important in neutrophil degranulation and regulation of cell migration as well (108). Rac2 encodes a protein with GTPase and protein kinase regulator activity and is involved in the regulation of leukocyte activation and leukocyte chemotaxis. It belongs to the Rho family GTPases which are master regulators of actin cytoskeletal organization and associated with a number of neuropsychiatric and neurodegenerative diseases, including AD (109).

We further identified several cluster of differentiation (Cd) encoding genes to be upregulated in APP/PS1 AD mice. Most of these Cd genes, i.e., Cd48, Cd52 and Cd68, are involved in antigen binding, signaling and receptor activity. Some, like Cd68, were reported to be involved in aging and cellular response to organic substances with critical localization in lysosomes within the brain. The gene product of Trem2 forms a receptor signaling complex that activates myeloid cells, including dendritic cells and microglia. In humans and mice, mutations in Trem2 may serve as risk factors for the development of AD (110).

Further relevant genes are related to the interleukin, interferon, and cytokine system, e.g., the C-C motif chemokine 6, encoded by Ccl6, one of the top FC candidates in our study. The latter displays expression in microglia and astrocytes. Interestingly, treatment of neurons with CCL6 revealed protective effects against glutamate neurotoxicity. This neuroprotective effect of CCL6 is mediated via the chemokine receptor CCR1, and downstream by PI3K and likely to be another protective mechanism in AD pathophysiology (111). For additional description of the DEGs, please refer also to S Table 16.

Intersectional genes in phagocytic activation.

Phagocytosis is an important field in AD pathogenesis (112) and numerous DEGs in our study groups are related to this process. Capg (encoding for capping actin protein, gelsolin like), for example, is a member of the gelsolin/villin family of actin-regulatory proteins and was reported to play an important role in phagocytic vesicles and actin-related intracellular processes (113). Others, like Fcgr (FC gamma receptor) are also involved in endocytosis similar to Fcgr2b (114). The latter is located in dendritic spines and involved in phagocytosis, cytotoxicity and cellular response to A� (114). Studies in both mice and humans suggest that FcγR mediates a pro-inflammatory response which could also be responsible for vascular side effects following application of therapeutic antibodies against A� (115). Increasing evidence has emerged that FcγR expression in microglia and neurons is enhanced upon aging and involved in the etiopathogenesis of AD (115). Glial fibrillary acid protein encoded by Gfap is another example of a structural constituent of the cytoskeleton that is involved in chaperone-mediated autophagy and intracellular protein transport. It was reported to be involved in long-term potentiation (LTP) and neurogenesis. The Gfap gene product is currently under discussion for its utility as a plasma biomarker for AD-related pathologies (116). Itg2 (coding for integrin beta2), Laptm5 (encoding for lysosomal-associated protein transmembrane 5) and Lcp1 (encoding for lymphocyte cytosolic protein 1) are involved in cargo receptor processes and endocytosis, lysosomal transport and filament bundle assembly located in phagocytic cups (117). Notably, we were able to validate Laptm5 transcript upregulation via qPCR in the RS cortex of both male and female APP/PS1 mice.

The prion protein encoded by Prnp exhibited one of the highest fold-changes in our study. It serves several functions, including A� binding activity, regulation of protein localization to membranes, protein phosphorylation, aspartic-type endopeptidase inhibitor activity, negative regulation of apoptotic processes and modulation of phagocytosis (118). There is likely to be an association between AD and cellular prion protein (PrPC) levels. PrPC exerts high affinity for oligomeric Aβ (particularly A�42) and mediates A�-induced neurotoxicity in AD pathogenesis (119). Synaptotoxic Aβ assemblies further mediate long-term depression (LTD) via action on PrPC (120). PrPC is decreased in the hippocampus and temporal cortex in aging and sporadic AD but not in familial Alzheimer’s disease (FAD) (121). Another study also suggested a reduced expression of PrPC in AD patients compared to healthy individuals, which pointed to a potential protective role of PrPC expression in AD (122), although this remains controversial (123). However, with symptomatic clinical manifestation, PrPC expression seemed to decrease again (123). There is clearly an ambiguous picture of PrPC expression levels in AD that needs future clarification (118). Interestingly, Prnp was strongly co-upregulated in the RS cortex and hippocampus of both female and male APP/PS1 mice in our study.

One of the highest FC candidates in our study turned out to be glycosidase lysozyme 2 (Lyz2). Alzheimer’s disease patients were reported to exhibit increased lysozyme levels in the cerebrospinal fluid. In addition, lysozyme 2 co-localizes with A� and exerts protective effects against A� associated neurodegenerative action. Importantly, lysozyme was proposed as a potential biomarker and therapeutic target in AD (124).

The gene product of Siglech (sialic acid binding Ig-like lectin H, possibly an ortholog of CD33 in humans) has cargo receptor activity and acts upstream of or within receptor-mediated endocytosis and was reported to serve as a risk factor for late-onset AD (125). Importantly, Siglech was upregulated in each APP/PS1 subgroup in our study despite the hippocampus of male AD animals. Our qPCR experiments further validated the upregulation of Siglech in the cortex of APP/PS1 mice. For additional DEG profile information see also S Table 16.

Intersectional genes encoding for endopeptidases in APP/PS1 AD mice.

Endopeptidases play an important role in AD pathology and are potential drug target candidates and biomarkers with high predictive efficacy (126). We identified several endopeptidases to be upregulated in the different APP/PS1 subgroups. The latter include Bcl2a1d (encoding for B cell leukemia/lymphoma 2 related protein A1d), a cysteine-type endopeptidase involved in apoptotic processes and Ctsd, an aspartic-type endopeptidase (cathepsin D) localized in the extracellular space and lysosomes and involved in autophagosome assembly (127). Bcl2a1d turned out to be a top fold-change candidate in our study. Other upregulated endopeptidases include Ctsh, a cysteine protease (cathepsin H) that can form both aminopeptidase and endopeptidase activity depending on its posttranslational processing and assembly. Ctss represents another upregulated peptidase C1 family member of cysteine protease (cathepsin S, like Ctsh). Naip2 (NLR family, apoptosis inhibitory protein 2) encodes a protein with ATP binding activity and cysteine-type endopeptidase inhibitor activity involved in apoptotic process (128). Naip2 was upregulated in all four APP/PS1 subgroups in our study.

Again, our study revealed sex-specific characteristics in differential regulation: the top fold-change gene candidate Bcl2a1d was upregulated in all APP/PS1 groups, whereas Ctsd and Ctss were not upregulated in the female hippocampus, and Ctsh was not upregulated in the RS cortex of male and female APP/PS1 animals. For further details please refer to S Table 16.

Oxidative stress and reactive oxygen species in APP/PS1 AD mice.

Reactive oxygen species (ROS) play an important role in AD pathogenesis and microglia-associated neuroinflammation (129). DEGs such as Cyba and Ncf are part of the NADPH oxidase complex. They are involved in ROS metabolic processes, Ca2+ homeostasis and AD pathogenesis (130). The Nrf2 encoded NF-E2-related-factor 2 was reported to suppress inflammation and oxidative stress in an APP knock-in AD mouse models and could thus exert protective effects (131). For additional information see also S Table 16.”

We have further introduced individual subsections for the discussion of selected signature genes in the individual subgroups:

“Sex- and region-specific alterations in gene transcript levels in APP/PS1 mice.

In some AD mouse models and humans, sex-specific differences in transcriptome profiles have been partially documented. Transcriptional human-mouse intersections are mainly related to A�/� pathology and microglia and exhibit sex-dependent spatiotemporal signatures upon disease progression. The results from RNA-seq of human post mortem brain tissue stressed the relevance of microglial and inflammatory mechanisms in AD pathogenesis, among other pathways (73, 132-134). Notably, different from transcriptome profiles in human brains, mouse models allow more controlled experimental conditions that can be used to investigate causation, isolated effects of specific molecular or pharmacological interventions, and elaborate spatiotemporal alterations. Although transcriptional overlaps have been identified between AD mouse models and human brains in some studies (73, 135-137), other reports have questioned the overall degree of conservation (113, 138). Regardless of these discrepancies, sex/gender has been suggested as a strong modulator in AD etiopathogenesis (73, 139, 140).

Most studies carried out in APP/PS1 mice and other AD models do not provide sex-specific analysis of transcriptome data (see S Table 15). In the following, we have elaborated the sex-specific profile of the major DEGs detected in our study. Importantly, these gene candidates turned out to be signature genes (Fig 5E). In addition, many of them contribute to the devastating AD pathophysiology. However, some (as outline below) also exert counteracting, autoprotective effects.

Functional relevance of signature genes in the hippocampus of female APP/PS1 mice.

In our study design, 17 gene candidates turned out to be exclusively upregulated in the hippocampus of female APP/PS1 AD mice, i.e., Prss23, Slc38a5, Ifitm3, Icam1, Dclk1, Rfx3, Tubb6, Slfn5, Flnc, Apobec1, Lsp1, Myo1g, Vim, 5830408C22Rik, Ptgr1, Pla2g4a, Rab32. The latter are related to the following GO categories: epithelial cell development, membrane trafficking, supramolecular fiber organization, and the adaptive immune system. Two related signature genes are highlighted below:

Intercellular adhesion molecule-1 (Icam)

An interesting candidate in our study turned out to be Icam1, encoding for the intercellular adhesion molecule-1, a transmembrane glycoprotein that is reported to be overexpressed in many neuropathological settings. Icam1, which is known to play an important role in the immune response, is expressed in the CNS, particularly in astrocytes, microglial and endothelial cells in the white and gray matter of the human forebrain. Icam1 has been attributed to the etiopathogenesis of neurodegenerative, e.g., AD and neuropsychiatric disorders, due to its functional involvement in blood–brain barrier function and as a marker for inflammation (141). Interestingly, the Icam1 gene product was suggested to be neuroprotective and a potential candidate in cytokine-mediated therapy of AD (141). Studies in humans revealed that the Icam1 gene product was increased in the cerebrospinal fluid (CSF) in preclinical, prodromal, and dementia stages of AD, associated with cortical thinning and cognitive deterioration and could act as a biomarker for early neuroinflammation and an AD risk assessment factor (142). Although Icam1 was previously suggested to serve as a CSF biomarker for AD, we found Icam1 to be significantly upregulated (FC > 1.5) only in the hippocampus of female APP/PS1 mice and it turned out to be one of the differentially expressed fingerprint genes in this subgroup.

Tubulin beta 6 (Tubb6)

Tubb6 encodes for a tubulin isotype, i.e., tubulin beta 6 class V with GTP binding activity, a constituent of the cytoskeleton being involved in microtubule and actin dynamics. Microtubule dysfunction is a prominent feature in many neurodegenerative diseases including AD (143). Tubb6 mRNA was previously detected in the hippocampus of C57BL/6 mice and mRNA levels were reported to be low and to further decrease in early age (144). Proteome studies indicated an overrepresentation of tubulin beta 6 in mouse models of AD (145). Importantly, in our transcriptome studies, Tubb6 was only upregulated in the hippocampus of female APP/PS1 mice, not in the RS cortex and not in male APP/PS1 mice.

Functional relevance of signature genes in the RS cortex of female APP/PS1 mice.

In the RS cortex of female APP/PS1 AD mice, 25 candidates exhibited signature character, i.e., Gusb, Slc14a1, Mir142hg, Cyth4, Gpr34, Plek, Ccl9, Aif1, Psmb9, Lair1, Hk3, Csf2rb, Tlr12, Adora3, Apbb1ip, Pycard, Vav1, Oas1a, Ifi27l2a, Hexb, Ncf4, Fcrls, Irf5, Havcr2, Mlxipl. These specific DEGs turned out be related to the fields of inflammatory response, Tyrobp causal network in microglia, neutrophil degranulation, phagocytosis and the RAF/MAP kinase cascade. Some of the related gene candidates and their implications in AD are further illuminated below:

Pleckstrin (Plek)

A most interesting DEG selectively upregulated in the RS cortex of female APP/PS1 mice turned out to be Plek. Pleckstrin is an important protein in cytoskeleton reorganization and neurite outgrowth and is supposed to be of functional relevance in AD pathophysiology as the structural and functional integrity/dynamics of the cytoskeleton are a prerequisite for neuronal function and survival (146, 147). Pleckstrin (Plek) turned to be a negatively regulated target of miR-409-5p, a miRNA that impairs neurite outgrowth. Interestingly, Aβ1-42 peptides significantly decrease cortical expression of miR-409-5p, resulting in less decrease of neuronal viability, and reduction in subsequent Aβ1-42-induced pathologies (147). In contrast, overexpression of miR-409-5p exerts neurotoxic effects in neuronal cells and impairs differentiation. Overexpression of Plek, on the other hand, harbors the capability to rescue neurite outgrowth from this neurotoxicity (147). It has previously reported that miR-409-5p is early downregulated in APP/PS1 mice and that Plek is upregulated at 9 and 12, but not 4 and 6 months-old APP/PS1 animals (147). Our study reveals that Plek is upregulated in 8 months old animals, and according to our transcriptome results, a signature gene in the RS cortex of female APP/PS1 mice.

Strikingly, the previously reported early downregulation of miR-409-5p in combination with the observed upregulation of Plek in AD progression might thus point to a counteracting, compensatory and autoprotective reaction to alleviate the structural and functional synaptic impairment induced by Aβ. Plek and its gene product Pleckstrin could thus serve as potential early biomarker in AD.

Adenosine receptor type 3 (Adora 3)

Some upregulated genes, such as Adora 3, are not only involved in microglia activation but also neutrophil degranulation and leucocyte migration/chemotaxis. Adora3 encodes a G-protein coupled adenosine receptor type 3, which is supposed to mediate inhibition of neutrophil degranulation and therefore potentially prevents cell damage. However, it was also reported to exhibit ambiguous effects in neuroprotection and neurodegeneration (108, 148, 149) stressing that modulation of the adenosinergic system in AD, in terms of pathology and therapeutics, is a sophisticated issue (150). Despite this potential intractability and the lack of medicines targeting adenosinergic receptors for AD treatment, substantial efforts are made in drug research and development. Importantly, Adora3 turned out to be a unique fingerprint candidate for the RS cortex of female APP/PS1 mice in our study. This sex-specific and potentially neuroprotective profile of Adora3 warrants future attention also in drug development and potential individualized pharmaceutical treatment in AD.

Vav Guanine Nucleotide Exchange Factor 1 (Vav1)

Vav1 encodes a guanine nucleotide exchange factor (GEF) for Rho family GTPases that activate pathways leading to actin cytoskeletal rearrangements and transcriptional alterations. It serves as a functional regulator of Rac2. It acts upstream of or within several processes, including integrin-mediated signaling pathway and neutrophil chemotaxis (151). Vav1 also seems to promote learning by activating HIF-1 and GLUT-1 and thereby contributing to glucose distribution to the brain, another important factor in AD (152). Interestingly, Vav1 turned out to be a unique fingerprint DEG for the RS cortex of female APP/PS1 mice.

Apoptosis inducing factor (Aif)

The apoptosis inducing factor (Aif) is responsible for caspase-independent programmed cell death (153). Aif was reported to mediate cell death in the TgCRND8 AD mouse model in a region-specific manner with aging-related cell death in the cortex but not the hippocampus (153). Notably, this holds true for our findings in APP/PS1 AD mice as well. Indeed, Aif turned out to be a unique fingerprint gene in the cortex of APP/PS1 mice, however, only in females and not in males – a phenomenon that has not been specified before (153).

Hexosaminidase subunit beta (Hexb)

The gene product of Hexb (hexosaminidase subunit beta) has been associated with AD based on genome-wide association studies (GWAS) (154). In addition, �-hexosaminidase can lead to a reduction of A� complexes, the aggregation and accumulation of which has been accelerated by exposure to gangliosides (155). Importantly, Hexb was unique for the RS cortex of female APP/PS1 mice.

Proteasome 20S subunit beta 9 (Psmb9)

The gene products of Psmb9 and Psmb8 (encoding for proteasome 20S subunit beta 9 and 8 respectively) form part of a proteasome (prosome, macropain) core complex which plays an important role in AD (156). The ubiquitin-proteasome system (UPS) is a major regulator of protein homeostasis. In contrast, the immunoproteasomes constitute a specialized form of proteasomes, which modulate inflammatory processes in AD via clearance of oxidant-damaged proteins. In APP/PS1 mice, Aβ-deposition was reported to parallel upregulation of immunoproteasomes. The latter turned out to be upregulated by pro-inflammatory cytokines, e.g., type I and type II interferons (IFNs). Wagner et al. (2017) suggested that immunoproteasomes are upregulated in the innate immune response towards extracellular Aβ-accumulation, although previous rodent studies revealed inconsistent data with decreased, unchanged, and increased proteasomal activity with aging (156-158). Wagner et al. (2017) used both males and females. However, they did not analyze for sex-specific differences (156). Strikingly, Psmb8 was upregulated only in the RS cortex and hippocampus of females and not in males in our study. In contrast, Psmb9 evolved as a unique fingerprint gene for the RS cortex of female APP/PS1 mice. The potential neuroprotective upregulation of Psmb9 thus seems to be attributable to the predilected brain region and sex in AD.

PYD and CARD domain containing gene (Pycard)

Pycard (PYD and CARD domain containing gene, encoding for adapter protein apoptosis-associated speck-like protein containing a CARD (ASC)) codes for an inflammasome component and is involved in regulation of autophagy. Previous reports demonstrated that released ASC specks can bind to Aβ, enhance its aggregation, and increase its toxicity (159). In our study, Pycard turned out to be a unique fingerprint for the RS cortex of female APP/PS1 mice. As Pycard was recently suggested to be a promising diagnostic target in early AD patients (160), sex-specific characteristics require further attention.

G protein-coupled receptor 34 (Gpr34)

The G-protein-coupled receptor 34 (Gpr34) was found to be highly expressed in the hippocampus of APP/PS1 mice (161) and to be involved in microglia phagocytic activity, complement activation and synaptic pruning (174). In vitro and in vivo Gpr34 knockdown approaches resulted in decreased levels of TNF-α, IL-1β, IL-6 and iNOS and suppression of ERK/NF-κB signal activation. Similarly, Gpr34-overexpression resulted in activation of ERK/NF-κB signalling and upregulation of TNF-α, IL-1β, IL-6 and iNOS. Systemically, Gpr34 knockdown relieved cognitive deficits in APP/PS1 mice and limited neuroinflammation and microglial activation, most probably via the ERK/NF-κB pathway (161).

Functional implications of signature genes in the hippocampus and RS cortex of male APP/PS1 mice.

In the hippocampus of male APP/PS1 AD mice, exclusive upregulation was observed in 18 gene candidates, i.e., Sgk1, Ncf1, Adamtsl4, Hck, St8sia6, Gm20743, Irf9, Pfkfb3, B2m, Arrdc2, Cebpa, Synpo2, Cep126, Rhoh, Hist1h2be, Clec5a, Cd86, Ptpn18. Some are discussed below in more detail:

Hematopoietic cell kinase (Hck)

The hematopoietic cell kinase encoded by HcK serves as a factor of the innate immune system. It has previously been demonstrated that the Hck pathway exerts important functions in the regulation of microglial neuroprotective functions during the early stage of AD (162). Strikingly, hematopoietic cell kinase deficiency aggravated cognitive impairment along with elevated Aβ levels and plaque formation. It further attenuated Aβ phagocytosis and enhanced iNOS expression in microglia (163). Thus, Hck is likely to exert a prominent neuroprotective function via modulation of microglial function and could attenuate early AD development.

Serum- and glucocorticoid-inducible kinase 1 (Sgk1)

Serum- and glucocorticoid-regulated kinase 1 (encoded by Sgk1) is a serine/threonine protein kinase that activates certain K+, Na+, and Cl- channels, involved in cell survival and neuronal excitability (164). SGK1 is engaged in various neurodegenerative pathways related to, e.g., apoptosis, autophagy, neuroinflammation, and ion channel regulation. Interestingly, hippocampal overexpression of Sgk1 improved spatial memory, reduced the devastating impact of Aβ accumulation and rescued actin cytoskeleton polymerization in middle-aged APP/PS1 mice (164, 165). It has been speculated that Sgk1 could exert a protective role against oxidative stress and play an antiapoptotic role (166). Whether pharmacological stimulation of serum- and glucocorticoid-regulated kinase 1 is protective in AD as well needs to be determined in the future.

Synaptopodin (Synpo2)

Synaptopodin serves as an actin-binding protein that is tightly associated with the spine apparatus and plays an important role in synaptic plasticity (167). Synaptopodin forms clusters in spines and regulates Ca2+ release from internal stores via ryanodin receptors in dendritic spines (167, 168). Deficiency of synaptopodin resulted in LTP deficits, impaired spatial memory and a lack of synaptic plasticity (167, 169, 170). Interestingly, synaptopodin expression was reported to be downregulated in the hippocampus of patients suffering from dementia with Lewy bodies, Parkinson's disease, mild cognitive impairments, and AD (170, 171). However, we observed a selective upregulation of synaptopodin in the hippocampus of male APP/PS1 mice. The later might point to a potential autoprotective role of Synpo2 in this experimental setting.

It should be emphasized that impairment of the structural and functional integrity of synapses is a hallmark in AD pathophysiology. Aβ causes disruption of NMDA and AMPA receptors, Ca2+ dyshomeostasis, reduced synaptic plasticity, suppression of LTP and aggravated LTD. Microglia/astrocyte activation, cytokine release, mitochondrial disruption, impairment of the cytoskeleton organization and deficits in energy metabolism further enhance synaptic dysfunction (120). Mechanistically, a number of synapse-related target molecules and signaling pathways are related to these phenomena, e.g., Wnt/β-catenin, IKK/NF-κB, JAK2/STAT3, JNK, Akt, MAPK, caspase-3, GSK-3β and CDK-5 (120). Many DEGs identified in our study (see above) enhance synaptic dysfunction, e.g. PrPC, whereas others, such as synaptopodin might exert synaptoprotective effects.

Interestingly, only two candidates, Derl3, and Mki67, exhibited unique upregulation in the RS cortex of male APP/PS1 AD animals (Fig 5E).

Derlin (Derl3)

Derlin 3 (encoded by Derl3) resides in the endoplasmic reticulum (ER) and is involved in the degradation of mis-folded glycoproteins in the ER. Dysfunction of Derlin 1 and 2 have previously been related to neurodegenerative diseases (75).

Ki-67 (Mki67)

Mki67 encoding for Ki-67 serves as cell proliferation marker and is maximally expressed in G2 phase and mitosis. Interestingly, previous studies identified MKi67 (and also Top2a, Mcm2, Tubb5) to be enriched in DNA replication and chromatin rearrangement in a specific subgroup of microglial cells, i.e., cycling/proliferating microglia (CPMs), that contributes only 0.3%–1.2% of the total microglial population (76). It has further been suggested that Ki-67 is involved in the pathogenesis of neurofibrillary degeneration in AD (172).

As outlined previously, the number of downregulated genes in the individual subgroups turned out to be significantly lower compared to the upregulated ones (Fig 6). However, it was possible to characterize unique profiles of downregulated genes in the individual subgroups. In the hippocampus of female APP/PS1 AD mice, these signature genes included 13 candidates (Asic4, Pcsk1, Etnppl, Myh7b, Spag5, Gpr1, Myh8, Ccdc184, Pdzph1, Trim66, BC030499, Rims3, Prr16), only two candidates (LOC105247294, Ptpn6) in the RS cortex of female APP/PS1 AD mice, six gene candidates in the hippocampus of male APP/PS1 AD mice (4933407I18Rik, Hlcs, Hdac9, Pla2g4e, Wnt9a. Shisa9) and four candidates in the RS cortex of male APP/PS1 AD animals (6530402F18Rik, Ctr9, Scyl2, 1700001L05Rik) (Fig 6E).

Notably, the number of intersectional, upregulated genes was much higher in the RS cortex of APP/PS1 mice compared to the hippocampus (52 versus 32, Fig 5C, D). Many of these additional, intersectional candidates in the RS cortex (Fig 5C) are recruited from gene candidates that turned out to be selectively expressed in the male or female hippocampus (Fig 5D). Consequently, more gene candidates are specifically upregulated in the female hippocampus compared to the RS cortex (45 vs. 39) and the male hippocampus compared to the RS cortex (25 vs. 5). Thus, sex-specific differences in DEGs are more overt in the hippocampus than in the RS cortex.

It turned out that several DEGs are region-specific as well. It has been shown for example, that microglia in different brain regions can differ in gene expression, particularly in genes related to bioenergetic and immunoregulatory pathways (173). Interestingly, the number of co-upregulated DEGs in the RS cortex and hippocampus of females was much higher than the related number in male APP/PS1 mice (55 vs. 33, Fig 5).

These findings are supported by symptomatic sex dimorphism profiling in 12 months old male and female APP/PS1 AD mice performed by Jiao et al. (2016) (139). Indeed, there is increasing evidence, that sex-biased and BROI-specific patterns may originate from differences in vulnerabilities/susceptibilities and/or resilience in different brain regions at different AD progression states. Female APP/PS1 mice exhibited a higher parenchymal A� load compared to male APP/PS1 animals. The latter was most prominent in the hippocampus (139). In addition, cerebral amyloid angiopathy and related microhemorrhage was more frequent in female APP/PS1 mice. Although top-ranked, intersectional genes associated with neuroinflammation, microglia activation and immune response are found in both sexes and both the RS cortex and hippocampus, specific gene sets exert fingerprint character in the individual subgroups. Notably, related parameters, including levels of phosphorylated �, proinflammatory cytokines, microgliosis, astrocytosis, and synaptic/neuronal disintegration were particularly enhanced in female APP/PS1 animals (139). Some DEGs that predominate in females are related to mitochondria and ROS. In females, mitochondrial function seems to be more resistant against Aβ mediated neurotoxicity. This phenomenon is probably due to a reduction of ROS and suppression of apoptogenic signals via estrogen (174). This may be one reason why females suffer more from AD with decreasing estrogen levels following menopause. Brain-region und sex-related specificities in DEGs, as observed in our study, might also be due to differences in brain region connectivity/architecture between males and females. The latter suggests that network characteristics in females favor a more rapid spread of neurofibrillary tangles in the CNS (175). In addition, different BROIs are disproportionally modulated via complex spatiotemporal activity of sex hormones (176-178).

In summary, the determination of sex- and brain region-specific transcriptional profiles as presented here, is highly necessary for future characterization and validation of DEGs as potential biomarkers and personalized medicinal approaches.”

We have further added a brief paragraph elaborating the co-appearance of disease-aggravating and counteracting, self-protective DEGs in our APP/PS1 subgroups:

“Disease aggravating versus neuroprotective DEGs in APP/PS1 mice.

Our transcriptome studies in APP/PS1 AD mice have revealed a high number of significantly upregulated and downregulated gene candidates. Whereas many of them turned out to be intersectional, some exhibited signature characteristics. It’s interesting to note that not all DEGs contribute to the devastating complex pathophysiology and pathobiochemistry of AD. Indeed, some candidates, such as Plek, Adora3 and Psmb9 in the female RS cortex, Icam1 in the female hippocampus, and Hck and Sgk1 in the male hippocampus of APP/PS1 mice seem to exert counteracting, autoprotective functions in the individual subgroups. Some intersectional genes such as, Ccl6, Lyz2 and Nrf2 also exhibit self-protective effects. They clearly deserve special attention in drug research and development in the future.”

We also added a chapter to the revised Discussion section, elaborating the potential interference of our transcriptome results with APP overexpression in the APP/PS1 model:

“Influence of APP overproduction and transcriptome profiles.

Overexpression of WT APP and mutant APP variants has been used to establish many well-characterized transgenic AD mouse lines, including APP/PS1 (184). It’s noteworthy that APP overexpression in these first-generation transgenic mouse models can cause overproduction of different APP fragments in addition to A The latter can make it challenging to distinguish the functional impact of these fragments and might affect the translational relevance of such models (185, 186). Potentials limitations of mutant APP- and APP/PS-overexpressing mouse models, such as APP/PS1 include, e.g., the lack of non-coding regions of the APP gene that affect splicing of APP mRNA and transcriptional regulation, unphysiological interaction of overexpressed APP and overproduced non A� fragments with cellular proteins such as kinesin via JIP-1, non-specific ER impairment upon APP/PS-overexpression, appearance of A� entities that are different from those identified in clinical AD brain, different region specificity of A� pathology, and inconsistent drug effects (187). Notably, second-generation mouse models are likely to overcome these intrinsic limitations of the APP overexpression paradigm as they utilize an APP knock-in strategy in order to selectively overexpress, e.g., A�42, but not APP. The latter was achieved, e.g., in single APP knock-in mouse models with a murine A� sequence carrying three humanized amino acids that differ between mice and humans (187). However, these models are facing numerous limitations as well (187). Thus, it’s important to be aware of the potential interference of APP- or APP/PS-overexpression with AD disease pathology in first generation AD mouse models (187).”

Finally, the impact of our findings on personalized medicinal approaches in the future was described in more detail:

“Conclusions.

Our APP/PS1 AD model-based data support the hypothesis that a major response in the brain to A� accumulation is related to microglia activation, immune response and neuroinflammation. Early and profound neuroinflammatory/immune response seems to be a fundamental driving force in AD pathogenesis. Many DEGs were claimed to be involved in these processes, however, sex- and region-specific differentiation of their functional relevance is rare. Our data clearly illustrate that sex-specific differences in the transcriptome profiles are of major relevance and that some DEGs, i.e., signature genes, are exclusively upregulated or downregulated in males or females (Table 3, 4). So far, numerous defective genes, transcriptional and translational alterations were shown to contribute to AD pathogenesis, further influenced by diverse environmental factors and epigenetic phenomena. The consideration of sex, age, comorbidity factors, disease progression state and most important, the individual genomic profile, are mandatory for a valid, promising, personalized pharmacotherapeutic approach in the future. The latter will require distinct protocols and strategies in drug research and development, the planning of clinical trials to optimize therapeutics and the establishment of new diagnostic approaches. The analysis of the patient’s individual genome including gene sets associated with disease pathogenesis, mechanisms of drug action, drug metabolism, drug transporters and multifaceted cascades and metabolic pathways, will dramatically facilitate the personalized therapeutic approach in AD.”

 

Reviewer 2:

In this article entitled “Sex- and region-specific cortical and hippocampal whole genome transcriptome profiles from control and APP/PS1 Alzheimer’s disease mice”, Papazoglou et al have showed differentially expressed gene signatures from retrosplenial cortex and hippocampus of age-matched male and female APP/PS1 AD mice and control animals. They found genes for activation of microglial, astrocytic and neutrophilic cells, innate immune response/immune effector response and neuroinflammation, phagosome and proteasome activation, and synaptic transmission were upregulated, while the genes for synaptic vesicle docking/fusion machinery, synaptic transmission, rRNA processing, ubiquitination and proteasome degradation, histone modification and cellular senescence were downregulated.

Although sex dependent transcriptome profiling is not very novel in hippocampus, and there are a few studies which have done transcriptome profiling in a sex dependent manner. Although the study is not completely novel it is interesting in terms of the systematic profiling of both regions (cortex and hippocampus) in a sex dependent fashion.

Response: We thank the reviewer for the thorough appraisal of our work. We agree that there are rare previous sex specific transcriptome studies in AD models/individuals from different species. However, as we have elaborated in the Supplementary Table 15, we present here for the first time, comparative transcriptome results from both sexes and two different brain regions of APP/PS1 mice at an age of eight months. As far as novelty is concerned, such distinct analyses have not been carried out before. Previous transcriptome studies were performed either in male APP/PS1 mice only, or both sexes were mixed (in a balanced way or there were no further specifications about the sex distribution). In addition, previous studies were mainly carried out in one brain region only (e.g., the hippocampus) or the brain area was not further specified. In some cases, the entire brain was used or the entire cortex without further specification, although this is of central relevance in Alzheimer’s disease (AD) (see also Supplementary table 15). Given the current knowledge about the overt- sex- and brain region-specific alterations in AD pathogenesis, we are confident that our revised manuscript provides substantial new information for the reader about the distinct sex- and brain-region specific profiles.

Please address these concerns:

Point 1: Line 111 suggest Sudden death in APP/PS1 mice peaks around 3–4 months of age, This is not commonly seen at 3-4 months there are many studies that reported is the time when the synaptic deficits starts to appear.

Response: We thank the reviewer for this comment. The information provided in Line 111 is given by Minkeviciene et al. (2009). Here it says: “Sudden deaths in these mice may occur at any age but peak around 3–4 months of age when the first plaques appear in the cortex and the hippocampus (Garcia-Alloza et al., 2006; Shemer et al., 2006).” We agree with the reviewer that maximum sudden death around the time point of first plaques formation is questionable in the APP/PS1 AD mouse line and we thus removed this statement from the revised version of our manuscript.

Point 2: Please discuss the impact of the study based on region and sex specific changes in transcriptome on the pathogenesis and prevention.

Response: We followed the reviewer’s suggestion and further discussed the relevance of our sex- and region-specific transcriptome alterations on pathogenesis and prevention in the revised Discussion section. The following changes were made:

“It turned out that several DEGs are region-specific as well. It has been shown for example, that microglia in different brain regions can differ in gene expression, particularly in genes related to bioenergetic and immunoregulatory pathways (173). Interestingly, the number of co-upregulated DEGs in the RS cortex and hippocampus of females was much higher than the related number in male APP/PS1 mice (55 vs. 33, Fig 5).

These findings are supported by symptomatic sex dimorphism profiling in 12 months old male and female APP/PS1 AD mice performed by Jiao et al. (2016) (139). Indeed, there is increasing evidence, that sex-biased and BROI-specific patterns may originate from differences in vulnerabilities/susceptibilities and/or resilience in different brain regions at different AD progression states. Female APP/PS1 mice exhibited a higher parenchymal A� load compared to male APP/PS1 animals. The latter was most prominent in the hippocampus (139). In addition, cerebral amyloid angiopathy and related microhemorrhage was more frequent in female APP/PS1 mice. Although top-ranked, intersectional genes associated with neuroinflammation, microglia activation and immune response are found in both sexes and both the RS cortex and hippocampus, specific gene sets exert fingerprint character in the individual subgroups. Notably, related parameters, including levels of phosphorylated �, proinflammatory cytokines, microgliosis, astrocytosis, and synaptic/neuronal disintegration were particularly enhanced in female APP/PS1 animals (139). Some DEGs that predominate in females are related to mitochondria and ROS. In females, mitochondrial function seems to be more resistant against Aβ mediated neurotoxicity. This phenomenon is probably due to a reduction of ROS and suppression of apoptogenic signals via estrogen (174). This may be one reason why females suffer more from AD with decreasing estrogen levels following menopause. Brain-region und sex-related specificities in DEGs, as observed in our study, might also be due to differences in brain region connectivity/architecture between males and females. The latter suggests that network characteristics in females favor a more rapid spread of neurofibrillary tangles in the CNS (175). In addition, different BROIs are disproportionally modulated via complex spatiotemporal activity of sex hormones (176-178).

In summary, the determination of sex- and brain region-specific transcriptional profiles as presented here, is highly necessary for future characterization and validation of DEGs as potential biomarkers and personalized medicinal approaches.”

Point 3: The role of immune mechanisms in the etiology of AD is well studied, so discuss in terms of comparisons of region and sex.

Response: We thank the reviewer for this helpful comment. Please note that we have addressed this issue together with our response to Point 2, as both comments point to the discussion of sex- and region-specific differences. The following information was added to the revised Discussion section (see also our response to Point 2):

“It turned out that several DEGs are region-specific as well. It has been shown for example, that microglia in different brain regions can differ in gene expression, particularly in genes related to bioenergetic and immunoregulatory pathways (173). Interestingly, the number of co-upregulated DEGs in the RS cortex and hippocampus of females was much higher than the related number in male APP/PS1 mice (55 vs. 33, Fig 5).

These findings are supported by symptomatic sex dimorphism profiling in 12 months old male and female APP/PS1 AD mice performed by Jiao et al. (2016) (139). Indeed, there is increasing evidence, that sex-biased and BROI-specific patterns may originate from differences in vulnerabilities/susceptibilities and/or resilience in different brain regions at different AD progression states. Female APP/PS1 mice exhibited a higher parenchymal A� load compared to male APP/PS1 animals. The latter was most prominent in the hippocampus (139). In addition, cerebral amyloid angiopathy and related microhemorrhage was more frequent in female APP/PS1 mice. Although top-ranked, intersectional genes associated with neuroinflammation, microglia activation and immune response are found in both sexes and both the RS cortex and hippocampus, specific gene sets exert fingerprint character in the individual subgroups. Notably, related parameters, including levels of phosphorylated �, proinflammatory cytokines, microgliosis, astrocytosis, and synaptic/neuronal disintegration were particularly enhanced in female APP/PS1 animals (139). Some DEGs that predominate in females are related to mitochondria and ROS. In females, mitochondrial function seems to be more resistant against Aβ mediated neurotoxicity. This phenomenon is probably due to a reduction of ROS and suppression of apoptogenic signals via estrogen (174). This may be one reason why females suffer more from AD with decreasing estrogen levels following menopause. Brain-region und sex-related specificities in DEGs, as observed in our study, might also be due to differences in brain region connectivity/architecture between males and females. The latter suggests that network characteristics in females favor a more rapid spread of neurofibrillary tangles in the CNS (175). In addition, different BROIs are disproportionally modulated via complex spatiotemporal activity of sex hormones (176-178).

In summary, the determination of sex- and brain region-specific transcriptional profiles as presented here, is highly necessary for future characterization and validation of DEGs as potential biomarkers and personalized medicinal approaches.”

Point 4: Are these upregulated genes pathogenic or is there any neuroprotective genes?

Response: We thank the reviewer for this important comment. Some of the DEGs that we detected in our transcriptome study were reported to be pro-amyloidogenic and seem to constitute critical factors in Alzheimer’s disease pathophysiology. However, we also detected DEGs that exhibit anti-amyloidogenic properties and that were upregulated in brain regions of APP/PS1 mice. Thus, transcriptional alterations in APP/PS1 mice can also be “self-protective” and exert counteracting neuroprotective effects. We have reorganized the Discussion section focusing on the signature DEGs in the individual subgroups. Potential neuroprotective DEGs are discussed here (see below).

“Sex- and region-specific alterations in gene transcript levels in APP/PS1 mice.

In some AD mouse models and humans, sex-specific differences in transcriptome profiles have been partially documented. Transcriptional human-mouse intersections are mainly related to A�/� pathology and microglia and exhibit sex-dependent spatiotemporal signatures upon disease progression. The results from RNA-seq of human post mortem brain tissue stressed the relevance of microglial and inflammatory mechanisms in AD pathogenesis, among other pathways (73, 132-134). Notably, different from transcriptome profiles in human brains, mouse models allow more controlled experimental conditions that can be used to investigate causation, isolated effects of specific molecular or pharmacological interventions, and elaborate spatiotemporal alterations. Although transcriptional overlaps have been identified between AD mouse models and human brains in some studies (73, 135-137), other reports have questioned the overall degree of conservation (113, 138). Regardless of these discrepancies, sex/gender has been suggested as a strong modulator in AD etiopathogenesis (73, 139, 140).

Most studies carried out in APP/PS1 mice and other AD models do not provide sex-specific analysis of transcriptome data (see S Table 15). In the following, we have elaborated the sex-specific profile of the major DEGs detected in our study. Importantly, these gene candidates turned out to be signature genes (Fig 5E). In addition, many of them contribute to the devastating AD pathophysiology. However, some (as outline below) also exert counteracting, autoprotective effects.

Functional relevance of signature genes in the hippocampus of female APP/PS1 mice.

In our study design, 17 gene candidates turned out to be exclusively upregulated in the hippocampus of female APP/PS1 AD mice, i.e., Prss23, Slc38a5, Ifitm3, Icam1, Dclk1, Rfx3, Tubb6, Slfn5, Flnc, Apobec1, Lsp1, Myo1g, Vim, 5830408C22Rik, Ptgr1, Pla2g4a, Rab32. The latter are related to the following GO categories: epithelial cell development, membrane trafficking, supramolecular fiber organization, and the adaptive immune system. Two related signature genes are highlighted below:

Intercellular adhesion molecule-1 (Icam)

An interesting candidate in our study turned out to be Icam1, encoding for the intercellular adhesion molecule-1, a transmembrane glycoprotein that is reported to be overexpressed in many neuropathological settings. Icam1, which is known to play an important role in the immune response, is expressed in the CNS, particularly in astrocytes, microglial and endothelial cells in the white and gray matter of the human forebrain. Icam1 has been attributed to the etiopathogenesis of neurodegenerative, e.g., AD and neuropsychiatric disorders, due to its functional involvement in blood–brain barrier function and as a marker for inflammation (141). Interestingly, the Icam1 gene product was suggested to be neuroprotective and a potential candidate in cytokine-mediated therapy of AD (141). Studies in humans revealed that the Icam1 gene product was increased in the cerebrospinal fluid (CSF) in preclinical, prodromal, and dementia stages of AD, associated with cortical thinning and cognitive deterioration and could act as a biomarker for early neuroinflammation and an AD risk assessment factor (142). Although Icam1 was previously suggested to serve as a CSF biomarker for AD, we found Icam1 to be significantly upregulated (FC > 1.5) only in the hippocampus of female APP/PS1 mice and it turned out to be one of the differentially expressed fingerprint genes in this subgroup.

Tubulin beta 6 (Tubb6)

Tubb6 encodes for a tubulin isotype, i.e., tubulin beta 6 class V with GTP binding activity, a constituent of the cytoskeleton being involved in microtubule and actin dynamics. Microtubule dysfunction is a prominent feature in many neurodegenerative diseases including AD (143). Tubb6 mRNA was previously detected in the hippocampus of C57BL/6 mice and mRNA levels were reported to be low and to further decrease in early age (144). Proteome studies indicated an overrepresentation of tubulin beta 6 in mouse models of AD (145). Importantly, in our transcriptome studies, Tubb6 was only upregulated in the hippocampus of female APP/PS1 mice, not in the RS cortex and not in male APP/PS1 mice.

Functional relevance of signature genes in the RS cortex of female APP/PS1 mice.

In the RS cortex of female APP/PS1 AD mice, 25 candidates exhibited signature character, i.e., Gusb, Slc14a1, Mir142hg, Cyth4, Gpr34, Plek, Ccl9, Aif1, Psmb9, Lair1, Hk3, Csf2rb, Tlr12, Adora3, Apbb1ip, Pycard, Vav1, Oas1a, Ifi27l2a, Hexb, Ncf4, Fcrls, Irf5, Havcr2, Mlxipl. These specific DEGs turned out be related to the fields of inflammatory response, Tyrobp causal network in microglia, neutrophil degranulation, phagocytosis and the RAF/MAP kinase cascade. Some of the related gene candidates and their implications in AD are further illuminated below:

Pleckstrin (Plek)

A most interesting DEG selectively upregulated in the RS cortex of female APP/PS1 mice turned out to be Plek. Pleckstrin is an important protein in cytoskeleton reorganization and neurite outgrowth and is supposed to be of functional relevance in AD pathophysiology as the structural and functional integrity/dynamics of the cytoskeleton are a prerequisite for neuronal function and survival (146, 147). Pleckstrin (Plek) turned to be a negatively regulated target of miR-409-5p, a miRNA that impairs neurite outgrowth. Interestingly, Aβ1-42 peptides significantly decrease cortical expression of miR-409-5p, resulting in less decrease of neuronal viability, and reduction in subsequent Aβ1-42-induced pathologies (147). In contrast, overexpression of miR-409-5p exerts neurotoxic effects in neuronal cells and impairs differentiation. Overexpression of Plek, on the other hand, harbors the capability to rescue neurite outgrowth from this neurotoxicity (147). It has previously reported that miR-409-5p is early downregulated in APP/PS1 mice and that Plek is upregulated at 9 and 12, but not 4 and 6 months-old APP/PS1 animals (147). Our study reveals that Plek is upregulated in 8 months old animals, and according to our transcriptome results, a signature gene in the RS cortex of female APP/PS1 mice.

Strikingly, the previously reported early downregulation of miR-409-5p in combination with the observed upregulation of Plek in AD progression might thus point to a counteracting, compensatory and autoprotective reaction to alleviate the structural and functional synaptic impairment induced by Aβ. Plek and its gene product Pleckstrin could thus serve as potential early biomarker in AD.

Adenosine receptor type 3 (Adora 3)

Some upregulated genes, such as Adora 3, are not only involved in microglia activation but also neutrophil degranulation and leucocyte migration/chemotaxis. Adora3 encodes a G-protein coupled adenosine receptor type 3, which is supposed to mediate inhibition of neutrophil degranulation and therefore potentially prevents cell damage. However, it was also reported to exhibit ambiguous effects in neuroprotection and neurodegeneration (108, 148, 149) stressing that modulation of the adenosinergic system in AD, in terms of pathology and therapeutics, is a sophisticated issue (150). Despite this potential intractability and the lack of medicines targeting adenosinergic receptors for AD treatment, substantial efforts are made in drug research and development. Importantly, Adora3 turned out to be a unique fingerprint candidate for the RS cortex of female APP/PS1 mice in our study. This sex-specific and potentially neuroprotective profile of Adora3 warrants future attention also in drug development and potential individualized pharmaceutical treatment in AD.

Vav Guanine Nucleotide Exchange Factor 1 (Vav1)

Vav1 encodes a guanine nucleotide exchange factor (GEF) for Rho family GTPases that activate pathways leading to actin cytoskeletal rearrangements and transcriptional alterations. It serves as a functional regulator of Rac2. It acts upstream of or within several processes, including integrin-mediated signaling pathway and neutrophil chemotaxis (151). Vav1 also seems to promote learning by activating HIF-1 and GLUT-1 and thereby contributing to glucose distribution to the brain, another important factor in AD (152). Interestingly, Vav1 turned out to be a unique fingerprint DEG for the RS cortex of female APP/PS1 mice.

Apoptosis inducing factor (Aif)

The apoptosis inducing factor (Aif) is responsible for caspase-independent programmed cell death (153). Aif was reported to mediate cell death in the TgCRND8 AD mouse model in a region-specific manner with aging-related cell death in the cortex but not the hippocampus (153). Notably, this holds true for our findings in APP/PS1 AD mice as well. Indeed, Aif turned out to be a unique fingerprint gene in the cortex of APP/PS1 mice, however, only in females and not in males – a phenomenon that has not been specified before (153).

Hexosaminidase subunit beta (Hexb)

The gene product of Hexb (hexosaminidase subunit beta) has been associated with AD based on genome-wide association studies (GWAS) (154). In addition, �-hexosaminidase can lead to a reduction of A� complexes, the aggregation and accumulation of which has been accelerated by exposure to gangliosides (155). Importantly, Hexb was unique for the RS cortex of female APP/PS1 mice.

Proteasome 20S subunit beta 9 (Psmb9)

The gene products of Psmb9 and Psmb8 (encoding for proteasome 20S subunit beta 9 and 8 respectively) form part of a proteasome (prosome, macropain) core complex which plays an important role in AD (156). The ubiquitin-proteasome system (UPS) is a major regulator of protein homeostasis. In contrast, the immunoproteasomes constitute a specialized form of proteasomes, which modulate inflammatory processes in AD via clearance of oxidant-damaged proteins. In APP/PS1 mice, Aβ-deposition was reported to parallel upregulation of immunoproteasomes. The latter turned out to be upregulated by pro-inflammatory cytokines, e.g., type I and type II interferons (IFNs). Wagner et al. (2017) suggested that immunoproteasomes are upregulated in the innate immune response towards extracellular Aβ-accumulation, although previous rodent studies revealed inconsistent data with decreased, unchanged, and increased proteasomal activity with aging (156-158). Wagner et al. (2017) used both males and females. However, they did not analyze for sex-specific differences (156). Strikingly, Psmb8 was upregulated only in the RS cortex and hippocampus of females and not in males in our study. In contrast, Psmb9 evolved as a unique fingerprint gene for the RS cortex of female APP/PS1 mice. The potential neuroprotective upregulation of Psmb9 thus seems to be attributable to the predilected brain region and sex in AD.

PYD and CARD domain containing gene (Pycard)

Pycard (PYD and CARD domain containing gene, encoding for adapter protein apoptosis-associated speck-like protein containing a CARD (ASC)) codes for an inflammasome component and is involved in regulation of autophagy. Previous reports demonstrated that released ASC specks can bind to Aβ, enhance its aggregation, and increase its toxicity (159). In our study, Pycard turned out to be a unique fingerprint for the RS cortex of female APP/PS1 mice. As Pycard was recently suggested to be a promising diagnostic target in early AD patients (160), sex-specific characteristics require further attention.

G protein-coupled receptor 34 (Gpr34)

The G-protein-coupled receptor 34 (Gpr34) was found to be highly expressed in the hippocampus of APP/PS1 mice (161) and to be involved in microglia phagocytic activity, complement activation and synaptic pruning (174). In vitro and in vivo Gpr34 knockdown approaches resulted in decreased levels of TNF-α, IL-1β, IL-6 and iNOS and suppression of ERK/NF-κB signal activation. Similarly, Gpr34-overexpression resulted in activation of ERK/NF-κB signalling and upregulation of TNF-α, IL-1β, IL-6 and iNOS. Systemically, Gpr34 knockdown relieved cognitive deficits in APP/PS1 mice and limited neuroinflammation and microglial activation, most probably via the ERK/NF-κB pathway (161).

Functional implications of signature genes in the hippocampus and RS cortex of male APP/PS1 mice.

In the hippocampus of male APP/PS1 AD mice, exclusive upregulation was observed in 18 gene candidates, i.e., Sgk1, Ncf1, Adamtsl4, Hck, St8sia6, Gm20743, Irf9, Pfkfb3, B2m, Arrdc2, Cebpa, Synpo2, Cep126, Rhoh, Hist1h2be, Clec5a, Cd86, Ptpn18. Some are discussed below in more detail:

Hematopoietic cell kinase (Hck)

The hematopoietic cell kinase encoded by HcK serves as a factor of the innate immune system. It has previously been demonstrated that the Hck pathway exerts important functions in the regulation of microglial neuroprotective functions during the early stage of AD (162). Strikingly, hematopoietic cell kinase deficiency aggravated cognitive impairment along with elevated Aβ levels and plaque formation. It further attenuated Aβ phagocytosis and enhanced iNOS expression in microglia (163). Thus, Hck is likely to exert a prominent neuroprotective function via modulation of microglial function and could attenuate early AD development.

Serum- and glucocorticoid-inducible kinase 1 (Sgk1)

Serum- and glucocorticoid-regulated kinase 1 (encoded by Sgk1) is a serine/threonine protein kinase that activates certain K+, Na+, and Cl- channels, involved in cell survival and neuronal excitability (164). SGK1 is engaged in various neurodegenerative pathways related to, e.g., apoptosis, autophagy, neuroinflammation, and ion channel regulation. Interestingly, hippocampal overexpression of Sgk1 improved spatial memory, reduced the devastating impact of Aβ accumulation and rescued actin cytoskeleton polymerization in middle-aged APP/PS1 mice (164, 165). It has been speculated that Sgk1 could exert a protective role against oxidative stress and play an antiapoptotic role (166). Whether pharmacological stimulation of serum- and glucocorticoid-regulated kinase 1 is protective in AD as well needs to be determined in the future.

Synaptopodin (Synpo2)

Synaptopodin serves as an actin-binding protein that is tightly associated with the spine apparatus and plays an important role in synaptic plasticity (167). Synaptopodin forms clusters in spines and regulates Ca2+ release from internal stores via ryanodin receptors in dendritic spines (167, 168). Deficiency of synaptopodin resulted in LTP deficits, impaired spatial memory and a lack of synaptic plasticity (167, 169, 170). Interestingly, synaptopodin expression was reported to be downregulated in the hippocampus of patients suffering from dementia with Lewy bodies, Parkinson's disease, mild cognitive impairments, and AD (170, 171). However, we observed a selective upregulation of synaptopodin in the hippocampus of male APP/PS1 mice. The later might point to a potential autoprotective role of Synpo2 in this experimental setting.

It should be emphasized that impairment of the structural and functional integrity of synapses is a hallmark in AD pathophysiology. Aβ causes disruption of NMDA and AMPA receptors, Ca2+ dyshomeostasis, reduced synaptic plasticity, suppression of LTP and aggravated LTD. Microglia/astrocyte activation, cytokine release, mitochondrial disruption, impairment of the cytoskeleton organization and deficits in energy metabolism further enhance synaptic dysfunction (120). Mechanistically, a number of synapse-related target molecules and signaling pathways are related to these phenomena, e.g., Wnt/β-catenin, IKK/NF-κB, JAK2/STAT3, JNK, Akt, MAPK, caspase-3, GSK-3β and CDK-5 (120). Many DEGs identified in our study (see above) enhance synaptic dysfunction, e.g. PrPC, whereas others, such as synaptopodin might exert synaptoprotective effects.

Interestingly, only two candidates, Derl3, and Mki67, exhibited unique upregulation in the RS cortex of male APP/PS1 AD animals (Fig 5E).

Derlin (Derl3)

Derlin 3 (encoded by Derl3) resides in the endoplasmic reticulum (ER) and is involved in the degradation of mis-folded glycoproteins in the ER. Dysfunction of Derlin 1 and 2 have previously been related to neurodegenerative diseases (75).

Ki-67 (Mki67)

Mki67 encoding for Ki-67 serves as cell proliferation marker and is maximally expressed in G2 phase and mitosis. Interestingly, previous studies identified MKi67 (and also Top2a, Mcm2, Tubb5) to be enriched in DNA replication and chromatin rearrangement in a specific subgroup of microglial cells, i.e., cycling/proliferating microglia (CPMs), that contributes only 0.3%–1.2% of the total microglial population (76). It has further been suggested that Ki-67 is involved in the pathogenesis of neurofibrillary degeneration in AD (172).

As outlined previously, the number of downregulated genes in the individual subgroups turned out to be significantly lower compared to the upregulated ones (Fig 6). However, it was possible to characterize unique profiles of downregulated genes in the individual subgroups. In the hippocampus of female APP/PS1 AD mice, these signature genes included 13 candidates (Asic4, Pcsk1, Etnppl, Myh7b, Spag5, Gpr1, Myh8, Ccdc184, Pdzph1, Trim66, BC030499, Rims3, Prr16), only two candidates (LOC105247294, Ptpn6) in the RS cortex of female APP/PS1 AD mice, six gene candidates in the hippocampus of male APP/PS1 AD mice (4933407I18Rik, Hlcs, Hdac9, Pla2g4e, Wnt9a. Shisa9) and four candidates in the RS cortex of male APP/PS1 AD animals (6530402F18Rik, Ctr9, Scyl2, 1700001L05Rik) (Fig 6E).

Notably, the number of intersectional, upregulated genes was much higher in the RS cortex of APP/PS1 mice compared to the hippocampus (52 versus 32, Fig 5C, D). Many of these additional, intersectional candidates in the RS cortex (Fig 5C) are recruited from gene candidates that turned out to be selectively expressed in the male or female hippocampus (Fig 5D). Consequently, more gene candidates are specifically upregulated in the female hippocampus compared to the RS cortex (45 vs. 39) and the male hippocampus compared to the RS cortex (25 vs. 5). Thus, sex-specific differences in DEGs are more overt in the hippocampus than in the RS cortex.

It turned out that several DEGs are region-specific as well. It has been shown for example, that microglia in different brain regions can differ in gene expression, particularly in genes related to bioenergetic and immunoregulatory pathways (173). Interestingly, the number of co-upregulated DEGs in the RS cortex and hippocampus of females was much higher than the related number in male APP/PS1 mice (55 vs. 33, Fig 5).

These findings are supported by symptomatic sex dimorphism profiling in 12 months old male and female APP/PS1 AD mice performed by Jiao et al. (2016) (139). Indeed, there is increasing evidence, that sex-biased and BROI-specific patterns may originate from differences in vulnerabilities/susceptibilities and/or resilience in different brain regions at different AD progression states. Female APP/PS1 mice exhibited a higher parenchymal A� load compared to male APP/PS1 animals. The latter was most prominent in the hippocampus (139). In addition, cerebral amyloid angiopathy and related microhemorrhage was more frequent in female APP/PS1 mice. Although top-ranked, intersectional genes associated with neuroinflammation, microglia activation and immune response are found in both sexes and both the RS cortex and hippocampus, specific gene sets exert fingerprint character in the individual subgroups. Notably, related parameters, including levels of phosphorylated �, proinflammatory cytokines, microgliosis, astrocytosis, and synaptic/neuronal disintegration were particularly enhanced in female APP/PS1 animals (139). Some DEGs that predominate in females are related to mitochondria and ROS. In females, mitochondrial function seems to be more resistant against Aβ mediated neurotoxicity. This phenomenon is probably due to a reduction of ROS and suppression of apoptogenic signals via estrogen (174). This may be one reason why females suffer more from AD with decreasing estrogen levels following menopause. Brain-region und sex-related specificities in DEGs, as observed in our study, might also be due to differences in brain region connectivity/architecture between males and females. The latter suggests that network characteristics in females favor a more rapid spread of neurofibrillary tangles in the CNS (175). In addition, different BROIs are disproportionally modulated via complex spatiotemporal activity of sex hormones (176-178).

In summary, the determination of sex- and brain region-specific transcriptional profiles as presented here, is highly necessary for future characterization and validation of DEGs as potential biomarkers and personalized medicinal approaches.”

We have further added a new, brief paragraph elaborating the co-appearance of disease-aggravating and counteracting, self-protective DEGs in our APP/PS1 subgroups:

“Disease aggravating versus neuroprotective DEGs in APP/PS1 mice.

Our transcriptome studies in APP/PS1 AD mice have revealed a high number of significantly upregulated and downregulated gene candidates. Whereas many of them turned out to be intersectional, some exhibited signature characteristics. It’s interesting to note that not all DEGs contribute to the devastating complex pathophysiology and pathobiochemistry of AD. Indeed, some candidates, such as Plek, Adora3 and Psmb9 in the female RS cortex, Icam1 in the female hippocampus, and Hck and Sgk1 in the male hippocampus of APP/PS1 mice seem to exert counteracting, autoprotective functions in the individual subgroups. Some intersectional genes such as, Ccl6, Lyz2 and Nrf2 also exhibit self-protective effects. They clearly deserve special attention in drug research and development in the future.”

Point 5: In synaptic plasticity, upregulation of certain genes can lead to increase in synaptic transmission, while upregulation of some genes can lead to downregulation of synaptic plasticity. Please discuss.

Response: We very much appreciate the reviewer’s comment. Please note that we have pointed out the functional implications in synaptic transmission in the discussion of the individual DEGs (see also our response to Point 4). In addition, we have specifically addressed the aspect of altered transcription of synaptic plasticity related genes in the discussion of synaptopodin (Synpo2):

“ … It should be emphasized that impairment of the structural and functional integrity of synapses is a hallmark in AD pathophysiology. Aβ causes disruption of NMDA and AMPA receptors, Ca2+ dyshomeostasis, reduced synaptic plasticity, suppression of LTP and aggravated LTD. Microglia/astrocyte activation, cytokine release, mitochondrial disruption, impairment of the cytoskeleton organization and deficits in energy metabolism further enhance synaptic dysfunction (120). Mechanistically, a number of synapse-related target molecules and signaling pathways are related to these phenomena, e.g., Wnt/β-catenin, IKK/NF-κB, JAK2/STAT3, JNK, Akt, MAPK, caspase-3, GSK-3β and CDK-5 (120). Many DEGs identified in our study (see above) enhance synaptic dysfunction, e.g. PrPC, whereas others, such as synaptopodin might exert synaptoprotective effects.”

Point 6: APP/PS1 is a model where pathology also involves APP over production, so how would you negate its effects?

Response: We very much appreciate the reviewer’s comment. We agree with the reviewer that APP overproduction in APP/PS1 pathophysiology warrants specific attention. Indeed, the overproduction of APP and the resultant increase in diverse APP cleavage products (not only A�1-42 for example� potentially serves as a confounding factor, not only in our transcriptome analysis, but in all studies carried out in so called first generation transgenic APP AD mouse models. Therefore, interference with this phenomenon cannot be negated.

In the revised version of our manuscript, we have elaborated the potential effects of APP overproduction on our Discussion and shed light on the pros and cons of first generation AD models (with APP overproduction) and second generation APP knock-in models (without APP overproduction).

The following related paragraph was added to the revised Discussion section:

“Influence of APP overproduction and transcriptome profiles.

Overexpression of WT APP and mutant APP variants has been used to establish many well-characterized transgenic AD mouse lines, including APP/PS1 (184). It’s noteworthy that APP overexpression in these first-generation transgenic mouse models can cause overproduction of different APP fragments in addition to A The latter can make it challenging to distinguish the functional impact of these fragments and might affect the translational relevance of such models (185, 186). Potentials limitations of mutant APP- and APP/PS-overexpressing mouse models, such as APP/PS1 include, e.g., the lack of non-coding regions of the APP gene that affect splicing of APP mRNA and transcriptional regulation, unphysiological interaction of overexpressed APP and overproduced non A� fragments with cellular proteins such as kinesin via JIP-1, non-specific ER impairment upon APP/PS-overexpression, appearance of A� entities that are different from those identified in clinical AD brain, different region specificity of A� pathology, and inconsistent drug effects (187). Notably, second-generation mouse models are likely to overcome these intrinsic limitations of the APP overexpression paradigm as they utilize an APP knock-in strategy in order to selectively overexpress, e.g., A�42, but not APP. The latter was achieved, e.g., in single APP knock-in mouse models with a murine A� sequence carrying three humanized amino acids that differ between mice and humans (187). However, these models are facing numerous limitations as well (187). Thus, it’s important to be aware of the potential interference of APP- or APP/PS-overexpression with AD disease pathology in first generation AD mouse models (187).”

Point 7: As some DEGs are exclusively upregulated or downregulated in males or females, this could be a target for gender specific medicine. Please discuss.

Response: We very much appreciate the reviewer’s comment. The brain-region and sex-specific gene profiles are a central aspect of our manuscript. We have pointed out the aspect of sex- / gender-specific in the original submission. We further elaborated the aspect in the revised version of the Discussion section and stressed the relevance for gender-specific / personalized medicine in more detail in the final Conclusion section:

“Conclusions.

Our APP/PS1 AD model-based data support the hypothesis that a major response in the brain to A� accumulation is related to microglia activation, immune response and neuroinflammation. Early and profound neuroinflammatory/immune response seems to be a fundamental driving force in AD pathogenesis. Many DEGs were claimed to be involved in these processes, however, sex- and region-specific differentiation of their functional relevance is rare. Our data clearly illustrate that sex-specific differences in the transcriptome profiles are of major relevance and that some DEGs, i.e., signature genes, are exclusively upregulated or downregulated in males or females (Table 3, 4). So far, numerous defective genes, transcriptional and translational alterations were shown to contribute to AD pathogenesis, further influenced by diverse environmental factors and epigenetic phenomena. The consideration of sex, age, comorbidity factors, disease progression state and most important, the individual genomic profile, are mandatory for a valid, promising, personalized pharmacotherapeutic approach in the future. The latter will require distinct protocols and strategies in drug research and development, the planning of clinical trials to optimize therapeutics and the establishment of new diagnostic approaches. The analysis of the patient’s individual genome including gene sets associated with disease pathogenesis, mechanisms of drug action, drug metabolism, drug transporters and multifaceted cascades and metabolic pathways, will dramatically facilitate the personalized therapeutic approach in AD.”

Please note that we also introduced this aspect in the last sentence of the Abstract:

“The latter will be of central relevance in future preclinical and clinical AD related studies, biomarker characterization and personalized medicinal approaches.”

And the importance of the personalized medicinal approach is also mentioned in the Introduction section:

“The latter has substantial implications for AD drug research and development and individualized/personalized pharmacological AD treatment.”

And also the Results section:

“In summary, the determination of sex- and brain region-specific transcriptional profiles as presented here, is highly necessary for future characterization and validation of DEGs as potential biomarkers and personalized medicinal approaches.”

Finally, please note that we improved the layout of Figure 1-4, and that we performed new qPCRs to validate some DEGs (see new Figure 7, S Table 4 and S Table 14).

 

References

1. Papazoglou A, Henseler C, Weickhardt S, Daubner J, Schiffer T, Broich K, et al. Whole genome transcriptome data from the WT cortex and hippocampus of female and male control and APP/PS1 Alzheimer's disease mice. Data Brief. 2023;50:109594.

2. The RC, Petrov AI, Kay SJE, Kalvari I, Howe KL, Gray KA, et al. RNAcentral: a comprehensive database of non-coding RNA sequences. Nucleic Acids Res. 2017;45(D1):D128-D34.

3. Kalvari I, Nawrocki EP, Argasinska J, Quinones-Olvera N, Finn RD, Bateman A, Petrov AI. Non-Coding RNA Analysis Using the Rfam Database. Curr Protoc Bioinformatics. 2018;62(1):e51.

4. Wang L, Park HJ, Dasari S, Wang S, Kocher JP, Li W. CPAT: Coding-Potential Assessment Tool using an alignment-free logistic regression model. Nucleic Acids Res. 2013;41(6):e74.

5. Zhao L, Wang J, Li Y, Song T, Wu Y, Fang S, et al. NONCODEV6: an updated database dedicated to long non-coding RNA annotation in both animals and plants. Nucleic Acids Res. 2021;49(D1):D165-D71.

6. Briggs JA, Wolvetang EJ, Mattick JS, Rinn JL, Barry G. Mechanisms of Long Non-coding RNAs in Mammalian Nervous System Development, Plasticity, Disease, and Evolution. Neuron. 2015;88(5):861-77.

7. Li D, Zhang J, Li X, Chen Y, Yu F, Liu Q. Insights into lncRNAs in Alzheimer's disease mechanisms. RNA Biol. 2021;18(7):1037-47.

8. Ni Y, Huang H, Chen Y, Cao M, Zhou H, Zhang Y. Investigation of Long Non-coding RNA Expression Profiles in the Substantia Nigra of Parkinson's Disease. Cell Mol Neurobiol. 2017;37(2):329-38.

9. Lyu Y, Bai L, Qin C. Long noncoding RNAs in neurodevelopment and Parkinson's disease. Animal Model Exp Med. 2019;2(4):239-51.

10. Johnson R. Long non-coding RNAs in Huntington's disease neurodegeneration. Neurobiol Dis. 2012;46(2):245-54.

11. Lan Z, Chen Y, Jin J, Xu Y, Zhu X. Long Non-coding RNA: Insight Into Mechanisms of Alzheimer's Disease. Front Mol Neurosci. 2021;14:821002.

12. Mercer TR, Dinger ME, Mattick JS. Long non-coding RNAs: insights into functions. Nat Rev Genet. 2009;10(3):155-9.

13. Wilusz JE, Sunwoo H, Spector DL. Long noncoding RNAs: functional surprises from the RNA world. Genes Dev. 2009;23(13):1494-504.

14. Lee DY, Moon J, Lee ST, Jung KH, Park DK, Yoo JS, et al. Distinct Expression of Long Non-Coding RNAs in an Alzheimer's Disease Model. J Alzheimers Dis. 2015;45(3):837-49.

15. Fang M, Zhang P, Zhao Y, Liu X. Bioinformatics and co-expression network analysis of differentially expressed lncRNAs and mRNAs in hippocampus of APP/PS1 transgenic mice with Alzheimer disease. Am J Transl Res. 2017;9(3):1381-91.

16. Paesler K, Xie K, Hettich MM, Siwek ME, Ryan DP, Schroder S, et al. Limited effects of an eIF2alphaS51A allele on neurological impairments in the 5xFAD mouse model of Alzheimer's disease. Neural Plast. 2015;2015:825157.

17. Siwek ME, Muller R, Henseler C, Trog A, Lundt A, Wormuth C, et al. Altered theta oscillations and aberrant cortical excitatory activity in the 5XFAD model of Alzheimer's disease. Neural Plast. 2015;2015:781731.

18. Papazoglou A, Soos J, Lundt A, Wormuth C, Ginde VR, Muller R, et al. Gender-Specific Hippocampal Dysrhythmia and Aberrant Hippocampal and Cortical Excitability in the APPswePS1dE9 Model of Alzheimer's Disease. Neural Plast. 2016;2016:7167358.

19. Papazoglou A, Soos J, Lundt A, Wormuth C, Ginde VR, Muller R, et al. Motor Cortex Theta and Gamma Architecture in Young Adult APPswePS1dE9 Alzheimer Mice. PLoS One. 2017;12(1):e0169654.

20. Schmittgen TD, Livak KJ. Analyzing real-time PCR data by the comparative C(T) method. Nat Protoc. 2008;3(6):1101-8.

21. Ricciarelli R, d'Abramo C, Massone S, Marinari U, Pronzato M, Tabaton M. Microarray analysis in Alzheimer's disease and normal aging. IUBMB Life. 2004;56(6):349-54.

22. Loring JF, Wen X, Lee JM, Seilhamer J, Somogyi R. A gene expression profile of Alzheimer's disease. DNA Cell Biol. 2001;20(11):683-95.

23. Castillo E, Leon J, Mazzei G, Abolhassani N, Haruyama N, Saito T, et al. Comparative profiling of cortical gene expression in Alzheimer's disease patients and mouse models demonstrates a link between amyloidosis and neuroinflammation. Sci Rep. 2017;7(1):17762.

24. Wang C, Zong S, Cui X, Wang X, Wu S, Wang L, et al. The effects of microglia-associated neuroinflammation on Alzheimer's disease. Front Immunol. 2023;14:1117172.

25. Sala Frigerio C, Wolfs L, Fattorelli N, Thrupp N, Voytyuk I, Schmidt I, et al. The Major Risk Factors for Alzheimer's Disease: Age, Sex, and Genes Modulate the Microglia Response to Abeta Plaques. Cell Rep. 2019;27(4):1293-306 e6.

26. Gatta V, D'Aurora M, Granzotto A, Stuppia L, Sensi SL. Early and sustained altered expression of aging-related genes in young 3xTg-AD mice. Cell Death Dis. 2014;5(2):e1054.

27. Zaletel I, Schwirtlich M, Perovic M, Jovanovic M, Stevanovic M, Kanazir S, Puskas N. Early Impairments of Hippocampal Neurogenesis in 5xFAD Mouse Model of Alzheimer's Disease Are Associated with Altered Expression of SOXB Transcription Factors. J Alzheimers Dis. 2018;65(3):963-76.

28. Wu ZL, Ciallella JR, Flood DG, O'Kane TM, Bozyczko-Coyne D, Savage MJ. Comparative analysis of cortical gene expression in mouse models of Alzheimer's disease. Neurobiol Aging. 2006;27(3):377-86.

29. Rothman SM, Tanis KQ, Gandhi P, Malkov V, Marcus J, Pearson M, et al. Human Alzheimer's disease gene expression signatures and immune profile in APP mouse models: a discrete transcriptomic view of Abeta plaque pathology. J Neuroinflammation. 2018;15(1):256.

30. Audrain M, Fol R, Dutar P, Potier B, Billard JM, Flament J, et al. Alzheimer's disease-like APP processing in wild-type mice identifies synaptic defects as initial steps of disease progression. Mol Neurodegener. 2016;11:5.

31. Palop JJ, Mucke L. Network abnormalities and interneuron dysfunction in Alzheimer disease. Nat Rev Neurosci. 2016;17(12):777-92.

32. Saito T, Matsuba Y, Mihira N, Takano J, Nilsson P, Itohara S, et al. Single App knock-in mouse models of Alzheimer's disease. Nat Neurosci. 2014;17(5):661-3.

33. Sasaguri H, Nilsson P, Hashimoto S, Nagata K, Saito T, De Strooper B, et al. APP mouse models for Alzheimer's disease preclinical studies. EMBO J. 2017;36(17):2473-87.

---

## [Editor Report · Decision Letter 1]

21 Dec 2023

Sex- and region-specific cortical and hippocampal whole genome transcriptome profiles from control and APP/PS1 Alzheimer’s disease mice

PONE-D-23-30186R1

Dear Dr. Weiergräber,

We’re pleased to inform you that your manuscript has been judged scientifically suitable for publication and will be formally accepted for publication once it meets all outstanding technical requirements.

Kind regards,

Stephen D. Ginsberg, Ph.D.

Section Editor

PLOS ONE

---

## [Editor Report · Acceptance letter]

25 Jan 2024

PONE-D-23-30186R1 

PLOS ONE

Dear Dr. Weiergräber, 

I'm pleased to inform you that your manuscript has been deemed suitable for publication in PLOS ONE. Congratulations! Your manuscript is now being handed over to our production team.

Kind regards, 

on behalf of

Dr. Stephen D. Ginsberg 

Section Editor

PLOS ONE